# A Comprehensive Analysis on the Learning Curve in Kernel Ridge Regression

**Tin Sum Cheng, Aurelien Lucchi**
Department of Mathematics and Computer Science
University of Basel, Switzerland
tinsum.cheng@unibas.ch, aurelien.lucchi@unibas.ch

**Anastasis Kratsios**
Department of Mathematics
McMaster University and The Vector Institute
Ontario, Canada
kratsioa@mcmaster.ca

**David Belius**
Faculty of Mathematics and Computer Science
UniDistance Suisse
Switzerland david.belius@cantab.ch

## Abstract

This paper conducts a comprehensive study of the learning curves of kernel ridge regression (KRR) under minimal assumptions. Our contributions are three-fold: 1) we analyze the role of key properties of the kernel, such as its spectral eigen-decay, the characteristics of the eigenfunctions, and the smoothness of the kernel; 2) we demonstrate the validity of the Gaussian Equivalent Property (GEP), which states that the generalization performance of KRR remains the same when the whitened features are replaced by standard Gaussian vectors, thereby shedding light on the success of previous analyzes under the Gaussian Design Assumption; 3) we derive novel bounds that improve over existing bounds across a broad range of setting such as (in)dependent feature vectors and various combinations of eigen-decay rates in the over/underparameterized regimes.

## 1 Introduction

Kernel ridge regression (KRR) is a central tool in machine learning due to its ability to provide a flexible and efficient framework for capturing intricate patterns within data. Additionally, it stands as one of the earliest endeavors in statistical machine learning, with ongoing research into its generalization properties [12, 45]. Over the past few years, kernels have experienced a resurgence in importance in the field of deep learning theory [49, 8, 6], partly because many deep neural networks (DNNs) can be interpreted as approaching specific kernel limits as they converge [25, 1, 9].

One central topic in machine learning theory is the **learning curve** of the regressor in the fixed input dimensional setting as the sample size grows to infinity. Formally: let $n$ be the sample size, $\lambda = \lambda(n)$ be the ridge regularization parameter depending on $n$ and $\mathcal{R}_n$ be the test error/excess risk of the ridge regression. For large $n$, the test error $\mathcal{R}_n$ should decay with $n$ as $\mathcal{R}_n = \mathcal{O}_{n,\mathbb{P}}(g(n))$ for some function $g : \mathbb{R} \to \mathbb{R}$ such that $g(n) \xrightarrow{n \to \infty} 0$. The decay of $g$ with respect to $n$ provides an upper bound on the learning curve of the ridge regressor and will be the main focus of this paper. To conduct our analysis, we concentrate on several crucial properties of the kernel, including its spectral eigen-decay and the characteristics of the eigenfunctions, which we will elaborate on next.

**Properties of the eigenfunctions** In a series of studies [10, 16, 34], the feature vectors are substituted by random Gaussian vectors following the Gaussian Design (GD) Assumption, and the

38th Conference on Neural Information Processing Systems (NeurIPS 2024).

learning curve is derived using the Replica method. In a separate body of research [29, 30, 31], it is demonstrated that *similar* learning curves occur for Hölder continuous kernels under an assumption called the Embedding Condition (EC) (see Section A for more details). Consequently, there is a fundamental mismatch between the distribution of the feature vector and the Gaussian random vectors used in [10, 16, 34]: in the former case, each coordinate is highly dependent on the others, whereas in the Gaussian Design case, each coordinate operates independently from the others. Astonishingly, however, both settings share similar learning curves. This phenomenon, initially identified by [21, 3, 42, 20], is termed the *Gaussian Equivalence Property*. This prompts the question:

*Q1: When and why does the Gaussian Equivalence Property exist?*

**Spectral eigen-decay**    Many recent papers [10, 35, 44, 36] have attempted to characterize the test error solely by the (kernel) eigenspectrum decay. It is for instance common to differentiate between different eigenspectrum decays: [29, 30, 31] assumes the embedding condition (EC) and Hölder continuity to kernel with polynomial eigen-decay; [33] assumes either polynomial or exponential eigen-decay (with noiseless labels) under the Maximum Degree-of-Freedom (MaxDof) Assumption; [36] assumes some concentration and the so-called hypercontractivity on eigenfunctions.

However, [14] pointed out that the characterization of generalization performance solely by the spectral eigen-decay might oversimplify the generalization performance of ridge regression. In relation to the second question, we further ask:

*Q2: Under what conditions is the generalization error fully determined by the eigen-decay?*

**Additional assumptions and settings**    In addition to the two properties above, other hyperparameters or settings, including capacity of the kernels/feature vectors, the ridge regularization decay, the source condition of the target function, the noise level in the output label, and the amount of over-parameterization, play an important role in the analysis of the learning curve of ridge regression. Within the present body of research, various papers establish bounds on the test error of KRR across diverse assumptions and settings (we refer the reader to Section A for further elaboration). It is therefore of significant interest to ask:

*Q3: Is there a unifying theory explaining the generalization under minimal assumptions?*

**Contributions**    We address questions *Q1-3* through the following contributions:

(i) **Unified theory:** We provide a unifying theory of the test error of KRR across a wide variety of settings (see Subsection 2.2 and Table 1 in Section 3).

(ii) **Validation and GEP:** We show that the generalization performance with independent (Gaussian) features and dependent (kernel) features coincides asymptotically and it solely depends on the eigen-decay under strong ridge regularization, hence validating the Gaussian Equivalent Property (GEP) (see Subsection 3.2) .

(iii) **New and sharpened bounds:** We provide novel bounds of the KRR test error that improve over prior work across various settings (see Subsections 3.2).

(iv) **Smoothness and generalization:** We relate the spectral eigen-decay to kernel smoothness (see Appendix B.2) and hence to the kernel's generalization performance.

## 2   Setting

In this section, we introduce the basic notation for ridge regression, which includes high-dimensional linear regression and kernel ridge regression as special cases.

### 2.1   Notations

Suppose $p \in \mathbb{N} \cup \{\infty\}$. Let $\mathbf{x} = (x_k)_{k=1}^{p} \in \mathbb{R}^p$ be a random (feature) vector sampled from some distribution $\mu$ on $\mathbb{R}^p$. Let $n \in \mathbb{N}$ be an integer and denote by $\mathbf{x}_1, ..., \mathbf{x}_n$ n i.i.d. draw of $\mathbf{x}$. Denote the input matrix $\mathbf{X} \in \mathbb{R}^{n \times p}$ to be a matrix with rows $\mathbf{x}_i^{\top}$. By fixing an orthonormal basis, we assume that the covariance matrix is diagonal:

$$\mathbf{\Sigma} \stackrel{\text{def.}}{=} \mathbb{E}_\mu \left[ \mathbf{x}\mathbf{x}^{\top} \right] = \text{diag}(\lambda_1, \lambda_2, ..., \lambda_p) \in \mathbb{R}^{p \times p},$$

where the eigenvalues $\lambda_1 \geq \lambda_2 \geq ... \geq \lambda_p > 0$ is a decreasing sequence of positive numbers. [1]

We also assume that $\mathbf{y} \in \mathbb{R}^n$ is an output vector such that

$$\mathbf{y} = \mathbf{X}\boldsymbol{\theta}^* + \boldsymbol{\epsilon} \tag{1}$$

where $\boldsymbol{\theta}^* \in \mathbb{R}^p$ is a deterministic vector, $\boldsymbol{\epsilon} \in \mathbb{R}^n$ is a random vector whose entries are i.i.d. drawn from a centered random variable $\epsilon$ with variance $\mathbb{E}\left[\epsilon^2\right] = \sigma^2 < \infty$ and independent to $\mathbf{X}$.

Then the linear regressor

$$\hat{\boldsymbol{\theta}}(\mathbf{y}) \stackrel{\text{def.}}{=} \mathbf{X}^\top (\mathbf{X}\mathbf{X}^\top + n\lambda\mathbf{I}_n)^{-1}\mathbf{y} \tag{2}$$

is the minimizer of the empirical mean square loss (MSE) problem:

$$\min_{\boldsymbol{\theta} \in \mathbb{R}^p} \frac{1}{n}\|\mathbf{X}\boldsymbol{\theta} - \mathbf{y}\|_2^2 + \lambda\|\boldsymbol{\theta}\|_2^2, \tag{3}$$

where $\lambda \geq 0$ is the ridge[2]. This paper focuses on bounding the test error, with which we can analyse the learning curve of the regressor. To do so, we use the following well-known bias-variance decomposition.

**Definition 2.1** (Bias-variance decomposition). Consider input-output pairs $(\mathbf{X}, \mathbf{y})$ of sample size $n$ and a ridge $\lambda \geq 0$. Define the test error $\mathcal{R}$ to be the population mean squared error between the regressor and the true label averaged over noise.

$$\mathcal{R} \stackrel{\text{def.}}{=} \mathbb{E}_{\mathbf{x}, \boldsymbol{\epsilon}}\left[\left(\mathbf{x}^\top\hat{\boldsymbol{\theta}}(\mathbf{y}) - \mathbf{x}^\top\boldsymbol{\theta}^*\right)^2\right] \tag{4}$$

Note that $\mathcal{R}$ is a random variable depending on the samples $(\mathbf{X}, \mathbf{y})$ and the ridge $\lambda \geq 0$. Hence, we can also view $\mathcal{R} = \mathcal{R}_n$ as a random variable indexed in $n$, where the samples $(\mathbf{X}, \mathbf{y})$ are $n$ i.i.d. drawn input-output pairs and $\lambda$ is chosen to depend on $n$.

We decompose the test error into a bias $\mathcal{B}$ and variance $\mathcal{V}$, which is typical for most KRR literature [32, 23, 6, 46, 29, 30, 31, 13, 14]:

$$\mathcal{R} = \mathcal{B} + \mathcal{V} \tag{5}$$

where $\mathcal{B} \stackrel{\text{def.}}{=} \mathbb{E}_{\mathbf{x}}\left[\left(\mathbf{x}^\top\hat{\boldsymbol{\theta}}(\mathbf{X}\boldsymbol{\theta}^*) - \mathbf{x}^\top\boldsymbol{\theta}^*\right)^2\right]$, $\mathcal{V} \stackrel{\text{def.}}{=} \mathbb{E}_{\mathbf{x}, \boldsymbol{\epsilon}}\left[\left(\mathbf{x}^\top\hat{\boldsymbol{\theta}}(\boldsymbol{\epsilon})\right)^2\right]$.

*Remark* 2.2 (Noiseless labels). If there is no noise in the label, that is, $\boldsymbol{\epsilon} = 0$, the test error $\mathcal{R}$ is simply the bias $\mathcal{B}$. Hence, the analysis of the bias term $\mathcal{B}$ in this paper is directly applicable to the noiseless labels setting.

We now summarize the combinations of assumptions and settings made in this paper.

## 2.2 Assumptions and Settings

**Polynomial/exponential eigen-decay**   We consider two types of spectral decay rates, namely, *polynomial* and *exponential* decay rates, because: 1) polynomial eigen-decay is, roughly speaking , equivalent to the case where the RKHS is comprised of at most finitely many continuous derivatives; 2) the exponential eigen-decay is, possibly up to a canonical change in the relevant function space, equivalent to the case where the RKHS consists of smooth (infinitely differentiable) functions. For the formal definition of the eigen-decay, see Assumptions (PE) and (EE). For further details and explanation on the relationship between eigen-decay and smoothness, we refer the reader to Section B.

**Source condition**   Many previous works [6, 16, 7, 30] include the so-called source condition as assumptions on the target. If the task is proper, that is, $\boldsymbol{\theta}^* \in \mathcal{H} = \mathcal{H}^1$, we have $s \geq 1$. More generally, a larger source coefficient $s$ implies a smoother target $\boldsymbol{\theta}^*$ in the RKHS $\mathcal{H}$.

---

[1]For $p = \infty$, we regard $\mathbf{x}$ as a vector of an infinite dimensional Hilbert space and $\boldsymbol{\Sigma}$ as a Hilbert Schmidt operator, meaning $\text{Tr}[\boldsymbol{\Sigma}] < \infty$. With abuse of notation, we write $\mathbf{x}$ as vector and $\boldsymbol{\Sigma}$ as matrix throughout the paper. In some proofs, we would apply some linear algebraic results on some $\mathbb{R}^{p \times p}$ matrices. When dealing with $p = \infty$, replace those results with their Hilbert space counterparts.

[2]For $\lambda$ without subscript, we mean the ridge regularization coefficient; for $\lambda_k$ with subscript $k$, we mean the eigenvalues of the covariance $\boldsymbol{\Sigma} = \mathbb{E}\left[\mathbf{x}\mathbf{x}^\top\right]$.

**Definition 2.3** (Interpolation space). Let $s \geq 0$ be a real number. Define the interpolation space

$$\mathcal{H}^s \overset{\text{def.}}{=} \{\boldsymbol{\theta} \in \mathbb{R}^p : \|\boldsymbol{\theta}\|_{\boldsymbol{\Sigma}^{1-s}} < \infty\}.$$

**Assumption (SC)** (Source Condition). *The* source coefficient *of a target coefficient* $\boldsymbol{\theta}^*$ *is defined as*

$$s = \inf\{t > 0 : \boldsymbol{\theta}^* \in \mathcal{H}^t\}.$$

See Subsection A.5 for more elaborations for the source coefficient $s$ in polynomial or in exponential decay.

**Strong/weak ridge** We set the ridge $\lambda = \lambda(n) \geq 0$ to depend on the sample size $n$. The ridge is considered strong (relative to the eigen-decay) if $\lambda \succcurlyeq \lambda_{\min\{n,p\}}$, that is, if $\lambda/\lambda_{\min\{n,p\}} \xrightarrow{n \to \infty} 0$; otherwise, it is considered weak. Intuitively, the ridge is weak when it is negligible compared to the entries in the kernel matrix, effectively making it ridgeless.

To summarize the assumptions discussed previously, let $(\lambda_k)_{k=1}^p$ be the eigenvalues of the kernel $K$, and $\boldsymbol{\theta}^* \overset{\text{def.}}{=} (\theta_k^*)_{k=1}^p$ the coefficients of the target function being learned in the eigen-basis defined by $K$. Then we assume either of the following assumptions:

**Assumption (PE)** (Polynomial Eigen-decay). *Assume that* $\lambda_k = \Theta_k\left(k^{-1-a}\right)$, $|\theta_k^*| = \Theta_k\left(k^{-r}\right)$, $\lambda = \Theta_n\left(n^{-b}\right)$ *for some constants* $a, b, r > 0$, *where* $a + 2 \neq 2r$ *unless specified.* [3] *Hence, if Assumption (SC) holds, the source coefficient is* $s = \frac{2r+a}{1+a}$. *We call the ridge* $\lambda$ *strong if* $b \in (0, 1+a]$, *and weak if* $b \in (1+a, \infty]$, *under the convention that* $b = \infty$ *implies* $\lambda = 0$.

**Assumption (EE)** (Exponential Eigen-decay). *Assume that* $\lambda_k = \Theta_k\left(e^{-ak}\right)$, $\theta_k^* = \Theta_k\left(e^{-rk}\right)$, $\lambda = \Theta_n\left(e^{-bn}\right)$ *for some constants* $a, b, r > 0$, *where* $a \neq 2r$ *unless specified.* [4] *Hence, if Assumption (SC) holds, the source coefficient is* $s = \frac{2r+a}{a} = \frac{2r}{a} + 1$. *We call the ridge* $\lambda$ *strong if* $b \in (0, a]$, *and weak if* $b \in (a, \infty]$, *under the convention that* $b = \infty$ *implies* $\lambda = 0$.

**Generic/independent features** Our analysis centers on the assumptions regarding feature vectors, with a focus on the dependencies between coordinates, particularly exploring two cases:

1. Generic features (GF): include the cases where the feature vectors are dependent on each other, for example, the feature vectors from the following kernels:
   - dot-product kernels on hyperspheres;
   - kernels with bounded eigenfunctions;
   - radial base function (RBF) and shift-invariant kernels;
   - kernels on hypercubes,

   satisfy Assumption (GF). Most previous literature [30, 33, 36, 19] have assumptions that only a proper subset of the above kernels satisfies. Therefore, we believe that we are operating under the minimal assumptions that exist in the field.

2. Independent features (IF): replace the feature vector with sub-Gaussian random vector with independent coordinates. A special case is the Gaussian Design assumption (GD) used in literature [44, 35, 16].

For further explanations regarding the assumptions, we refer the reader to Section A.

## 3 Main result

We first present an overview of the test error bounds across various properties, assumptions, and regimes. Our main results, summarized in Table 1, describe the learning curve in the over-parameterized regime, in terms of the bias $\mathcal{B}$ and variance $\mathcal{V}$ decomposition (see Equation (5)). Then, we will discuss the implications of our results in depth.

---

[3]The condition $a + 2 \neq 2r$ is purely technical and aims to simplify the results.
[4]Similar as above.

### 3.1 Overview

Table 1 summarizes many of our results in the over-parameterized regime under various combinations of the assumptions described in subsection 2.2. The bounds are expressed in terms of the sample size $n$ as $\mathcal{O}_n(\cdot)$ or $\tilde{\mathcal{O}}_n(\cdot)$ (ignoring logarithmic terms). Whenever we can also prove a matching lower bound, we replace $\mathcal{O}_n(\cdot)$ with $\Theta_n(\cdot)$. We write $(\cdot)_+ \overset{\text{def.}}{=} \max\{\cdot, 0\}$.

| Ridge | | strong | | weak | |
|---|---|---|---|---|---|
| Feature | | (IF) | (GF) | (IF) | (GF) |
| Poly (PE) | $\mathcal{B}$ | $\Theta\left(n^{-b\tilde{s}}\right)$ | $\mathcal{O}\left(n^{-b\tilde{s}}\right)$ | $\Theta\left(n^{-(1+a)\tilde{s}}\right)$ | $\begin{cases}\mathcal{O}\left(n^{-(1+a)\tilde{s}}\right), & s > 1 \\ \tilde{\mathcal{O}}\left(n^{-(\min\{2(r-a),2-a\})_+}\right), & s \le 1\end{cases}$ |
| | $\mathcal{V}$ | $\Theta\left(\sigma^2 n^{-1+\frac{b}{a+1}}\right)$ | $\mathcal{O}\left(\sigma^2 n^{-1+\frac{b}{a+1}}\right)$ | $\Theta\left(\sigma^2\right)$ | $\tilde{\mathcal{O}}\left(\sigma^2 n^{2a}\right)$ |
| Exp (EE) | $\mathcal{B}$ | $\Theta\left(e^{-b\tilde{s}n}\right)$ | $\mathcal{O}\left(e^{-b\tilde{s}n}\right)$ | $\mathcal{O}\left(e^{-a\tilde{s}n}\right), s > 1$ | $\mathcal{O}\left(e^{-a\tilde{s}n}\right), s > 1$ |
| | $\mathcal{V}$ | $\Theta\left(\sigma^2 n^{-1+\frac{b}{a}}\right)$ | $\mathcal{O}\left(\sigma^2 n^{-1+\frac{b}{a}}\right)$ | catastrophic overfitting | |

Table 1: *Learning curve in the over-parameterized regime ($p > n$):* $n$ is the sample size, $a, r > 0$ define the *eigen-decay rates* of the kernel and target function, $b > 0$ controls the decay rate of the ridge regularization parameter (Assumptions (PE) and (EE)), $\sigma^2 \overset{\text{def.}}{=} \mathbb{E}\left[\epsilon^2\right]$ is the *noise level*, and $s > 0$ is a technical parameter often determined by $a$ and $r$ (e.g. under Assumption (SC)). Here $\tilde{s} \overset{\text{def.}}{=} \min\{s, 2\}$.

Results in blue indicate either previously unstudied regimes or improvements in available rates in a studied regime. See Table 6 for more comparisons and Subsection 2.2 for details on various settings.

Before delving into the detailed discussion of our comprehensive results in Subsection 3.2, let us highlight some important observations from Table 1.

**Asymptotic bounds**   The upper bound illustrates the asymptotic relationship between the test error and sample size $n$ as well as the following constants: $a$ related to the eigen-decay, $b$ related to the ridge, $r$ related to the target function and $\sigma^2$ related to the label noise.

**Independent feature (IF) versus generic features (GF)**   The bounds in both cases coincide under strong ridge (see the left columns of Table 1); meanwhile, under weak ridge (see the right columns of Table 1), the bounds with generic features are looser than those with independent features. In Subsection 3.2, we will explain the necessity of this difference and hence showcase the previous limitations in the literature, which has widely adopted the Gaussian Design Assumption (GD) under the weak ridge/interpolation regime.

**Novel bound of bias under weak ridge**   A notably sophisticated bound (on the upper right corner of Table 1)

$$\mathcal{B} = \begin{cases}\mathcal{O}\left(n^{-(1+a)\tilde{s}}\right), & s > 1 \\ \tilde{\mathcal{O}}\left(n^{-(\min\{2(r-a),2-a\})_+}\right), & s \le 1\end{cases} \tag{6}$$

is novel to the best of our knowledge. Our improvement compared to previous literature [7] under various eigen-decay (in terms of $a$) and target coefficients' decay (in terms of $r$) is shown in Figure 1. By comparison, we can see that the decay of our novel bound in Equation 6 is faster than previous results. Also, we prove that the upper bound in the middle green region, where $s \in (1, 2)$, is sharp.

Experimental validations of the results in Table 1 are given in Section 6.

In the under-parameterized regime, the bias $\mathcal{B}$ and variance $\mathcal{V}$ terms can be bounded similarly as in over-parameterized regime. We postpone the details of these results to Section F.

### 3.2 Detailed discussion

In this subsection, we elaborate more on the details of our results shown in Table 1.

**Independent and generic features**   Table 1 indicates that the test error exhibits the same upper bound with either independent or generic features under strong ridge conditions in the over-parameterized regime. This similarity arises from the bounds in both cases being derived from the

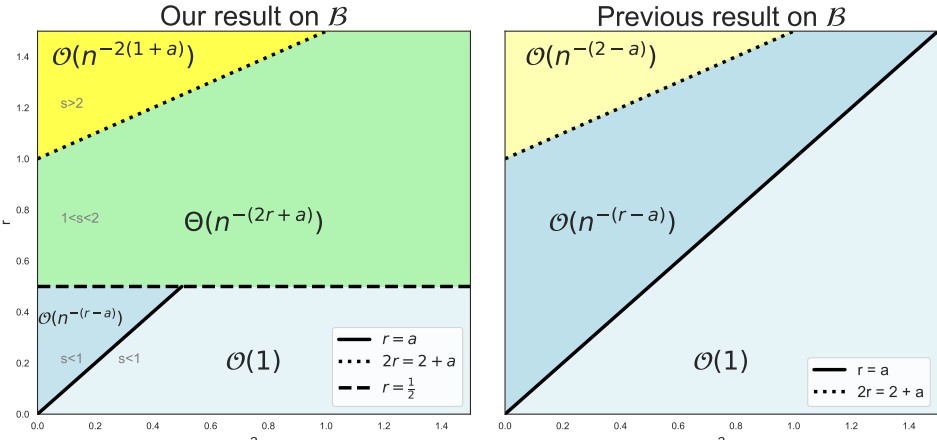

Figure 1: Phase diagram of the bound (Equation( 6)) of the bias term $\mathcal{B}$ under weak ridge and polynomial eigen-decay. $\lambda_k = \Theta_k \left( k^{-1-a} \right)$, $|\theta_k^*| = \Theta_k \left( k^{-r} \right)$, for some $a, r > 0$. Our result (Propositions D.5+D.6+E.1) is on the left, which improves over previous result from [7] (Proposition D.6) on the right. On the left plot, the range of the source coefficient $s = \frac{2r+a}{1+a}$ in Assumption (SC) is shown in gray font in each colored region.

same Master inequalities that we will introduce later (see Section 4). However, under weak ridge conditions, the empirical kernel spectrum displays qualitative differences, as reported in [7, 14]. From Figure 2, we can see that $\mathcal{V} = \mathcal{O}(1)$ with Laplacian kernel in the left plot and $\mathcal{V}$ diverges with the neural tangent kernel (with 1 hidden layer) in the right plot. Hence under weak ridge and polynomial eigen-decay, the case distinction of the bound

$$\mathcal{V} = \begin{cases} \Theta\left(\sigma^2\right), & \text{Assumption (IF) holds,} \\ \mathcal{O}\left(\sigma^2 n^{2a}\right), & \text{Assumption (GF) holds} \end{cases} \tag{7}$$

in Table 1 is necessary, as Assumption (GF) includes the cases of Gaussian Design (GD) (or more generally independent features (IF)) and Laplacian kernel which yields $\mathcal{V} = \Theta(1)$, the so-called tempered overfitting from [35]; as well as the case of neural tangent kernel (NTK) which yields $\mathcal{V} \xrightarrow{n \to \infty} \infty$, the so-called catastrophic overfitting. In particular, our proof shows that the Gaussian Equivalent Property (GEP) does not hold under weak ridge.

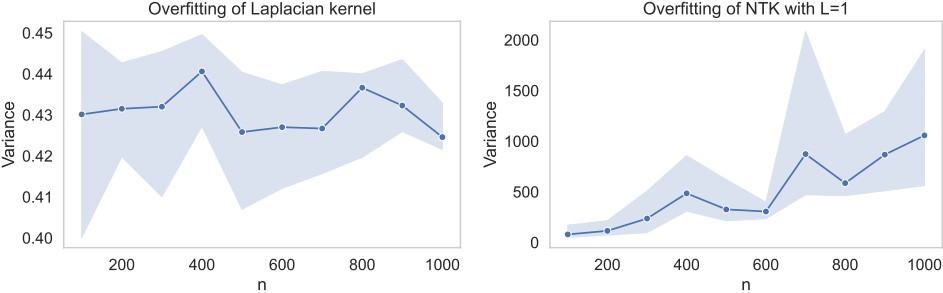

Figure 2: Variance $\mathcal{V}$ against sample size $n$ for the Laplacian kernel (left) and the neural tangent kernel with 1 hidden-layer (right) defined on the unit 2-disk, validating Equation (7) where the variance with generic features (GF) can be as good as with independent features (IF) ($\mathcal{V} = \Theta_n (1)$) or qualitatively different ($\mathcal{V} \xrightarrow{n \to \infty} \infty$). See Section 6 for more details.

We are now prepared to address the first question posed in Section 1:

*Q1: When and why does the Gaussian Equivalence Property (GEP) exist?*

**The Master inequalities provide the same non-asymptotic bounds for both cases under a strong ridge. However, GEP does not hold under weak ridge!** [5]

In particular, our work implies that previous works [10, 16, 34, 44, 35] under the Gaussian Design assumption (GD) can be applied only when the ridge is strong.

Upon finalizing the preparation for this paper, we became aware of concurrent research conducted by [19], which also concerns the Gaussian Equivalence Property (GEP) in the non-asymptotic setting. For more comparison between their assumptions and ours, we refer the reader to Section A.

**Importance of ridge**    To address the second question posed in Section 1, it is evident that in either the under-parameterized setting with any ridge (see Section F) or the over-parameterized regime with a strong ridge (see Table 1), the bounds for both $\mathcal{B}$ and $\mathcal{V}$ remain the same, irrespective of the features:

> *Q2: Under what conditions is the generalization error fully determined by the eigen-decay?*
> **Either (i) in the under-parameterized setting; or (ii) with a strong ridge in the over-parameterized regime.**

Several results [39, 33, 36] have suggested that the test error bound can be characterized by the covariance spectrum $\Sigma$, but they implicitly require the ridge $\lambda > 0$ to be larger than some threshold. This paper clearly demonstrates the necessity of the presence of a strong ridge in such analyses.

From Table 1, we can see that the eigen-decay also affects the test error qualitatively. As mentioned in [33], the bias term $\mathcal{B}$ decays polynomially (or, respectively, exponentially) when the eigen-decay is polynomial (or, respectively, exponential). However, we prove that $\mathcal{V}$ decays only polynomially at a rate at most $\mathcal{O}\left(\frac{1}{n}\right)$, regardless of the eigen-decay. Hence, in a noisy setting with polynomial eigen-decay, one can find an optimal ridge $\lambda = \Theta_n\left(n^{-\frac{1+a}{(1+a)\tilde{s}+1}}\right)$ to balance both terms $\mathcal{B}$ and $\mathcal{V}$ as in [16, 31]. In contrast, in noisy settings with exponential eigen-decay, $\mathcal{V}$ dominates the test error.

The bound of the variance term $\mathcal{V}$ with exponential eigen-decay under weak ridge is omitted in Table 1 due to the so-called catastrophic overfitting phenomenon observed in [35, 14].

**Improved upper bound**    During our research, we discovered that we can improve the upper bound of $\mathcal{B}$ (Equation 6) in the over-parameterized regime with polynomial decay, weak ridge, and Assumption (GF) by adapting the result from [31] and integrating it with the insights from [7] (see the upper right corner in Table 1 or 6, and Figure 1 for visualization).

**Theorem 3.1** (Improved upper bound). *Suppose Assumption (GF) holds. Assume the eigen-spectrum and the target coefficient both have polynomial decay, that is, $\lambda_k = \Theta_k\left(k^{-1-a}\right)$ and $|\theta_k^*| = \Theta_k\left(k^{-r}\right)$. Let $s = \frac{2r+a}{1+a}$ be the source coefficient defined in Definition (SC). Then the kernel ridgeless regression has the following learning rate for the bias term $\mathcal{B}$:*

$$\mathcal{B} = \begin{cases} \mathcal{O}_n\left(n^{-(r-a)_+}\right) & \text{for } s < 1; \\ \Theta_n\left(n^{-(2r+a)}\right) & \text{for } 1 \leq s \leq 2; \\ \mathcal{O}_n\left(n^{-2(1+a)}\right) & \text{for } s > 2. \end{cases}$$

*where $n$ is the sample size and $(\cdot)_+ \overset{\text{def.}}{=} \max(\cdot, 0)$.*

For a detailed explanation of our improvements, refer to our novel Proposition D.5 and the known Proposition D.6 from [7] in Section D.

**Lower bound of test error**    It is of theoretical interest to provide lower bounds as well. For independent features, we can prove that the upper bound is tight using the result from [46], generalizing the result from [16], which only showed upper bounds in Gaussian Design Assumption (GD). However, for generic features, we can only provide lower bounds in some but not all settings, using the result from [31]. We summarize our results in Table 2.

---

[5]Here we only concern the upper bound of the asymptotic learning rate for GEP. As far as we know, there is no other literature also concerning the lower bound. For readers interested in whether a matching lower bound for GEP exists, please refer to the paragraph "**Lower bound of the test error**" and Table 2 below.

| Ridge | | strong | | weak | |
|---|---|---|---|---|---|
| Feature | | (IF) | (GF) | (IF) | (GF) |
| Poly (PE) or Exp (EE) | $\mathcal{B}$ | ✓ | ✓ | ✓ | ✓ (when $1 \leq s \leq 2$) |
| | $\mathcal{V}$ | ✓ | unknown | ✓ | ✗ because of Figure 2 |

Table 2: The table shows whether the lower bound is matching the upper bound deduced in this paper.

See Section E for details on our proof of matching lower bounds. We note that there is some KRR literature, such as [31, 36, 19], that discusses matching upper and lower bounds of test error under more assumptions, which is beyond the scope of this paper. For a comparison of assumptions in different papers, see Section A.

Finally, we summarize the above discussion by answering the third question we raised in Section 1:

*Q3: Is there a unifying theory on the generalization performance under minimal assumptions?*
**Yes, this paper considers assumptions (IF) and (GF) which cover a wide range of kernel settings under any regularization and source conditions.**

## 4 Proof sketch

All the above results in Table 1 can be derived using the following proof strategy:

1. We prove a concentration result on the whitened features **z** under Assumptions (GF) or (IF) (see Section G).

2. Using the above result, we bound the condition number of the (truncated) kernel matrix (see Section G) as in [7, 14], which will be used to bound the test error in the next step.

3. Combining the bound of the condition number with the result from [6, 46], which we call the *Master inequalities* (see the paragraph below for details), we can compute the non-asymptotic test error bound for various settings (see Section D).

4. In the over-parameterized regime, we derive the asymptotic behavior of the learning curves by plugging in the eigen-decay and the choice of the target function and ridge in the above non-asymptotic bound (see Section D).

5. Using the results from [46], we are able to show that the asymptotic upper bound for independent features (IF) is tight (see section E).

6. For generic features (GF), we only provide tight bound results in limited settings (see section E). As shown in Figure 2 in Section 3, the generic feature Assumption (GF) includes a broad variety of features where a universal matching lower bound does not exist.

7. In the under-parameterized regime, the Master inequalities also give upper bounds for $\mathcal{B}$ and $\mathcal{V}$, which might not be tight if the ridge $\lambda$ is strong. However, we present another way to obtain tight bounds without the Master inequalities (see Section F for more details).

We summarize the proof techniques with a flowchart in Figure 4.

**Master Inequalities** We will now briefly introduce the key inequality used in our analysis (further details can be found in the Appendix). We will use the subscript $> k$ or $\leq k$ to refer to the submatrix of a matrix consisting of columns with index $> k$ or $\leq k$ (see definition C.3). The analysis relies on two crucial matrices: $\mathbf{A}_k = \mathbf{X}_{>k}\mathbf{X}_{>k}^{\top} + n\lambda\mathbf{I}_n \in \mathbb{R}^{n \times n}$, defined for any $k \in \mathbb{N}$, which is employed to partition the spectrum, and a whitened input matrix $\mathbf{Z} \stackrel{\text{def.}}{=} \mathbf{X}\mathbf{\Sigma}^{-1/2} \in \mathbb{R}^{n \times p}$. Additionally, we require the following definitions.

**Definition 4.1** (Concentration coefficients [7]). Define the quantities:

$$\rho \stackrel{\text{def.}}{=} \frac{n\|\mathbf{\Sigma}_{>k}\|_{\text{op}} + s_1(\mathbf{A}_k)}{s_n(\mathbf{A}_k)}; \quad \zeta \stackrel{\text{def.}}{=} \frac{s_1(\mathbf{Z}_{\leq k}^{\top}\mathbf{Z}_{\leq k})}{s_k(\mathbf{Z}_{\leq k}^{\top}\mathbf{Z}_{\leq k})}; \quad \xi \stackrel{\text{def.}}{=} \frac{s_1(\mathbf{Z}_{\leq k}^{\top}\mathbf{Z}_{\leq k})}{n},$$

where $s_i(\cdot)$ denotes the $i$-th largest singular value of the matrix.

**Definition 4.2** (Effective ranks [6]). Let $k \in \mathbb{N}$. Define two quantities:

$$r_k \stackrel{\text{def.}}{=} \frac{\text{Tr}[\boldsymbol{\Sigma}_{>k}]}{\|\boldsymbol{\Sigma}_{>k}\|_{\text{op}}} = \frac{\sum_{l=k+1}^{p} \lambda_l}{\lambda_{k+1}}, \qquad R_k \stackrel{\text{def.}}{=} \frac{\text{Tr}[\boldsymbol{\Sigma}_{>k}]^2}{\text{Tr}[\boldsymbol{\Sigma}_{>k}^2]} = \frac{\left(\sum_{l=k+1}^{p} \lambda_l\right)^2}{\sum_{l=k+1}^{p} \lambda_l^2}.$$

The Master inequalities provide upper bounds on the bias and the (scaled) variance term of the test error in the following form:

$$\mathcal{B} \leq \left(\frac{1 + \rho^2 \zeta^2 \xi^{-1} + \rho}{\delta}\right) \|\boldsymbol{\theta}_{>k}^*\|_{\boldsymbol{\Sigma}_{>k}}^2 + (\zeta^2 \xi^{-2} + \rho \zeta^2 \xi^{-1}) \frac{s_1(\mathbf{A}_k)^2}{n^2} \|\boldsymbol{\theta}_{\leq k}^*\|_{\boldsymbol{\Sigma}_{\leq k}^{-1}}^2;$$

$$\mathcal{V}/\sigma^2 \leq \rho^2 \left(\zeta^2 \xi^{-1} \frac{k}{n} + \frac{\text{Tr}[\mathbf{Z}_{>k} \boldsymbol{\Sigma}_{>k}^2 \mathbf{Z}_{>k}^\top]}{n \, \text{Tr}[\boldsymbol{\Sigma}_{>k}^2]} \frac{r_k(\boldsymbol{\Sigma})^2}{n R_k(\boldsymbol{\Sigma})}\right).$$

The term appearing in the above bound can be categorized into two distinct components: the "probably constant" part (highlighted in blue) and the "decay" part (highlighted in gray). The "probably constant" part consists of terms that, with high probability, are bounded both below and above by positive constants—these bounds represent a primary contribution of this paper. On the other hand, the "decay" part can be approximated using basic calculus, once a specific constant $k \in \mathbb{N}$, smaller than the sample size $n$, is selected. Together, these two parts allow us to derive the KRR learning rate for all combinations of eigen-spectra, features, and ridge parameters discussed in the preceding sections. Furthermore, [46] provides a corresponding lower bound under certain assumptions, demonstrating that the decay of the upper bound matches that of the lower bound. Establishing that Assumption (IF) satisfies these assumptions is another key contribution of this work. For the formal definitions of the terms in the above inequalities, we refer the reader to Propositions C.6 and C.7 in the appendix.

# 5 Related work

We briefly discuss related previous works and compare them with our results in the over-parameterized regime: (see Table 6 for more visual illustration)

1. [16] considered the upper bound of $\mathcal{B}$ and $\mathcal{V}$ in polynomial eigen-decay under any ridge and under the Gaussian Design Assumption. Our result proves *both* the matching upper and lower bound with the same decay rate under a weaker assumption (IF). This implies that we validate the Gaussian Equivalence Property for Sub-Gaussian Design.

2. [30, 31] proved tight upper bounds of $\mathcal{B}$ and $\mathcal{V}$ for Hölder continuous kernels under polynomial eigen-decay, strong ridge and the so-called Embedding condition (EC). [7] recovered the upper bounds under Assumption (GF).

3. For polynomial decay under weak ridge, [31] provided a tight upper bound of $\mathcal{B}$ when the source condition $s > 1$; while [7] provided a loose bound regardless of $s$. Hence we modify the proof in [31] under Assumption (GF) instead and combine it with [7] to obtain a novel upper bound on the bias.

4. [7] also provided an upper bound of $\mathcal{V}$ under polynomial decay, weak ridge and Assumption (GF).

5. [14] showed that $\mathcal{V}$ is bounded above and below by positive constants under polynomial eigen-decay, weak ridge and Assumption (IF), and it exhibits the so-called tempered overfitting from [35].

6. [33] provided tight upper bounds of $\mathcal{B}$ for both polynomial and exponential eigen-decay for kernels under the so-called Maximal Degree-of-Freedom (MaxDoF) Assumption. We recover their result under Assumption (GF) instead of Assumption (MaxDoF).

7. We apply the result from [46] and [31] to obtain a matching lower bound in some settings and strengthen our result from $\mathcal{O}_n(\cdot)$ to $\Theta_n(\cdot)$.

# 6 Experiments

Due to page constraints, this section focuses solely on experiments validating the Gaussian Equivalent Property (GEP). For detailed experiments on other contributions, refer to Section I.

We consider a simple example: let $\lambda_k = (\frac{2k-1}{2}\pi)^{-1-a}$ and $\psi_k(\cdot) = \sqrt{2}\sin\left(\frac{2k-1}{2}\pi\cdot\right)$ such that $\|\psi_k\|_{L^2_\mu} = 1$ for $\mu = \text{unif}[0,1]$; let $\theta^*_k = (\frac{2k-1}{2}\pi)^{-r}$. For $p = \infty$ and $a = 1$, the regression coincides with kernel ridge regression with kernel $k(x, x') = \min\{x, x'\}$ defined on the interval $[0,1]$ [48]. Similar experiments have been conducted on this kernel $k$ by [30, 33]. However, to simulate regression for independent features (IF), the feature rank $p$ must be finite. In the following experiment, we choose $p = 2000$, and the sample size $n$ ranges from 100 to 1000, with ridge parameter $\lambda = (\frac{2n-1}{2}\pi)^{-b}$ where $b \in [0, 1+a]$.

In Figure 3, our experiment demonstrates the GEP, as the learning curves with kernel features (Sine features) and independent features (Gaussian features $\mathbf{z} \sim \mathcal{N}(0, I_p)$ or Rademacher features $\mathbf{z} \sim (\text{unif}\{\pm 1\})^p$) coincide and match the theoretical decay.

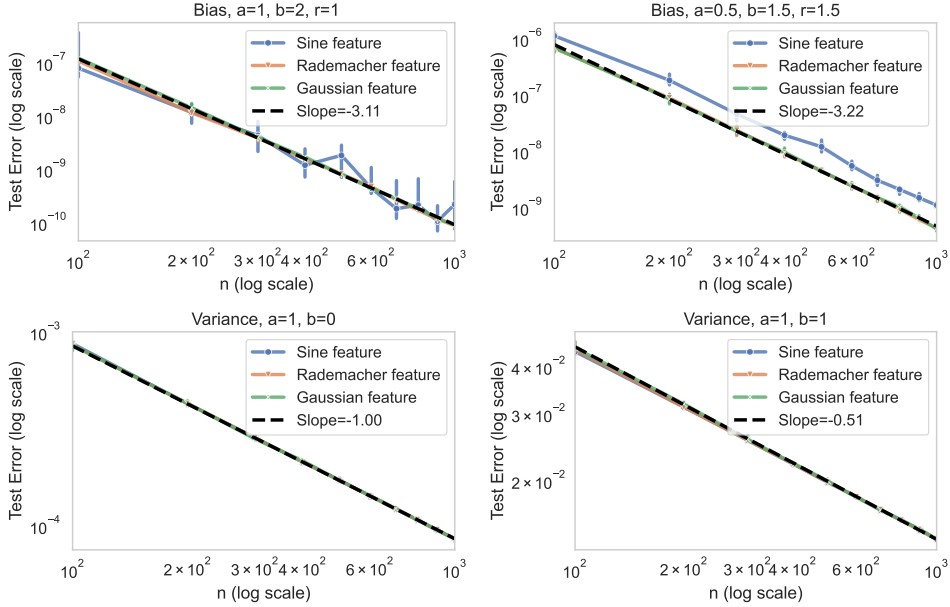

Figure 3: *Decay of the bias term $\mathcal{B}$ and the variance term $\mathcal{V}$ under different ridge decays and target coefficient decays.* All features demonstrate the same theoretical decay, validating the GEP for independent features.

## 7 Conclusion

In this paper, we present a unifying theory and a comprehensive analysis of the learning curve of kernel ridge regression under various settings. We elucidate the coincidence of learning curves between the Gaussian Design (GD) setting (or more generally, independent features (IF)) and the kernel setting (or more generally, generic features (GF)), and validate the Gaussian Equivalence Property under strong ridge. In addition to recovering previous results, we also improve test error bounds under specific circumstances, thus filling a gap in the existing literature.

**Future potential work** Our results also raise several theoretical questions. For instance: i) The upper bound in several regions of the phase diagram in Figure 1 is not known to be sharp. It would be of interest to either improve it or find matching lower bounds in those regions; ii) Why is there a qualitative difference in overfitting with the Laplacian kernel and the neural tangent kernel, as shown in Figure 2? Can one further distinguish different cases in the generic feature Assumption (GF) to explain such differences? Additionally, how tight would the bound $\mathcal{V} = \mathcal{O}\left(\sigma^2 n^{2a}\right)$ in Eq. (7) be?

Addressing these questions could provide deeper insights into the behavior of kernel ridge regression under various conditions and contribute to further advancing our understanding in machine learning.

## Acknowledgments

A. Kratsios acknowledges financial support from an NSERC Discovery Grant No. RGPIN-2023-04482 and No. DGECR-2023-00230. A. Kratsios also acknowledges that resources used in preparing this research were provided, in part, by the Province of Ontario, the Government of Canada through CIFAR, and companies sponsoring the Vector Institute[6].

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

# Appendix

The appendix will be organized in the following way:

- The detailed discussion of the assumptions used in this paper and the comparison to previous literature is presented in Section A;
- an extended discussion of the eigen-decay assumptions is presented in Section B;
- the main tool of this paper, the Master inequalities, is presented in Section C;
- the details of the proof on the asymptotic in over-parameterized regime in Table 1 is presented in Section D;
- the proof for a matching lower bound is presented in Section E;
- the details of the proof on the non-asymptotic bound in under-parameterized regime is presented in Section F;
- the concentration result on the whitened feature **z** and the conditioning of the kernel matrix are presented in Section G;
- related technical results from previous literature are collected in Section H;
- experimental results are shown in Section I;
- extra tables of summary can be found in Section J.

The summary of our proof can be found in the flowchart in Figure 4.

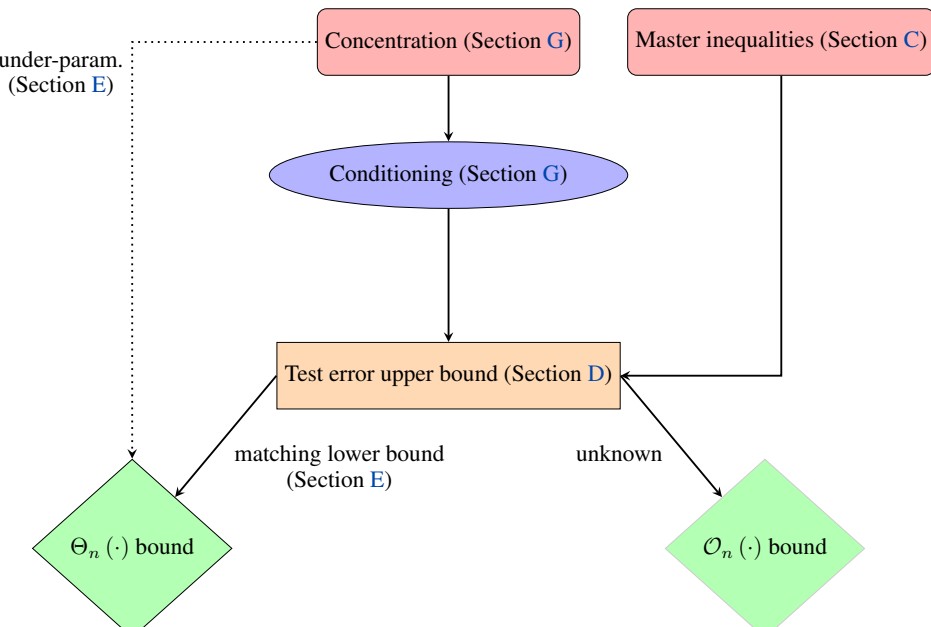

Figure 4: A flowchart about the proof techniques in this paper

**Big-O Notation**   We use the standard *Big-O* notations $\mathcal{O}.\left(\cdot\right), o.\left(\cdot\right), \Omega.\left(\cdot\right), \Theta.\left(\cdot\right)$ to represents the asymptotic behaviours of the bounds in this paper, where the subscript indicates that the constant in the bound is independent to the variables in the subscript. We use $\tilde{\cdot}$ to suppress logarithmic terms. With abuse of notation, we use the same notations for their probability versions, when the context is clear.

# A   Assumptions in details

The primary assumptions made in this paper concern the whitened feature vectors $\mathbf{z} \in \mathbb{R}^p$:

**Assumption (GF)** (Generic features [7])**.** *Let $p \in \mathbb{N} \cup \{\infty\}$. Assume the isotropic feature vector $\mathbf{z} \overset{\text{def.}}{=} \mathbf{\Sigma}^{-1/2}\mathbf{x} \in \mathbb{R}^p$ and the covariance matrix $\mathbf{\Sigma} \in \mathbb{R}^{p \times p}$ are such that:* [7]

$$\alpha_k \overset{\text{def.}}{=} \underset{\mathbf{z}}{\operatorname{ess\,inf}} \frac{\|\mathbf{z}_{>k}\|_{\mathbf{\Sigma}_{>k}}^2}{\operatorname{Tr}[\mathbf{\Sigma}_{>k}]} = \Theta_k(1),$$

$$\beta_k \overset{\text{def.}}{=} \underset{\mathbf{z}}{\operatorname{ess\,sup}} \max \left\{ \frac{\|\mathbf{z}_{\leq k}\|_2^2}{k}, \frac{\|\mathbf{z}_{>k}\|_{\mathbf{\Sigma}_{>k}}^2}{\operatorname{Tr}[\mathbf{\Sigma}_{>k}]}, \frac{\|\mathbf{z}_{>k}\|_{\mathbf{\Sigma}_{>k}^2}^2}{\operatorname{Tr}[\mathbf{\Sigma}_{>k}^2]} \right\} = \Theta_k(1),$$

*Remark* A.1 ($k = 0$). For $k = 0$, we take the convention that $\beta_0 \overset{\text{def.}}{=} \operatorname{ess\,sup}_{\mathbf{z}} \max \left\{ \frac{\|\mathbf{z}\|_{\mathbf{\Sigma}}^2}{\operatorname{Tr}[\mathbf{\Sigma}]}, \frac{\|\mathbf{z}\|_{\mathbf{\Sigma}^2}^2}{\operatorname{Tr}[\mathbf{\Sigma}^2]} \right\}$.

*Remark* A.2 (Intuition). The intuition is that the three fractions above have expected values of 1: $\mathbb{E}_{\mathbf{z}}\left[\frac{\|\mathbf{z}_{\leq k}\|_2^2}{k}\right] = \mathbb{E}_{\mathbf{z}}\left[\frac{\|\mathbf{z}_{>k}\|_{\mathbf{\Sigma}_{>k}}^2}{\operatorname{Tr}[\mathbf{\Sigma}_{>k}]}\right] = \mathbb{E}_{\mathbf{z}}\left[\frac{\|\mathbf{z}_{>k}\|_{\mathbf{\Sigma}_{>k}^2}^2}{\operatorname{Tr}[\mathbf{\Sigma}_{>k}^2]}\right] = 1$. Hence, Assumption (GF) imposes strong concentration on $\mathbf{z}$ at each truncation point $k \in \mathbb{N}$.

*Remark* A.3 (Weaken Assumption (GF)). One can further replace the essential supremum and infimum by a high probability guarantee. The argument on the test error bounds would follow analogously with a weaker probability. For simplicity reasons, we only consider the assumption in the above formulation.

Assumption (GF) encompasses both cases of dependent and independent features $z_k$, which can have qualitatively different effects on the test error.

To compare the realistic kernel setting with the Gaussian Design setting (GD) used in previous literature [10, 34, 16, 44, 35], which replaces the feature vectors with Gaussian random vectors (with independent entries), we consider a weaker version of the Gaussian Design assumption commonly required in KRR literature [6, 13, 5, 36, 19]:

**Assumption (IF)** (Independent sub-Gaussian features)**.** *Suppose $p \in \mathbb{N}$. Assume that the features $z_k$'s in the isotropic feature vector $\mathbf{z}$ are independent to each other and there exists a constant $G > 0$ such that the sub-Gaussian norm $\|z_k\|_{\psi_2}$ of any feature $z_k$ is bounded by $G$. In over-parameterized regime, we also assume that there exists some constant $\eta > 1$ independent to $n$ such that $p > \eta n$.*

*Remark* A.4 (Infinite rank). With Assumption (GF), the feature dimension $p$ can be taken as $\infty$; with Assumption (IF), however, if no further boundedness assumption is imposed, the norm $\|\mathbf{x}\|_2 = \sqrt{\sum_{k=1}^p \lambda_k z_k^2}$ can be undefined if $p = \infty$. However, if the sub-Gaussian variables $z_k$'s are all bounded in the sense of Assumption (GF), then $p$ can be chosen to be infinity as well.

*Remark* A.5 (Special case). If the features $z_k$ are additionally bounded, then Assumption (IF) becomes a special case of Assumption (GF). The former assumes independence of the features, while the latter does not.

Both assumptions (GF) and (IF) are much weaker than those in most KRR literature in the sense that: i) there is no explicit assumption on the input space $\mathcal{X}$; ii) the distribution of the feature vector need not even be continuous! It is surprising that even with such minimal assumptions, one can derive tight bounds on the test error.

For a more detailed comparison between the assumptions in this paper and those in previous literature, see Section A.

We next compare the assumptions (IF), (GF) with the assumptions in previous literature.

## A.1   Among assumptions with independent features

The Gaussian Design Assumption, often employed in various literature such as [10, 16, 34, 20, 35, 44], can be considered a special case of Assumption (IF):

---

[7] We write $\|\mathbf{v}\|_{\mathbf{M}} \overset{\text{def.}}{=} \sqrt{\mathbf{v}^\top \mathbf{M} \mathbf{v}}$ for PDS matrix $\mathbf{M}$ and any vector $\mathbf{v}$. See Definition C.1 Section C for more details.

**Assumption (GD)** (Gaussian Design). *The feature vector $\mathbf{z} \sim \mathcal{N}(0, \mathbf{I}_p)$ is distributed as isotropic Gaussian.*

Note that Assumption (IF) is much weaker than Assumption (GD) in the sense that: i) the distribution of the feature vector $\mathbf{z}$ is no longer rotational invariant or enjoys anti-concentration property; [8] ii) the distribution of each feature $z_k$ needs not be continuous; iii) the features $z_k$'s needs not be identical.

Meanwhile, Assumption (IF) appear commonly in KRR literature, for example [6, 46, 23, 5, 13, 14].

### A.2 Among assumptions with dependent features

Examples that satisfy Assumption (GF), which is first introduced by [7], includes:

1. dot-product kernels on hyperspheres;
2. kernels with bounded eigenfunctions;
3. radial base function (RBF) and shift-invariant kernels;
4. kernels on hypercubes.

In particular, the features from Assumption (IF) also satisfies the weakened version of Assumption (GF). See Remark A.3 for details.

Before comparing our assumptions with previous literature, we first define the $\infty$-norm in the ridge regression setting analog to its kernel counterpart:

**Definition A.6** ($\infty$-norm). Fix a distribution $\mu$ on $\mathbb{R}^p$ and let $\mathbf{v} \in \mathbb{R}^p$. Define

$$\|\mathbf{v}\|_\infty \overset{\text{def.}}{=} \underset{x \sim \mu}{\text{ess sup}} |\mathbf{x}^\top \mathbf{v}| = \sup\{a \in \mathbb{R} | a \leq |\mathbf{x}^\top \mathbf{v}| \text{ a.s. for all } \mathbf{x} \sim \mu\}$$

*Remark* A.7. Note that $\|\cdot\|_\infty$ is indeed a semi-norm and for all $\mathbf{v} \in \mathbb{R}^p$, we have

$$\|\mathbf{v}\|_{\boldsymbol{\Sigma}} \leq \|\mathbf{v}\|_\infty .$$

Choose $k = 0$ and Assumption (GF) and Remark A.1 imply:

$$\|\mathbf{x}\|_\infty^2 = \underset{\mathbf{x}}{\text{ess sup}} \|\mathbf{x}\|_2^2 = \underset{\mathbf{z}}{\text{ess sup}} \|\mathbf{z}\|_{\boldsymbol{\Sigma}}^2 \leq \beta_0 \overset{\text{def.}}{=} \underset{\mathbf{z}}{\text{ess sup}} \max \left\{ \frac{\|\mathbf{z}\|_{\boldsymbol{\Sigma}}^2}{\text{Tr}[\boldsymbol{\Sigma}]}, \frac{\|\mathbf{z}\|_{\boldsymbol{\Sigma}^2}^2}{\text{Tr}[\boldsymbol{\Sigma}^2]} \right\} < \infty,$$

as we assume that $\text{Tr}[\boldsymbol{\Sigma}] < \infty$.

**Embedding condition**  [29, 50, 30, 31] used the Embedding Condition to show tight bounds of learning curve with polynomial eigen-decay:

**Assumption (EC)** (Embedding Condition). *Given a random vector $\mathbf{x} \in \mathbb{R}^p$ with covariance $\boldsymbol{\Sigma} = \mathbb{E}[\mathbf{x}\mathbf{x}^\top]$, let $\mathbf{z} = \boldsymbol{\Sigma}^{-1/2}\mathbf{x}$ be the isotropic random vector. Define the embedding coefficient:*

$$\eta_0 = \inf_\eta \{\underset{\mathbf{x}}{\text{ess sup}} \|\mathbf{x}\|_{\boldsymbol{\Sigma}^{\eta-1}}^2 < \infty\} = \inf_\eta \{\underset{\mathbf{z}}{\text{ess sup}} \|\mathbf{z}\|_{\boldsymbol{\Sigma}^\eta}^2 < \infty\}.$$

*Suppose the eigen-decay is polynomial: $\lambda_k = \Theta_k \left(k^{-1-a}\right)$ for some $a > 0$. Assume the embedding coefficient $\eta_0 = \frac{1}{1+a}$.*

*Remark* A.8 (General range of $\eta_0$). Rewrite the expression:

$$\|\mathbf{x}\|_{\boldsymbol{\Sigma}^{\eta-1}}^2 = \mathbf{x}^\top \boldsymbol{\Sigma}^{\eta-1} \mathbf{x} = \sum_{k=1}^p \lambda_k^{\eta-1} x_k^2 = \sum_{k=1}^p \lambda_k^\eta z_k^2 = \Theta_k \left(\sum_{k=1}^p k^{-\eta(1+a)} z_k^2\right)$$

where we write $\mathbf{x} = (x_k)_{k=1}^p \in \mathbb{R}^p$ and $\mathbf{z} = (z_k)_{k=1}^p \in \mathbb{R}^p$. In the kernel case, it is obvious that $\eta_0 \leq 1$: for all inputs $x \in \mathcal{X}$ with embedding $\mathbf{x} = K(x, \cdot)$ in the feature space $\mathcal{H}$, $\sup_x k(x, x) =$

---

[8]Rotational invariant, and respectively anti-concentration property, are crucial in some literature using Gaussian Design Assumption, for example [44], and respectively [14].

$\|\mathbf{x}\|_2^2 < \infty$. Since the eigenvalues have polynomial decay, we have $\eta_0 \geq \frac{1}{1+a}$, or otherwise there were some $\eta \in (\eta_0, \frac{1}{1+a})$ such that:

$$\mathbb{E}_{\mathbf{x}}\left[\|\mathbf{x}\|_{\mathbf{\Sigma}^{\eta-1}}^2\right] = \Omega\left(\mathbb{E}_{\mathbf{z}}\left[\sum_{k=1}^{p} k^{-\eta(1+a)}z_k^2\right]\right) = \Omega\left(\sum_{k=1}^{p} k^{-\eta(1+a)}\right) = \infty,$$

contradicting Assumption (EC). Thus one have $\eta_0 \in [\frac{1}{1+a}, 1]$ in general.

Hence Assumption (GF) is a much weaker assumption than Assumption (EC) in the sense that the former one only requires $\eta_0 \leq 1$, which holds in general; while the latter one requires $\eta_0 = \frac{1}{1+a}$, which is the smallest possible size of $\eta_0$. Hence, many examples mentioned in [50], that satisfy Assumption (EC) also satisfy Assumption (GF). For instance, the examples include:

1. dot-product kernels on hyperspheres;

2. kernels with bounded eigenfunctions;

3. shift-invariant periodic kernels.

Nevertheless, [29, 50, 30, 31] also require Hölder continuity of the kernel $K$ while Assumption (GF) requires no continuity assumption.

**Maximal degree of freedom**    We rewrite and simplify the assumption used in [33] with our notation:

**Assumption (MaxDoF)** (Maximal Degree-of-Freedom). *For any $\lambda > 0$ and $s \geq 1$, define:*

$$F^s(\lambda) \stackrel{\text{def.}}{=} \operatorname*{ess\,sup}_{\mathbf{x}} \left\|(\mathbf{\Sigma} + \lambda\mathbf{I}_p)^{-s/2}\mathbf{x}\right\|_{\mathbf{\Sigma}^{s-1}}^2 .$$

*Assume: there exists $n \in \mathbb{N}$ such that*

$$n \geq 5F^s(\lambda_n)\max\{1, \log(14F^s(\lambda_n))/\delta\}$$

*for some constant $\delta \in (0, 1)$.*

Usually, the constant $s$ is chosen as the source coefficient in Assumption (SC). Again, Assumption (MaxDoF) requires some extra boundedness condition on the norm of $\mathbf{x}$. Examples that satisfy Assumption (MaxDoF) include:

1. dot-product kernels on hyperspheres;

2. kernels with bounded eigenfunctions;

3. periodic kernels on hypercubes,

which also satify Assumption (GF).

**Low/High-degree feature concentration**    [36] proved the convergence of the test error to its "deterministic equivalent" using a list of assumptions for each fixed sample size $n$. To avoid introducing extra notations, we list their assumption here in a semi-rigorous way:

**Assumption (LH)** (Low/High-degree feature concentration). *Fix $n \in \mathbb{N}$ and $\lambda \geq 0$. Suppose there exists an integer $k \leq p$ such that:*

1. $r_k^\lambda \stackrel{\text{def.}}{=} \frac{\lambda + \text{Tr}[\mathbf{\Sigma}_{>k}]}{\lambda_{k+1}} \geq 2n$;

2. *the low-degree feature vector $\mathbf{x}_{\leq k} \in \mathbb{R}^k$ is highly concentrated;*

3. *with high probability, the high-degree feature vector $\mathbf{x}_{>k} \in \mathbb{R}^{p-k}$ satisfies:*

$$\left\|\mathbf{X}_{>k}\mathbf{X}_{>k}^\top - \text{Tr}[\mathbf{\Sigma}_{>k}]\mathbf{I}_n\right\|_{op} \leq C\sqrt{\frac{n}{r_k^\lambda}} \cdot (\lambda + \text{Tr}[\mathbf{\Sigma}_{>k}])$$

*where $C > 1$ is a constant depending on $n, k$.*

Point 3 in Assumption (LH) is closely related to Assumption (GF) since:

$$\frac{1}{n}\sum_{i=1}^{n}\|\mathbf{z}_{>k}\|_{\boldsymbol{\Sigma}_{>k}}^{2} - \mathrm{Tr}[\boldsymbol{\Sigma}_{>k}] = \frac{1}{n}\mathrm{Tr}[\mathbf{X}_{>k}\mathbf{X}_{>k}^{\top}] - \mathrm{Tr}[\boldsymbol{\Sigma}_{>k}]\mathbf{I}_{n}] \leq \left\|\mathrm{Tr}[\mathbf{X}_{>k}\mathbf{X}_{>k}^{\top}] - \mathrm{Tr}[\boldsymbol{\Sigma}_{>k}]\mathbf{I}_{n}]\right\|_{\mathrm{op}},$$

if $\mathrm{ess\,sup}_{\mathbf{x}}\frac{\|\mathbf{z}_{>k}\|_{\boldsymbol{\Sigma}_{>k}}^{2}}{\mathrm{Tr}[\boldsymbol{\Sigma}_{>k}]} = \Theta_{k}(1)$, then $\frac{1}{n}\sum_{i=1}^{n}\|\mathbf{z}_{>k}\|_{\boldsymbol{\Sigma}_{>k}}^{2} - \mathrm{Tr}[\boldsymbol{\Sigma}_{>k}] = \Omega_{k}(1)$.

According to [36], examples that satisfy Assumption (LH) include:

1. kernel satisfying Assumption (IF);
2. dot-product kernel on hyperspheres.

**Moment-equivalent assumption** [19] use a geometric argument with Dvoretzky-Milman Theorem to bound the test error from above under the assumptions:

**Assumption (ME)** ($L^{2+\epsilon}$-$L^2$ Moment-Equivalent Assumption). *Fix a sample $(\mathbf{x}_i)_{i=1}^{n}$. Suppose there exists some $\epsilon, \kappa > 0$ such that:*

1. *for any $k \in \mathbb{N}$ and any function $f \in \mathcal{H}_{>k}$ in the high-degree feature space, we have* $\|f\|_{L^{2+\epsilon}} \leq \kappa\|f\|_{L^2}$.

    (a) *if $\epsilon > 2$, then no extra assumption is required;*

    (b) *if $\epsilon \in (0,2]$, then $\kappa n^{\frac{2-\epsilon}{2\epsilon+\epsilon^2}}\log n\left(\frac{\sqrt{n\,\mathrm{Tr}[\boldsymbol{\Sigma}_{>k}^2]}}{\mathrm{Tr}[\boldsymbol{\Sigma}_{>k}]}\right)$ is small.*

2. *with high probability, $\max_{i=1,\dots,n}\frac{\|(\mathbf{z}_i)_{>k}\|_{\boldsymbol{\Sigma}_{>k}}^{2}}{\mathrm{Tr}[\boldsymbol{\Sigma}_{>k}]} \approx 1$;*

3. *with high probability, $\max_{i=1,\dots,n}\frac{\|(\mathbf{z}_i)_{>k}\|_{\boldsymbol{\Sigma}_{>k}^2}^{2}}{\mathrm{Tr}[\boldsymbol{\Sigma}_{>k}^2]} \approx 1$;*

4. *with high probability, $\max_{i=1,\dots,n}\frac{\|(\mathbf{z}_i)_{\leq k}\|_{2}^{2}}{k} = \mathcal{O}_{k}(1)$.*

While points 2-4 are implied by the weakened version of Assumption (GF), point 1 is a strong geometric assumption on the RKHS $\mathcal{H}$. According to [19], the addition of point 1 can drop the logarithmic terms in the upper bound of test error under weak ridge, and thus strengthening some results in Table 1 from $\tilde{\Theta}(\cdot)$ to $\Theta(\cdot)$. Examples satisfying Assumption (ME) includes:

1. rotational invariant kernel.

## A.3 Between independent and dependent features: random features

A random feature setting can be considered as a midpoint between independent and dependent features: Let $\mathbf{W} \in \mathbb{R}^{p \times d}$ be a random matrix, typically with i.i.d. random entries, and $\mathbf{z} \in \mathbb{R}^d$ be the input vector, usually comprising independent entries. We define the random feature $\mathbf{x} \in \mathbb{R}^p$ as a one-hidden-layer neural network: $\mathbf{x} \stackrel{\text{def.}}{=} \varphi(\mathbf{W}\mathbf{z})$, where $\varphi : \mathbb{R} \to \mathbb{R}$ represents an activation function acting entry-wise on $\mathbf{W}\mathbf{z}$. Thus, the coordinates of the pre-activation are independent, while those of the post-activation are not.

However, it is worth noting the Gaussian Equivalent Property highlighted in [20], which asserts that in wide neural networks, the distribution of the feature vector $\mathbf{x}$ is approximately Gaussian. Consequently, it reduces to the case of independent features previously discussed.

## A.4 Interpolation space

Many KRR literature [29, 30, 31, 33] introduced the notation of interpolation space and the source condition on the $L^2$-integrable target function $f^*$. This can be easily translated to the notion of generalized vector norm.

**Definition A.9** (Interpolation space). Let $s \geq 0$ be a real number. Define the interpolation space

$$\mathcal{H}^s \stackrel{\text{def.}}{=} \{\boldsymbol{\theta} \in \mathbb{R}^p : \|\boldsymbol{\theta}\|_{\boldsymbol{\Sigma}^{1-s}} < \infty\}.$$

Evidently, if $s_1 \leq s_2$, then $\|\boldsymbol{\theta}\|_{\boldsymbol{\Sigma}^{1-s_2}} \leq \|\boldsymbol{\theta}\|_{\boldsymbol{\Sigma}^{1-s_1}}$ and hence $\mathcal{H}_{s_1} \supset \mathcal{H}_{s_2}$. The intuition is that, in the KRR setting, the space $\mathcal{H}^s$ interpolates between the $L^2$-space $L^2_\mu(\mathcal{X}) \supset \mathcal{H}^0$ and the RKHS $\mathcal{H} = \mathcal{H}^1$.

## A.5 Source condition

Given the asymptotic of the eigen-decay $\lambda_k$ and the target coefficient $\boldsymbol{\theta}^*$, we can find the largest possible source coefficient $s$. For instance, if $\lambda_k = \Theta_k\left(k^{-1-a}\right)$ for some $a > 0$ and $|\theta_k^*| = \Theta_k\left(k^{-r}\right)$, then we can choose $s$ to be $\frac{2r+a}{1+a}$:

$$\boldsymbol{\theta}^* \in \mathcal{H}^t \Leftrightarrow \|\boldsymbol{\theta}^*\|^2_{\boldsymbol{\Sigma}^{1-t}} < \infty$$

$$\Leftrightarrow \sum_{k=1}^p (\theta^*)^2 \lambda_k^{1-t} < \infty$$

$$\Leftrightarrow \sum_{k=1}^p k^{-2r} k^{(1-t)(-1-a)} < \infty$$

$$\Leftrightarrow \sum_{k=1}^p k^{-1-a+t+ta-2r} < \infty$$

$$\Leftrightarrow -1 - a + t + ta - 2r < -1$$

$$\Leftrightarrow t < \frac{2r+a}{1+a}.$$

Suppose $\lambda_k = \Theta_k\left(e^{-ak}\right)$, $|\theta_k^*| = \Theta_k\left(e^{-rk}\right)$ for $a, r > 0$. Then we can choose $s$ to be $\frac{2r}{a} + 1$:

$$\boldsymbol{\theta}^* \in \mathcal{H}^t \Leftrightarrow \|\boldsymbol{\theta}^*\|^2_{\boldsymbol{\Sigma}^{1-t}} < \infty$$

$$\Leftrightarrow \sum_{k=1}^p (\theta^*)^2 \lambda_k^{1-t} < \infty$$

$$\Leftrightarrow \sum_{k=1}^p e^{-2rk} (e^{-ak})^{1-t} < \infty$$

$$\Leftrightarrow \sum_{k=1}^p e^{-k(2r+a-at)} < \infty$$

$$\Leftrightarrow 2r + a - at > 0$$

$$\Leftrightarrow t < \frac{2r}{a} + 1.$$

## A.6 Kernel ridge regression

KRR can be regarded as ridge regression on the feature space, where the positive definite symmetric (PDS) kernel $K$ sends each input $x \in \mathcal{X}$ to a vector/function $K(x, \cdot)$ in the corresponding reproducing kernel Hilbert space (RKHS) $\mathcal{H}$, and the kernel regressor is simply a linear regressor in the RKHS $\mathcal{H}$. Here we briefly explain how to translate the notations above in the kernel ridge regression (KRR) setting.

With abuse of notation, let $\mu$ be a (data-generating) distribution on an input space $\mathcal{X}$. Let $K : \mathcal{X} \times \mathcal{X} \to \mathbb{R}$ be a positive definite symmetric kernel with corresponding reproducing kernel Hilbert space (RKHS) $\mathcal{H}$. By Mercer decomposition, we have for all $x, x' \in \mathrm{supp}(\mu) \subset \mathcal{X}$:

$$K(x, x') = \sum_{k=1}^p \lambda_k \psi_k(x) \psi_k(x'),$$

where $\lambda_k$'s are the eigenvalues of $K$ indexed in decreasing order and $\psi_k : \mathcal{X} \to \mathbb{R}$'s are eigenfunctions of $K$ forming an orthonormal basis on $L^2_\mu(\mathcal{X})$: $\mathbb{E}_{x \sim \mu}\left[\psi_k(x)\psi_l(x)\right] = \delta_{kl}$ for all $k, l \in \mathbb{N}$. Hence for

each $x \in \operatorname{supp}(\mu) \subset \mathcal{X}$, $K(x, \cdot)$ is a feature vector in the RKHS $\mathcal{H}$, which has an orthonormal basis $\left\{\phi_k = \frac{1}{\sqrt{\lambda_k}} \psi_k\right\}_{k=1}^{p}$.

Let $y = f^*(x) + \epsilon$ be the labels where $f^* \in L^2_\mu(\mathcal{X})$ and $\epsilon$ is a centered random variable with finite variance independent to $x$. Let $(x_i, y_i)_{i=1}^{n}$ be a set of i.i.d. drawn input-output pairs. The kernel ridge regression is in the form:

$$\hat{f} \stackrel{\text{def.}}{=} \min_{f \in \mathcal{H}} \frac{1}{n} \sum_{i=1}^{n} (f(x_i) - y_i)^2 + \lambda \|f\|_{\mathcal{H}}^2.$$

which minimum admits the form:

$$\hat{f}(x) = \mathbf{K}_x^\top (\mathbf{K} + n\lambda \mathbf{I}_n)^{-1} \mathbf{y}, \ \forall x \in \mathcal{X},$$

where $K_x = (K(x, x_i))_{i=1}^{n} \in \mathbb{R}^n$, $\mathbf{K} = [K(x_i, x_j)]_{i,j=1}^{n} \in \mathbb{R}^{n \times n}$ and $\mathbf{y} = (y_i)_{i=1}^{n}$. Note that if we write $\boldsymbol{\psi}_x = (\psi_k(x))_{k=1}^{p} \in \mathbb{R}^p$, $\boldsymbol{\Psi} = [\psi_k(x_i)]_{ik} \in \mathbb{R}^{n \times p}$ and $\boldsymbol{\Lambda} = \operatorname{diag}\{\lambda_k\}_{k=1}^{p} \in \mathbb{R}^{p \times p}$, by Mercer decomposition, we have $K_x = \boldsymbol{\Psi} \boldsymbol{\Lambda} \boldsymbol{\psi}_x$, $\mathbf{K} = \boldsymbol{\Psi} \boldsymbol{\Lambda} \boldsymbol{\Psi}^\top$. We are ready to translate our linear ridge regression to the KRR setting. See Table 5 for details.

# B    Smoothness and spectral decay rate

In this section, we will explain the reason to assume the eigen-decay to be either polynomial or exponential in the paper.

Concretely, the "polynomial decay regime" arises as the Neural Tangent Kernel (NTK) limit of Multi-Layer Perceptrons (MLPs) with an activation function of the form $\sigma_s(t) \overset{\text{def.}}{=} \max{0, t^s}$, where $s \geq 1$, as studied in [4]. In cases where $s$ is an integer, $\sigma_s$ represents the $(s-1)$-th iterated anti-derivative of the ReLU function, making it exactly $s$ times weakly differentiable.

Suppose $\mathcal{X}$ is the $d$-dimensional sphere $\mathbb{S}^d \overset{\text{def.}}{=} \{u \in \mathbb{R}^d : |u| = 1\}$ (with its usual Riemannian metric). In this scenario, every neural network function $f : \mathbb{S}^d \to \mathbb{R}$ belongs to the Sobolev space $H^s(\mathbb{S}^d)$ (see Proposition B.7 in Appendix B.2), characterized by having $s$ continuous and integrable (weak) partial derivatives on $\mathbb{S}^d$. Additionally, the NTK limit manifests as a "radially symmetric kernel", specifically of the form $k(x, y) = \kappa(x^\top y)$ for some suitable function $\kappa : [-1, 1] \to \mathbb{R}$.

It is worth noting that the choice of the smoothness level, and hence the decay rate, naturally emerges when approximately solving Partial Differential Equations (PDEs), as demonstrated in works such as [26, 15]. In these scenarios, it is desired that the kernel regressor exhibits the correct level of smoothness matching that of the theoretical solution to the PDE, with its higher-order derivatives also converging. It is emphasized that the solution to several PDEs only possesses finitely many derivatives, as seen in (viscosity) solutions to PDEs arising in stochastic control [37, 38].

In the remainder of this section, we justify the use of Assumptions (PE) and (EE) by establishing a connection between the spectral eigen-decay and the smoothness of the Reproducing Kernel Hilbert Space (RKHS), which is independent of the rest of the paper. Readers primarily interested in the proofs presented in Table 1 may opt to skip this section and proceed to Section C during their initial reading.

## B.1    Notations

Fix a positive integer $d \in \mathbb{N}$ and let $\mathcal{X}$ be a non-empty subset of $\mathbb{R}^d$. We equip $\mathcal{X}$ with a regular Borel probability measure $\mu$, which we specify shortly. We denote the space of (equivalence classes of) $\mu$-square-integrable on $\mathbb{R}^d$ by $L^2_\mu(\mathbb{R}^d)$, equipped with its usual $L^2_\mu$-norm $\|\cdot\|_{L^2_\mu}$.

Consider a kernel $\kappa : \mathcal{X} \times \mathcal{X} \to \mathbb{R}$ with its associated to the RKHS $(\mathcal{H}, \langle\cdot, \cdot\rangle_\kappa)$ (abbreviated by $\mathcal{H}$). We denote the induced norm on $\mathcal{H}$ by $\|\cdot\|_\kappa$. By the Mercer decomposition theorem, see [40, Theorem 2.30], there is some $M \in \mathbb{N} \cup \{\infty\}$ such that we may write

$$\kappa(x, y) = \sum_{i=0}^{M} \lambda_i \, \psi_i(x) \psi_i(y) \tag{8}$$

where $(\psi_i)_{i=0}^{M}$ are normalized eigenfunctions of the linear operator $T_\kappa(f)(x) = \int_{x \in \mathcal{X}} \kappa(x, u) f(u) \, du$ on $L^2_\mu(\mathcal{X})$ and each eigenvalue $(\lambda_i)_{i=0}^{\infty}$ is non-negative; which, form an orthonormal basis of $\mathcal{H}$.

We motivate our analysis, by the following connection between the spectral decay of dot-product kernels on spheres and the smoothness of functions in their RKHS.

**Example B.1** (Radial Kernels and Sobolev Spaces on Spheres). *In the setting of Examples B.3 and (B.5), suppose that there is a function $k : [-1, 1] \to \mathbb{R}$ such that*

$$\kappa(x, y) \overset{\text{def.}}{=} k(\langle x, y \rangle) \tag{9}$$

*for each $x, y \in \mathbb{S}^d$. As shown in [41], $\kappa$ is a kernel if and only if there is a summable sequence of non-negative real numbers $(\alpha_i)_{i=0}^{\infty}$ satisfying $k(t) = \sum_{i=0}^{\infty} \alpha_i t^i$ for each $t \in [-1, 1]$. Kernels of the form (9) are called radial. The RKHS $\mathcal{H}$ associated to $\kappa$ consists of all functions $f : \mathbb{S}^d \to \mathbb{R}$ with*

$$f = \sum_{j=0}^{\infty} \sqrt{\mu_j} \sum_{i=1}^{I_j} \alpha_{j,i} \, \bar{\psi}_{j,i} \tag{10}$$

*for a square-summable real sequence $(\alpha_{j,i})_{j,i=1}^{\infty, J_i}$; i.e. $\sum_{i=0}^{\infty} \sum_{j=1}^{J_i} \alpha_{j,i}^2 < \infty$ (where $(\alpha_{i,j})_{i,j=1}^{\infty, J_i}$ depends only on the radial function $k$, and thus only on the radial kernel $\kappa$; and $(\bar{\psi}_{j,i})_{j,i=1}^{\infty, J_i}$ are the*

*spherical harmonics described in Example B.3. As discussed in [24, Chapter 4], for any $k > 0$ $u \in \mathcal{H}$ belongs to the Sobolev space $H_0^k(\mathbb{S}^d)$ if and only if: for each $f \in \mathcal{H}$ the coefficients $(\mu_j)_{j=0}^\infty$ in the representation* (10) *satisfy*

$$\mu_j \in \Theta((j+1)^{-2s}).$$

*As discussed in [22, Lemma B.1] in this case the norms of $\mathcal{H}$ and on $H^s(\mathbb{S}^d)$ are equivalent[9].*

## B.2 Preliminaries: regular domains and Sobolev spaces

We consider the case where $\mathcal{X}$ is a sufficiently regular open subset of the $d$-dimensional Euclidean space so that we may comfortably describe the smoothness of functions on $\mathcal{X}$ using classical notions of tools. We therefore assume the following.

**Assumption (DR)** (Domain Regularity). *Either one of the following holds:*

(i) ***Compact manifold:*** *$(\mathcal{X}, g)$ is a $d$-dimensional compact Riemannian manifold, with $d \geq 2$, and $\mu = V_g(\mathcal{X})^{-1} V_g$ is the uniform law on $(\mathcal{X}, g)$ and $V_g$ is its volume measure,*

(ii) ***Bounded Euclidean domain:*** *$\mathcal{X}$ is a non-empty bounded open domain in $\mathbb{R}^d$ with Lipschitz boundary and $\mu$ is the normalized Lebesgue measure.*

In cases (i)-(ii), $\mathcal{X}$ can be viewed as a Riemannian manifold (possibly with boundary).

In what follows, we let $\Delta$ denote either of the following Laplacian, depending on which of assumption (DR) we have assumed. If (i) or (ii) holds then $\Delta$ will denote the usual Laplacian-Beltrami operator on $\mathcal{X}$, note that when (iii) holds then $\Delta = \sum_{i=1}^d \frac{\partial^2}{\partial x_i^2}$. We denote the eigenfunctions of $\Delta$ by $(\bar{\psi}_i)_{i \in \mathbb{N}}$ whose respective eigenvalues $\lambda_1 \leq \lambda_2 \leq \dots$ are arranged in a non-decreasing order. If (iii) holds, then $\Delta$ will denote the *Dirichlet Laplacian* (see [11, Section 6.1.2]) on $\mathcal{X}$ (with its usual Euclidean Riemannian metric) which acts on the spaces of functions vanishing on the boundary $\partial\mathcal{X}$ of $\mathcal{X}$; namely on the homogeneous Sobolev space $H_0^1(\mathcal{X})$.

**Example B.2.** *Under assumption (i), the eigenfunctions of $\Delta$ are the Hermite polynomials on $L_\mu^2(\mathbb{R}^d)$*

$$\bar{\psi}_i(x) = (-1)^i e^{x^2/2} \frac{\partial^i}{\partial x^i}(e^{-x^2/2}) \qquad \text{for each } i \in \mathbb{N}_+.$$

*For concreteness, the size, in $L_\mu^1$-norm, of the Hermite polynomials is computed in [28, Theorem 2.1] where it it shown that $\|\bar{\psi}_i\|_{L_\mu^1} \lesssim \frac{(i!)^{1/2}}{i^{1/2}}(1 + \mathcal{O}(1/i))$.*

**Example B.3.** *Suppose that $\mathcal{X}$ is the $d$-dimensional sphere $\mathbb{S}^d \overset{\text{def.}}{=} \{x \in \mathbb{R}^{d+1} : \|x\|_2 = 1\}$ equipped with the (usual) Riemannian metric induced by inclusion into the $d+1$-dimensional Euclidean space $\mathbb{R}^{d+1}$. Let $\mu$ be the normalized (uniform) Riemannian (volume) measure on $\mathbb{R}^d$. Then, $(\bar{\psi}_{i,j}(x))_{i=1,j=1}^{\infty, J_i}$ are an orthonrmal basis of $L_\mu^2$ consisting of spherical Harmonics (counting multiplicities $J_i \in \mathbb{N}_+$ for each eigenvalue $i \in \mathbb{N}_+$); that is the restriction of a homogeneous polynomial $f : \mathbb{R}^{d+1} \to \mathbb{R}$ satisfying $\Delta f = 0$ to there sphere $\mathbb{S}^d$. Spherical harmonics are not very large (in uniform norm) [17, Theorem 1.6] and one can show that $\|\bar{\psi}_{i,j}\| \lesssim \lambda_i^{(d-1)/4}$, for each $i \in \mathbb{N}_+$. See [2, Chapter 2] for further details on spherical harmonics.*

### B.2.1 Sobolev spaces on Riemannian manifolds

Let $(\mathcal{X}, g)$ be a compact (smooth) Riemannian manifold, possibly with a boundary $\partial\mathcal{X}$. For any $k \in \mathbb{N}_+$, we define $\mathcal{C}_k^2(M)$ be the set of smooth functions $f$ on $\mathcal{X}$ satisfying

$$\underbrace{\int_M |\nabla^i f|_g^2 \, d\mu < \infty}_{\text{Square-Integrable Partial Derivatives}} \quad \text{and} \quad \underbrace{f(x) = 0 \; (\forall x \in \partial\mathcal{X})}_{\text{Homogeneous Boundary Conditions}} \tag{11}$$

where $\mu$ is the Riemannian volume measure on $(\mathcal{X}, g)$, $\nabla$ is the covariant derivative thereon. Then, the Sobolev space $H_0^k(\mathcal{X}, g)$ is the Hilbert space obtained as the completion of $\mathcal{C}_k^p(\mathcal{X})$ with respect

---

[9]Meaning that, there exist constants $0 < c \leq C$ such that: for each $u \in H^s(\mathbb{S}^d)$ we have $c\|u\|_{\mathcal{H}} \leq \|\cdot\|_{H^s(\mathbb{S}^d)} \leq C\|\cdot\|_{\mathcal{H}}$.

to the norm

$$\|f\|_{H^k(\mathcal{X})} \stackrel{\text{def.}}{=} \sum_{i=1}^{k} \|\nabla^i f\|_{L^2_\mu}.$$

*Remark* B.4 (The Sobolev Spaces $H_0^k(\mathcal{X}, g)$ and $H^k(\mathcal{X}, g)$). Observe that when $\mathcal{X}$ is a compact Riemannian manifold without boundary, i.e. when $\partial\mathcal{X} = \emptyset$; then the condition $f(x) = 0$ for all $x$ in the boundary set $\partial\mathcal{X}$ holds vacuously. In this case, this condition can be ignored, and the spaces $H_0^k(\mathcal{X}, g)$ coincide with the usual Sobolev spaces $H^k(\mathcal{X}, g)$ for Riemannian manifolds without boundary; where $H^k(\mathcal{X}, g)$ is defined much the same way as $H_0^k(\mathcal{X}, g)$ but without requiring the homogeneous boundary condition in (11).

A more convenient expression for the norm of Sobolev spaces on spheres can be arrived at by manipulating the definition of the Laplacian. We summarize this norm in the next example.

**Example B.5** (Sobolev spaces on spheres). *Fix $k \in \mathbb{N}_+$. Following the discussion on [2, page 120-121], the $f \in H_0^k(\mathbb{S}^d, g)$ if and only if the following norm*

$$\|f\|_k^2 \stackrel{\text{def.}}{=} \sum_{j=0}^{\infty} \sum_{i=1}^{J_i} \underbrace{(j + c_d)^{2k}}_{\text{spectral decay}} \underbrace{\left| \int_{x \in \mathbb{S}^d} f(x)\bar{\psi}_{i,j}(x) \, \mu(dx) \right|^2}_{\text{projection onto } (j,i)^{th} \text{ spherical harmonic}} \tag{12}$$

*is finite; where $c_d \stackrel{\text{def.}}{=} (d - 2)/2$. That is, the norms $\|\cdot\|_k$ and $\|\cdot\|_{H^k(\mathcal{X})}$ are equivalent.*

The key takeaway of Example B.5, is that the *smoothness* of a function on there sphere $(\mathbb{S}^d, g)$, i.e. the largest $k$ for which $f \in H_0^k(\mathbb{S}^d, g)$, can be expressed in terms of the decay rate of its projection onto the basis of spherical harmonics. Indeed, this expression (12) implies that $f \in H_0^k(\mathbb{S}^d, g)$ only if $\left| \int_{x \in \mathbb{S}^d} f(x)\bar{\psi}_{i,j}(x) \, \mu(dx) \right|^2 \in \mathcal{O}\big((j + c_d)^{-2k-\epsilon}\big)$ for all $\varepsilon > 0$.

This observation was used in [24, Chapter 4] and circa [22, Lemma B.1] to characterize the smoothness of functions in the RKHS associated to certain radial kernels, e.g. certain NTK limits. We now generalize this argument, and use it to relate the *spectral decay* rate of a kernel to the smoothnes/regularity of the functions in its associated RKHS.

### B.3 Standardization operator

We require that the $L^2_\mu(\mathcal{X})$ norm of the eigenfunctions are non-vanishing and non-exploding.

**Assumption (ND)** (Non-degenerate $L^2$-Norm). *Suppose that:*

 *(i) Non-vanishing: $0 < \inf_{i \in [M]} \|\bar{\psi}_i\|_{L^2_\mu(\mathcal{X})}$,*

 *(ii) Non-exploding: $\sup_{i \in [M]} \|\bar{\psi}_i\|_{L^2_\mu(\mathcal{X})} < \infty$.*

*Again, if $(\bar{\psi}_i)_{i=0}^{\infty}$ is orthonormal then (i) and (ii) necessarily hold and this is a non-assumption.*

To generalize the discussion in Example B.1, we first define the linear *standardization operator*

$$A : \mathcal{H} \to L^2_\mu(\mathcal{X})$$

$$f = \sum_{i=0}^{M} \langle f, \psi_i \rangle_\kappa \psi_i \mapsto \sum_{i=0}^{M} \frac{\langle f, \psi_i \rangle_\kappa \|\psi_i\|_\kappa}{\|\bar{\psi}_i\|_{L^2_\mu(\mathcal{X})}} \bar{\psi}_i. \tag{13}$$

**Example B.6** (Standardization Is Inclusion for Radial Kernels on Spheres). *In the setting of Example B.1, we have that (up to reordering) $\psi_{i,j} = \bar{\psi}_{i,j}$. Since, in that case, for $k$ large enough on has $\mathcal{H} = H_0^k(\mathbb{S}^d, g)$ then $A$ is simply the inclusion operator of $H_0^k(\mathbb{S}^d, g)$ into $L^2_\mu(\mathbb{S}^d)$.*

In what follows, when we assume Assumption (DR) (iii), we will use $C^\infty(\overline{\mathcal{X}})$ to denote the set of smooth functions on the closure $\overline{\mathcal{X}}$ of $\mathcal{X}$ in the norm topology on $\mathbb{R}^d$. Under Assumption (DR) (i)-(ii), we use $C^\infty(\overline{\mathcal{X}})$ to denote $C^\infty(\mathcal{X})$; that is, the set of smooth functions on the smooth manifold $\mathcal{X}$.

**Proposition B.7** (Identification: Spectral Decay and Standardized Regularity). *Suppose that Assumption (DR) holds and that $(\psi_i)_{i=0}^M$ are orthonormal in $\mathcal{H}$.*

*Then, the linear operator $A$, defined in (13), is an isometric embedding when $M \leq \infty$ (resp. isometric isomorphism when $M = \infty$). Moreover, it characterizes the "regularity" of functions in $\mathcal{H}$ via:*

1. ***Sub-Exponential Decay:*** *If $\kappa$ has infinite-rank ($M = \infty$) and there exists an $r > 0$ such that $|\langle f, \psi_i \rangle_\kappa| \lesssim i^{-r}$ and $\inf_{\varepsilon > 0} \lim_{i \to \infty} |\langle f, \psi_i \rangle_\kappa| \, i^{-r+\varepsilon} = \infty$ then*

$$\underbrace{A(f) \in H_0^{\frac{(1+r)d}{2}}(\mathcal{X}) \setminus \bigcap_{\varepsilon > 0} H_0^{\frac{(1+r)d}{2}+\varepsilon}(\mathcal{X})}_{\textit{Sobolev Space Characterization}} \quad \textit{and} \quad \underbrace{A(f) \in \bigcap_{rd/2 > k} C^k(\mathcal{X})}_{C^k-\textit{Space Description}}.$$

2. ***Exponential Decay:*** *If $\kappa$ has infinite-rank ($M = \infty$) and there exists some $r > 0$ such that $|\langle f, \psi_i \rangle_\kappa| \lesssim e^{-ri}$, then*
$$A(f) \in C^\infty(\overline{\mathcal{X}}).$$

3. ***Finite-Rank:***

    (ii) *If Assumption (DR) (ii) holds then: $\kappa$ has finite rank ($M < \infty$) then $A(f)$ is smooth.*
    (iii) *If Assumption (DR) (iii) holds then: $\kappa$ has finite rank ($M < \infty$) then $A(f)$ is real-analytic.*

A direct corollary of the first statement in Proposition B.7 guarantees that $A$ preserves convergence. Consequentially, any finite-rank truncation or any sequence of learners converging to a limiting function in the RKHS do so if and only if their standardized versions do; moreover, both sequences converge at the same speed due to $A$ being an isometric embedding.

**Corollary B.8** (Preservation of Convergence Rates). *If $(f_N)_{N \in \mathbb{N}}$ is a sequence in $\mathcal{H}$ converging to some $f \in \mathcal{H}$ then: for each $N \in \mathbb{N}$ on has $\|f_N - f\|_\kappa = \|A(f_N) - A(f)\|_{L_\mu^2(\mathbb{R}^d)}$.*

We now are now prepared to prove Proposition B.7.

*Proof of Proposition B.7.* We first show that $A$ is a linear isometric embedding (resp. linear isomorphism). This allows us to relay the spectral decay rate of any function $f$ in $\mathcal{H}$ to that of its "standardization" $A(f)$ in $L_\mu^2(\mathcal{X})$. We then use this identification, together with results from spectral theory (see [11] for references) to obtain our conclusion.

**Step 1 - $A$ is a Linear Isometric Embedding (Resp. Linear Isomorphism):**
Observe that $A$ admits a linear left-inverse given by

$$B : L_\mu^2(\mathcal{X}) \to \mathcal{H}$$
$$f = \sum_{i=0}^\infty \langle f, \bar{\psi}_i \rangle_{L_\mu^2(\mathcal{X})} \, \bar{\psi}_i \mapsto \sum_{i=0}^M \langle f, \psi_i \rangle_{L_\mu^2(\mathcal{X})} \, \|\bar{\psi}_i\|_{L_\mu^2(\mathcal{X})} \, \psi_i.$$

In particular, $A$ is a linear isomorphism if and only if $M = \infty$; since otherwise $B$ is not a two-sided (linear) inverse. Furthermore, $A$ is an isometric embedding (resp. isometric isomorphism when $M = \infty$) since due to the following. For each $f \in \mathcal{H}$ we have that

$$\|A(f)\|_{L_\mu^2(\mathcal{X})}^2 = \sum_{i=0}^M \frac{|\langle f, \psi_i \rangle_\kappa|^2 \, \|\psi_i\|_\kappa^2}{\|\bar{\psi}_i\|_{L_\mu^2(\mathcal{X})}^2} \, \|\bar{\psi}_i\|_{L_\mu^2(\mathcal{X})}^2 \tag{14}$$

$$= \sum_{i=0}^M \frac{|\langle f, \psi_i \rangle_\kappa|^2 \cdot 1}{\|\bar{\psi}_i\|_{L_\mu^2(\mathcal{X})}^2} \, \|\bar{\psi}_i\|_{L_\mu^2(\mathcal{X})}^2 \tag{15}$$

$$= \sum_{i=0}^M \frac{|\langle f, \psi_i \rangle_\kappa|^2}{\|\bar{\psi}_i\|_{L_\mu^2(\mathcal{X})}^2} \, \|\bar{\psi}_i\|_{L_\mu^2(\mathcal{X})}^2$$

$$= \sum_{i=0}^M |\langle f, \psi_i \rangle_\kappa|^2 \, 1$$

$$= \sum_{i=0}^{M} |\langle f, \psi_i \rangle_\kappa|^2 \, \|\psi_i\|_\kappa^2 \tag{16}$$

$$= \|f\|_\kappa^2, \tag{17}$$

where (14) and (16) held by virtue of $(\psi_i)_{i=0}^M$ being an orthonormal basis of $\mathcal{H}$. From (14)-(17) we conclude that for each $f \in \mathcal{H}$ we have $\|A(f)\|_{L_\mu^2(\mathcal{X})} = \|f\|_\kappa$. Thus, $A$ is an isometric embedding.

**Finite Rank Kernel:**
Suppose that $M < \infty$. We consider two cases in Assumption (DR) separately.

Under Assumption (DR) (ii), [11, Corollary 9.28] guarantees that each $\bar{\psi}_0, \dots, \bar{\psi}_M$ is smooth. Since any finite sum of real analytic functions is again smooth then, in this case, $A(f)$ is smooth.

Under Assumption (DR) (iii), [11, Theorem 6.8] each $\bar{\psi}_0, \dots, \bar{\psi}_M$ is real-analytic. Since any finite sum of real analytic functions is again real-analytic then $A(f)$ is real-analytic.

**Infinite-Finite Rank Kernel:**
Suppose that $M = \infty$ and fix $s > 0$. By definition of $(\psi_i)_{i=0}^\infty$ and of the Sobolev space $H^s(\mathcal{X})$ we have that: for each $f \in L_\mu^2(\mathcal{X})$

$$
\begin{aligned}
\|f\|_{H^s(\mathcal{X})}^2 &= \sum_{i=0}^\infty |\langle f, \bar{\psi}_i \rangle_{L_\mu^2(\mathcal{X})}|^2 \, \langle (1-\Delta)^s \bar{\psi}_i, \bar{\psi}_i \rangle_{L_\mu^2(\mathcal{X})} \\
&= \sum_{i=0}^\infty |\langle f, \bar{\psi}_i \rangle_{L_\mu^2(\mathcal{X})}|^2 (1 - \bar{\lambda}_i)^s.
\end{aligned}
\tag{18}
$$

Since $f \in L_\mu^2(\mathcal{X})$ if and only if $\|f\|_{H^s(\mathcal{X})}$ is well-defined and finite. Consequentially, the elements of the Sobolev space $H^s(\mathcal{X})$ are characterized by the summability of the right-hand side of (18). That is, for each $f \in L_\mu^2(\mathcal{X})$ one has

$$f \in H^s(\mathcal{X}) \Leftrightarrow \sum_{i=0}^\infty |\langle f, \bar{\psi}_i \rangle_{L_\mu^2(\mathcal{X})}|^2 (1 - \bar{\lambda}_i)^s < \infty. \tag{19}$$

By Assumption (ND), both $0 < \underline{C} \overset{\text{def.}}{=} \inf_{i \in [M]} \|\bar{\psi}_i\|_{L_\mu^2(\mathcal{X})}$ and $\bar{C} \overset{\text{def.}}{=} \sup_{i \in [M]} \|\bar{\psi}_i\|_{L_\mu^2(\mathcal{X})} < \infty$ are well-defined, non-negative real numbers. Therefore, for every $i \in \mathbb{N}$

$$0 < \frac{|\langle f, \psi_i \rangle_\kappa|^2}{\bar{C}} \le \frac{|\langle f, \psi_i \rangle_\kappa|^2}{\|\bar{\psi}_i\|_{L_\mu^2(\mathcal{X})}} \le \frac{|\langle f, \psi_i \rangle_\kappa|^2}{\underline{C}} < \infty. \tag{20}$$

From (20) we deduce that: for each $f \in L_\mu^2(\mathcal{X})$ and every $s > 0$

$$\frac{1}{\bar{C}} \sum_{i=0}^\infty |\langle f, \psi_i \rangle_\kappa|^2 (1 - \bar{\lambda}_i)^s \le \sum_{i=0}^\infty \frac{|\langle f, \psi_i \rangle_\kappa|^2}{\|\bar{\psi}_i\|_{L_\mu^2(\mathcal{X})}} (1 - \bar{\lambda}_i)^s \le \frac{1}{\underline{C}} \sum_{i=0}^\infty |\langle f, \psi_i \rangle_\kappa|^2 (1 - \bar{\lambda}_i)^s.$$

Consequentially, (18) implies that

$$A(f) \in H^s(\mathcal{X}) \Leftrightarrow \sum_{i=0}^\infty |\langle f, \psi_i \rangle_\kappa|^2 (1 - \bar{\lambda}_i)^s < \infty. \tag{21}$$

**Step 2 - Asymptotics of Laplacian Eigenspectrum** Under Assumption (DR) (ii), Weyl's law for compact Riemannian manifolds (see e.g. [11, Corollary 9.35]) implies that there are dimensional (depending on $d$, $\mathcal{X}$, and on $g$) constants $0 < C_{W:1}^g \le C_{W:2}^g < \infty$ such that: for each $i \in \mathbb{N}$

$$0 < C_{W:1}^g \left( \frac{i}{v_d \, V_g(\mathcal{X})} \right)^{2/d} \le \bar{\lambda}_i \le C_{W:2}^g \left( \frac{i}{v_d \, V_g(\mathcal{X})} \right)^{2/d} < \infty, \tag{22}$$

where $v_d \overset{\text{def.}}{=} V(\{x \in \mathbb{R}^d : \|x\|_2 \le 1\}) = \frac{\pi^{d/2}}{\Gamma(1+d/2)}$ and where $V$ is the $d$-dimensional Lebesgue measure on $\mathbb{R}^d$.

Under Assumption (DR) (iii), Weyl's law (see e.g. [11, Theorem 6.27]) implies that: there are dimensional (depending on $d$ and on $\mathcal{X}$) constants $0 < \tilde{C}_{W:1} \leq \tilde{C}_{W:2} < \infty$ such that: for each $i \in \mathbb{N}$

$$0 < \tilde{C}_{W:1} \left( \frac{i}{v_d V(\mathcal{X})} \right)^{2/d} \leq \bar{\lambda}_i \leq \tilde{C}_{W:2} \left( \frac{i}{v_d V(\mathcal{X})} \right)^{2/d} < \infty. \tag{23}$$

With some abuse of notation, for $i = 1, 2$, we write $C_{W:i}$ for $C_{W:i}^g$ or $\tilde{C}_{W:i}$ depending on which of Assumptions (DR) (ii) or (iii) are being used. We therefore, wild a mild abuse of notation, we collect (22) and (23) into the following single expression:

$$0 < \tilde{C}_{W:1} \left( c\, i \frac{1}{v_d V(\mathcal{X})} \right)^{2/d} \leq \bar{\lambda}_i \leq \tilde{C}_{W:2} \left( c\, i \right)^{2/d} < \infty, \tag{24}$$

where the constant $c > 0$ is defined by

$$c \overset{\text{def.}}{=} \begin{cases} \frac{\pi^{d/2}}{\Gamma(1+d/2)\, V_g(\mathcal{X})} & : \text{if Assumptions (DR) (ii) holds} \\ \frac{\pi^{d/2}}{\Gamma(1+d/2)\, V(\mathcal{X})} & : \text{if Assumptions (DR) (iii) holds} \end{cases}.$$

Combining (21) with (24), we find that

$$A(f) \in H^s(\mathcal{X}) \Leftrightarrow \sum_{i=0}^{\infty} |\langle f, \psi_i \rangle_\kappa|^2 \, (c\, i)^{2s/d} < \infty. \tag{25}$$

Pulling out the common $c^{2s/d}$ factor in all summands on the right-hand side of (25) we find that: for any $s > 0$ any $f \in L_\mu^2(\mathcal{X})$ belongs to $H^s(\mathcal{X})$ if and only if

$$A(f) \in H^s(\mathcal{X}) \Leftrightarrow \sum_{i=0}^{\infty} |\langle f, \psi_i \rangle_\kappa|^2 \, i^{2s/d} < \infty. \tag{26}$$

1. **Sub-Exponential Decay:** Suppose that there exists an $r > 0$ such that $|\langle f, \psi_i \rangle_\kappa| \lesssim i^{-r}$ and for every $\varepsilon > 0$ we have $\lim_{i \to \infty} |\langle f, \psi_i \rangle_\kappa| \, i^{-r+\varepsilon} = \infty$ then (26) implies that

$$A(f) \in H_0^{\frac{(1+r)d}{2}}(\mathcal{X}) \setminus \bigcap_{\varepsilon > 0} H_0^{\frac{(1+r)d}{2}+\varepsilon}(\mathcal{X}).$$

If Assumption (DR) (ii) or (iii) holds the Sobolev embedding theorem for Riemannian manifolds with boundary, as formulated in [11, Theorem 9.26], implies that $H_0^{s_r}(\mathcal{X}) \subseteq C^{k_r}(\mathcal{X})$ where $s_r - d/2 > k_r$ for some $k_r \geq 0$ to be determined shortly; where for us $s_r \overset{\text{def.}}{=} (1+r)d/2$. Therefore, $dr/2 > k_r$. Consequentially, for each $f \in \mathcal{H}$ we have that

$$A(f) \in \bigcap_{rd/2 > k} C^k(\mathcal{X}).$$

2. **Exponential Decay:** Suppose that there exists some $r > 0$ such that $|\langle f, \psi_i \rangle_\kappa| \lesssim e^{-r\, i}$. Then, then (26) implies that $A(f) \in \cap_{s>0} H_0^s(\mathcal{X})$. By the Sobolev Embedding Theorem, in the bounded case ([11, Theorem 9.2.6])

$$A(f) \in C^\infty(\overline{\mathcal{X}}).$$

$\square$

## C    Master inequalities

In this section, we will present and proof the Master inequalities, which is crucial in our argument.

Our approach bounds each term in the bias-variance decomposition in line (5):

$$\mathcal{R} = \mathbb{E}_{\boldsymbol{\epsilon}}\left[\|\hat{\boldsymbol{\theta}}(\mathbf{X}\boldsymbol{\theta}^* + \boldsymbol{\epsilon}) - \boldsymbol{\theta}^*\|_{\boldsymbol{\Sigma}}^2\right] = \mathcal{B} + \mathcal{V},$$

under minimal assumptions.

### C.1    More notations

We first introduce some more notations. We define the empirical covariance matrix:

$$\hat{\boldsymbol{\Sigma}} \overset{\text{def.}}{=} \frac{1}{n}\mathbf{X}^\top\mathbf{X} = \frac{1}{n}\sum_{i=1}^n \mathbf{x}_i\mathbf{x}_i^\top \in \mathbb{R}^{p\times p},$$

and denote by

$$\mathbf{z} \overset{\text{def.}}{=} \boldsymbol{\Sigma}^{-1/2}\mathbf{x} \in \mathbb{R}^p$$

the whitened/isotropic feature vector where $\mathbb{E}_\mu\left[\mathbf{z}\mathbf{z}^\top\right] = \mathbf{I}_p \in \mathbb{R}^{p\times p}$. Denote the whitened input matrix by $\mathbf{Z} \overset{\text{def.}}{=} \mathbf{X}\boldsymbol{\Sigma}^{-1/2} \in \mathbb{R}^{n\times p}$, then the rows of $\mathbf{Z}$ are isotropic centered i.i.d. random vectors.

We introduce the notation of a generalized vector norm, which will be frequently used in this paper:

**Definition C.1** (Generalized vector norm). We denote $\|\mathbf{v}\|_{\mathbf{M}} \overset{\text{def.}}{=} \sqrt{\mathbf{v}^\top\mathbf{M}\mathbf{v}}$ for any positive definite matrix $\mathbf{M}$ and vector $\mathbf{v}$.

*Remark* C.2 (Examples). Let $\mathbf{v} \in \mathbb{R}^p$ and we have $\|\mathbf{v}\|_2^2 = \mathbf{v}^\top\mathbf{I}_p\mathbf{v} = \|\mathbf{v}\|_{\mathbf{I}_p}^2$, where $\mathbf{I}_p$ is the identity matrix. Another example is that the euclidean norm of $\mathbf{x}$ equals to the $\boldsymbol{\Sigma}$-norm of its whitened counterpart $\mathbf{z}$:

$$\|\mathbf{x}\|_2^2 = \mathbf{x}^\top\mathbf{x} = \mathbf{z}^\top\boldsymbol{\Sigma}\mathbf{z} = \|\mathbf{z}\|_{\boldsymbol{\Sigma}}^2.$$

Furthermore, if $\mathbf{M}_1 \preccurlyeq \mathbf{M}_2$, we have $\|\mathbf{v}\|_{\mathbf{M}_1} \le \|\mathbf{v}\|_{\mathbf{M}_2}$. In particular, since $\boldsymbol{\Sigma} \preccurlyeq \lambda_1\mathbf{I}_p$, we have

$$\|\mathbf{v}\|_{\boldsymbol{\Sigma}} \le \sqrt{\lambda_1}\|\mathbf{v}\|_2.$$

Lastly, it is convenient to write the expected values in a generalized vector norm: fix a distribution $\mu$, an input block $\mathbf{X} \in \mathbb{R}^{n\times p}$, where its rows are drawn i.i.d. by $\mu$, and denote by $\hat{\mu} = \frac{1}{n}\sum_{i=1}^n \delta_{\mathbf{x}_i}$ the empirical distribution of $\mu$, then for any $\mathbf{v} \in \mathbb{R}^p$, we have

$$\mathbb{E}_{\mathbf{x}\sim\mu}\left[(\mathbf{x}^\top\mathbf{v})^2\right] = \|\mathbf{v}\|_{\boldsymbol{\Sigma}}^2; \quad \mathbb{E}_{\mathbf{x}\sim\hat{\mu}}\left[(\mathbf{x}^\top\mathbf{v})^2\right] = \|\mathbf{v}\|_{\hat{\boldsymbol{\Sigma}}}^2.$$

**Definition C.3** (Truncated Terms). For any vector $\mathbf{v} \in \mathbb{R}^M$, write $\mathbf{v} = \mathbf{v}_{\le k} \oplus \mathbf{v}_{>k} \in \mathbb{R}^k \oplus \mathbb{R}^{M-k}$. For any square matrix $\mathbf{M} \in \mathbb{R}^{M\times M}$, write $\mathbf{M} = \begin{pmatrix} \mathbf{M}_{\le k} & * \\ * & \mathbf{M}_{>k} \end{pmatrix}$ with $\mathbf{M}_{\le k} \in \mathbb{R}^{k\times k}$ and $\mathbf{M}_{>k} \in \mathbb{R}^{(M-k)\times(M-k)}$. Analogously, for any non-square matrix $\mathbf{M} \in \mathbb{R}^{N\times M}$, we write $\mathbf{M} = \mathbf{M}_{\le k} \oplus \mathbf{M}_{>k} \in \mathbb{R}^{N\times k} \oplus \mathbb{R}^{N\times(M-k)}$. Also write $\mathbf{M}_{-k} \overset{\text{def.}}{=} \mathbf{M}_{\le k-1} \oplus \mathbf{M}_{>k} \in \mathbb{R}^{N\times(M-1)}$ and $\mathbf{M}_{l:j} = [\mathbf{M}_{i,k}]_{k=l+1}^j \in \mathbb{R}^{N\times(j-l)}$.

Denote $\mathbf{A} \overset{\text{def.}}{=} \mathbf{X}\mathbf{X}^\top + n\lambda\mathbf{I}_n \in \mathbb{R}^{n\times n}$. For any $k \in \mathbb{N}$, denote

$$\mathbf{A}_k = \mathbf{X}_{>k}\mathbf{X}_{>k}^\top + n\lambda\mathbf{I}_n \in \mathbb{R}^{n\times n}.$$

The symmetric matrix $\mathbf{A}_k$ plays an important role in the analysis. Intuitively, when $k$ is chosen appropriately, $\mathbf{A}_k$ is approximately equal to a scaled identity, with spectrum bounded by some constants from below and from above.

We introduce a few more definitions that have been used in the previous literature.

**Definition C.4** (Effective ranks [6]). Let $k \in \mathbb{N}$. Define two quantities:

$$r_k \overset{\text{def.}}{=} \frac{\text{Tr}[\boldsymbol{\Sigma}_{>k}]}{\|\boldsymbol{\Sigma}_{>k}\|_{\text{op}}} = \frac{\sum_{l=k+1}^p \lambda_l}{\lambda_{k+1}}, \quad R_k \overset{\text{def.}}{=} \frac{\text{Tr}[\boldsymbol{\Sigma}_{>k}]^2}{\text{Tr}[\boldsymbol{\Sigma}_{>k}^2]} = \frac{\left(\sum_{l=k+1}^p \lambda_l\right)^2}{\sum_{l=k+1}^p \lambda_l^2}.$$

Note that we have $r_k \le R_k \le r_k^2$. [10]

---

[10]See [6] for details.

**Definition C.5** (Concentration coefficients [7])**.** Define the quantities: [11]

$$\rho_{n,k} \stackrel{\text{def.}}{=} \frac{n\,\|\boldsymbol{\Sigma}_{>k}\|_{\text{op}} + s_1(\mathbf{A}_k)}{s_n(\mathbf{A}_k)}; \qquad \zeta_{n,k} \stackrel{\text{def.}}{=} \frac{s_1(\mathbf{Z}_{\leq k}^\top \mathbf{Z}_{\leq k})}{s_k(\mathbf{Z}_{\leq k}^\top \mathbf{Z}_{\leq k})}; \qquad \xi_{n,k} \stackrel{\text{def.}}{=} \frac{s_1(\mathbf{Z}_{\leq k}^\top \mathbf{Z}_{\leq k})}{n}.$$

where $s_i(\cdot)$ denotes the $i$-th largest singular value of the matrix.

To enhance readability, we will omit the subscript $\cdot_{n,k}$ when the context is clear. The intuition of the concentration coefficients is that for fixed $k \in \mathbb{N}$ and $n$ large enough, $\zeta$ and $\xi$ concentrate around some constants; when $k$ is smaller but scales with $n$, $\rho$ converges to a constant, as it essentially represents the condition number of $\mathbf{A}_k$. As mentioned earlier, the spectrum of $\mathbf{A}_k$ is bounded. Therefore, selecting an appropriate $k$ ensures that all three concentration coefficients converge to certain constants.

## C.2 Statements and proofs

Now we are ready to present the Master inequalities, which are proved in [46, 7]:

**Proposition C.6** (Master inequality for $\mathcal{B}$, [46, 7])**.** *Let $k \in \mathbb{N}$ be an integer. For any $\delta \in (0,1)$, with probability at least $1 - \delta$,*

$$\mathcal{B} \leq \left(\frac{1 + \rho^2\zeta^2\xi^{-1} + \rho}{\delta}\right) \|\boldsymbol{\theta}_{>k}^*\|_{\boldsymbol{\Sigma}_{>k}}^2 + (\zeta^2\xi^{-2} + \rho\zeta^2\xi^{-1})\frac{s_1(\mathbf{A}_k)^2}{n^2} \|\boldsymbol{\theta}_{\leq k}^*\|_{\boldsymbol{\Sigma}_{\leq k}^{-1}}^2.$$

*Proof.* We begin by bounding the terms that appear in Lemma H.8: plug in Definition C.5 of $\rho, \zeta, \xi$, since

$$\frac{s_1(\mathbf{A}_k^{-1})^2}{s_n(\mathbf{A}_k^{-1})^2} \frac{s_1(\mathbf{Z}_{\leq k}^\top \mathbf{Z}_{\leq k})}{s_k(\mathbf{Z}_{\leq k}^\top \mathbf{Z}_{\leq k})^2} = \frac{s_1(\mathbf{A}_k)^2}{s_n(\mathbf{A}_k)^2} \frac{s_1(\mathbf{Z}_{\leq k}^\top \mathbf{Z}_{\leq k})^2}{s_k(\mathbf{Z}_{\leq k}^\top \mathbf{Z}_{\leq k})^2} \frac{n}{s_1(\mathbf{Z}_{\leq k}^\top \mathbf{Z}_{\leq k})} \frac{1}{n} \leq \rho^2\zeta^2\xi^{-1} \cdot \frac{1}{n}$$

$$\frac{1}{s_n(\mathbf{A}_k^{-1})^2 s_k(\mathbf{Z}_{\leq k}^\top \mathbf{Z}_{\leq k})^2} = \frac{s_1(\mathbf{A}_k)^2}{s_k(\mathbf{Z}_{\leq k}^\top \mathbf{Z}_{\leq k})^2} = \frac{s_1(\mathbf{A}_k)^2}{n^2} \frac{s_1(\mathbf{Z}_{\leq k}^\top \mathbf{Z}_{\leq k})^2}{s_k(\mathbf{Z}_{\leq k}^\top \mathbf{Z}_{\leq k})^2} \frac{n^2}{s_1(\mathbf{Z}_{\leq k}^\top \mathbf{Z}_{\leq k})^2} = \frac{s_1(\mathbf{A}_k)^2}{n^2} \cdot \zeta^2\xi^{-2}$$

$$\|\boldsymbol{\Sigma}_{>k}\|_{\text{op}}\, s_1(\mathbf{A}_k^{-1}) = \frac{\|\boldsymbol{\Sigma}_{>k}\|_{\text{op}}}{s_n(\mathbf{A}_k)} = \frac{n\,\|\boldsymbol{\Sigma}_{>k}\|_{\text{op}}}{s_n(\mathbf{A}_k)} \frac{1}{n} \leq \rho \cdot \frac{1}{n}$$

$$\|\boldsymbol{\Sigma}_{>k}\|_{\text{op}} \frac{s_1(\mathbf{A}_k^{-1})}{s_n(\mathbf{A}_k^{-1})^2} \frac{s_1(\mathbf{Z}_{\leq k}^\top \mathbf{Z}_{\leq k})}{s_k(\mathbf{Z}_{\leq k}^\top \mathbf{Z}_{\leq k})^2} = \frac{n\,\|\boldsymbol{\Sigma}_{>k}\|_{\text{op}}}{s_n(\mathbf{A}_k)} \frac{s_1(\mathbf{A}_k)^2}{n^2} \frac{s_1(\mathbf{Z}_{\leq k}^\top \mathbf{Z}_{\leq k})^2}{s_k(\mathbf{Z}_{\leq k}^\top \mathbf{Z}_{\leq k})^2} \frac{n}{s_1(\mathbf{Z}_{\leq k}^\top \mathbf{Z}_{\leq k})} \leq \rho\zeta^2\xi^{-1} \frac{s_1(\mathbf{A}_k)^2}{n^2},$$

we have

$$\mathcal{B} \leq \|\boldsymbol{\theta}_{>k}^*\|_{\boldsymbol{\Sigma}_{>k}}^2 + \rho^2\zeta^2\xi^{-1} \frac{\|\mathbf{X}_{>k}\boldsymbol{\theta}_{>k}^*\|_2^2}{n}$$

$$+ \zeta^2\xi^{-2} \frac{s_1(\mathbf{A}_k)^2}{n^2} \|\boldsymbol{\theta}_{\leq k}^*\|_{\boldsymbol{\Sigma}_{\leq k}^{-1}}^2 \tag{27}$$

$$+ \rho \frac{\|\mathbf{X}_{>k}\boldsymbol{\theta}_{>k}^*\|_2^2}{n}$$

$$+ \rho\zeta^2\xi^{-1} \frac{s_1(\mathbf{A}_k)^2}{n^2} \|\boldsymbol{\theta}_{\leq k}^*\|_{\boldsymbol{\Sigma}_{\leq k}^{-1}}^2.$$

We regroup the terms into:

$$\mathcal{B} \leq \|\boldsymbol{\theta}_{>k}^*\|_{\boldsymbol{\Sigma}_{>k}}^2 + (\rho^2\zeta^2\xi^{-1} + \rho)\frac{\|\mathbf{X}_{>k}\boldsymbol{\theta}_{>k}^*\|_2^2}{n} + (\zeta^2\xi^{-2} + \rho\zeta^2\xi^{-1})\frac{s_1(\mathbf{A}_k)^2}{n^2} \|\boldsymbol{\theta}_{\leq k}^*\|_{\boldsymbol{\Sigma}_{\leq k}^{-1}}^2. \tag{28}$$

By Lemma G.6, we can further simplify the expression into a high probability bound: for any $\delta \in (0,1)$, with a probability at least $1 - \delta$, it holds that

$$\mathcal{B} \leq \left(1 + \frac{\rho^2\zeta^2\xi^{-1} + \rho}{\delta}\right) \|\boldsymbol{\theta}_{>k}^*\|_{\boldsymbol{\Sigma}_{>k}}^2 + (\zeta^2\xi^{-2} + \rho\zeta^2\xi^{-1})\frac{s_1(\mathbf{A}_k)^2}{n^2} \|\boldsymbol{\theta}_{\leq k}^*\|_{\boldsymbol{\Sigma}_{\leq k}^{-1}}^2. \tag{29}$$

Note that $1/\delta > 1$ and we obtain the result. □

---

[11]Alternatively, one can define a smaller $\rho_{n,k} \stackrel{\text{def.}}{=} \frac{\max\{n\|\boldsymbol{\Sigma}_{>k}\|_{\text{op}}, s_1(\mathbf{A}_k)\}}{s_n(\mathbf{A}_k)} \leq \frac{n\|\boldsymbol{\Sigma}_{>k}\|_{\text{op}} + s_1(\mathbf{A}_k)}{s_n(\mathbf{A}_k)}$. If $s_1(\mathbf{A}_k) = \Theta(n\lambda_k) = \Theta(n\|\boldsymbol{\Sigma}_{>k}\|_{\text{op}})$, the two definitions differs only by a constant.

**Proposition C.7** (Master inequality for $\mathcal{V}$, [46, 7]). *Let $k \in \mathbb{N}$ be an integer. Then the variance is bounded by:*

$$\mathcal{V}/\sigma^2 \le \rho^2 \left( \zeta^2 \xi^{-1} \frac{k}{n} + \frac{\operatorname{Tr}[\mathbf{Z}_{>k} \boldsymbol{\Sigma}_{>k}^2 \mathbf{Z}_{>k}^\top]}{n \operatorname{Tr}[\boldsymbol{\Sigma}_{>k}^2]} \frac{r_k(\boldsymbol{\Sigma})^2}{n R_k(\boldsymbol{\Sigma})} \right),$$

*where $\sigma^2 \overset{\text{def.}}{=} \mathbb{E}\left[\epsilon^2\right] \ge 0$ is the noise level.*

*Proof.* We start from Lemma H.9:

$$
\begin{aligned}
\mathcal{V}/\sigma^2 &\le \frac{s_1(\mathbf{A}_k^{-1})^2 \operatorname{Tr}[\mathbf{X}_{\le k} \boldsymbol{\Sigma}_{\le k}^{-1} \mathbf{X}_{\le k}^\top]}{s_n(\mathbf{A}_k^{-1})^2 s_k(\boldsymbol{\Sigma}_{\le k}^{-1/2} \mathbf{X}_{\le k}^\top \mathbf{X}_{\le k} \boldsymbol{\Sigma}_{\le k}^{-1/2})^2} + s_1(\mathbf{A}_k^{-1})^2 \operatorname{Tr}[\mathbf{X}_{>k} \boldsymbol{\Sigma}_{>k} \mathbf{X}_{>k}^\top] \\
&= \frac{s_1(\mathbf{A}_k)^2}{s_n(\mathbf{A}_k)^2} \frac{\operatorname{Tr}[\mathbf{Z}_{\le k} \mathbf{Z}_{\le k}^\top]}{s_k(\mathbf{Z}_{\le k}^\top \mathbf{Z}_{\le k})^2} + \frac{n^2 \|\boldsymbol{\Sigma}_{>k}\|_{\text{op}}^2}{s_n(\mathbf{A}_k)^2} \frac{\operatorname{Tr}[\mathbf{X}_{>k} \boldsymbol{\Sigma}_{>k} \mathbf{X}_{>k}^\top]}{n^2 \|\boldsymbol{\Sigma}_{>k}\|_{\text{op}}^2} \\
&\le \rho^2 \left( \frac{\operatorname{Tr}[\mathbf{Z}_{\le k} \mathbf{Z}_{\le k}^\top]}{s_k(\mathbf{Z}_{\le k}^\top \mathbf{Z}_{\le k})^2} + \frac{\operatorname{Tr}[\mathbf{X}_{>k} \boldsymbol{\Sigma}_{>k} \mathbf{X}_{>k}^\top]}{n^2 \|\boldsymbol{\Sigma}_{>k}\|_{\text{op}}^2} \right) \\
&\le \rho^2 \left( \frac{k s_1(\mathbf{Z}_{\le k}^\top \mathbf{Z}_{\le k})}{s_k(\mathbf{Z}_{\le k}^\top \mathbf{Z}_{\le k})^2} + \frac{\operatorname{Tr}[\mathbf{X}_{>k} \boldsymbol{\Sigma}_{>k} \mathbf{X}_{>k}^\top]}{n \operatorname{Tr}[\boldsymbol{\Sigma}_{>k}^2]} \frac{\operatorname{Tr}[\boldsymbol{\Sigma}_{>k}]^2}{n \|\boldsymbol{\Sigma}_{>k}\|_{\text{op}}^2} \frac{\operatorname{Tr}[\boldsymbol{\Sigma}_{>k}^2]}{\operatorname{Tr}[\boldsymbol{\Sigma}_{>k}]^2} \right) \\
&\le \rho^2 \left( \zeta^2 \xi^{-1} \frac{k}{n} + \frac{\operatorname{Tr}[\mathbf{X}_{>k} \boldsymbol{\Sigma}_{>k} \mathbf{X}_{>k}^\top]}{n \operatorname{Tr}[\boldsymbol{\Sigma}_{>k}^2]} \frac{r_k(\boldsymbol{\Sigma})^2}{n R_k(\boldsymbol{\Sigma})} \right) \\
&= \rho^2 \left( \zeta^2 \xi^{-1} \frac{k}{n} + \frac{\operatorname{Tr}[\mathbf{Z}_{>k} \boldsymbol{\Sigma}_{>k}^2 \mathbf{Z}_{>k}^\top]}{n \operatorname{Tr}[\boldsymbol{\Sigma}_{>k}^2]} \frac{r_k(\boldsymbol{\Sigma})^2}{n R_k(\boldsymbol{\Sigma})} \right)
\end{aligned}
$$

where $\xi_{n,k} \overset{\text{def.}}{=} \frac{s_1(\mathbf{Z}_{\le k}^\top \mathbf{Z}_{\le k})}{n}$, $\zeta_{n,k} \overset{\text{def.}}{=} \frac{s_1(\mathbf{Z}_{\le k}^\top \mathbf{Z}_{\le k})}{s_k(\mathbf{Z}_{\le k}^\top \mathbf{Z}_{\le k})}$, $\rho_{n,k} \overset{\text{def.}}{=} \frac{n \|\boldsymbol{\Sigma}_{>k}\|_{\text{op}} + s_1(\mathbf{A}_k)}{s_n(\mathbf{A}_k)}$, $r_k \overset{\text{def.}}{=} \frac{\operatorname{Tr}[\boldsymbol{\Sigma}_{>k}]}{\|\boldsymbol{\Sigma}_{>k}\|_{\text{op}}}$, $R_k \overset{\text{def.}}{=} \frac{\operatorname{Tr}[\boldsymbol{\Sigma}_{>k}]^2}{\operatorname{Tr}[\boldsymbol{\Sigma}_{>k}^2]}$. $\qquad\square$

## C.3  Sanity check

Recall that $\mathcal{B} \overset{\text{def.}}{=} \|\hat{\boldsymbol{\theta}}(\mathbf{X}\boldsymbol{\theta}^*) - \boldsymbol{\theta}^*\|_{\boldsymbol{\Sigma}}^2$. Let $k \in \mathbb{N}$ be an integer. For any $\delta \in (0,1)$, with probability at least $1 - \delta$, we have

$$\mathcal{B} \le \underbrace{\left( \frac{1 + \rho^2 \zeta^2 \xi^{-1} + \rho}{\delta} \right)}_{\text{constant}} \overbrace{\|\boldsymbol{\theta}_{>k}^*\|_{\boldsymbol{\Sigma}_{>k}}^2}^{\text{target's tail}} + \underbrace{(\zeta^2 \xi^{-2} + \rho \zeta^2 \xi^{-1})}_{\text{constant}} \overbrace{\frac{s_1(\mathbf{A}_k)^2}{n^2}}^{\approx (\lambda_k + \lambda)^2} \underbrace{\|\boldsymbol{\theta}_{\le k}^*\|_{\boldsymbol{\Sigma}_{\le k}^{-1}}^2}_{\text{target's head}}. \tag{30}$$

We see that:

1. the regressor barely learns the tail of the target (the eigenfunctions with small eigenvalues), thus the first term $\|\boldsymbol{\theta}_{>k}^*\|_{\boldsymbol{\Sigma}_{>k}}^2$ in the upper bound is just a multiple of the $\boldsymbol{\Sigma}$-norm square (or $L_\mu^2$-norm square in KRR setting) of the target's tail;

2. the second term indicates that the regressor learns well if the ridge $\lambda$ is small, showing the trad-off between (noiseless) interpolation and regularization;

3. the term $\|\boldsymbol{\theta}_{\le k}^*\|_{\boldsymbol{\Sigma}_{\le k}^{-1}}^2$ corresponds to norm square of the target's head in the interpolation space $\mathcal{H}^2$. This term will explain the so-called saturation effect reported in [29] where the bias ceases to improve once the source coefficient $s$ in Assumption (SC) surpasses 2. Consequently, the bias term $\mathcal{B}$ bound in Table 1 is closely related to the constant $\min s, 2$.

[46, 7] also derived a non-asymptotic bound of $\mathcal{V}$. Recall that $\mathcal{V} \overset{\text{def.}}{=} \mathbb{E}_{\boldsymbol{\epsilon}} \left[ \|\hat{\boldsymbol{\theta}}(\boldsymbol{\epsilon})\|_{\boldsymbol{\Sigma}}^2 \right]$. Let $k \in \mathbb{N}$ be an integer. Then

$$\mathcal{V} \le \underbrace{\sigma^2}_{\text{noise level}} \rho^2 \left( \underbrace{\zeta^2 \xi^{-1}}_{\text{constant}} \frac{k}{n} + \underbrace{\frac{\text{Tr}[\mathbf{Z}_{>k} \boldsymbol{\Sigma}_{>k}^2 \mathbf{Z}_{>k}^\top]}{n \, \text{Tr}[\boldsymbol{\Sigma}_{>k}^2]} \frac{r_k(\boldsymbol{\Sigma})^2}{n R_k(\boldsymbol{\Sigma})}}_{=\mathcal{O}\left(\frac{k}{n}\right)} \right). \tag{31}$$

We see that:

1. the upper bound is proportional to noise level $\sigma^2 = \mathbb{E}\left[\epsilon^2\right] \ge 0$;

2. we will show later that for $n$ large enough, the second term is approximately $\frac{k}{n}$, hence the decay rate of $\mathcal{V}$ is at most $\mathcal{O}_{n,k}\left(n^{-1}\right)$.

The derivation of Eq (30) and (31), which we refer to as the *Master inequalities* in this paper, is purely algebraic, and the statement holds regardless of the assumptions on the eigen-decay and the features. One can optimally choose the integer $k \in \mathbb{N}$ to achieve a tight upper bound, as we will demonstrate later in this paper. Surprisingly, the Master inequalities alone are sufficient to establish tight bounds for most cases under various settings.

# D Over-parameterized regime

In this section, we provide the proof for the upper bounds presented in Table 1. For all the statements in this section, the big O notation $\tilde{\mathcal{O}}(\cdot)$ can be replaced by $\mathcal{O}(\cdot)$ if $a + 2 \neq 2r$.

A brief summary of the propositions can be found in Table 3.

| Ridge | | strong | | weak | |
|---|---|---|---|---|---|
| Feature | | (IF) | (GF) | (IF) | (GF) |
| Poly (PE) | $\mathcal{B}$ | Proposition D.2 | | Proposition D.6 | Proposition D.5+D.6 |
| | $\mathcal{V}$ | Proposition D.7 | | | Proposition D.8 |
| Exp (EE) | $\mathcal{B}$ | Proposition D.3 | | | Proposition D.5 |
| | $\mathcal{V}$ | Proposition D.9 | | | - |

Table 3: Brief summary of propositions derived in Section D used for proving the upper bounds on Table 1.

## D.1 Bias under strong ridge

**Proposition D.1** (Asymptotic upper bound of the bias term with polynomial decay). *Suppose:* $\lambda_k = \Theta_k\left(k^{-1-a}\right)$, $|\theta_k^*| = \mathcal{O}_k\left(k^{-r}\right)$ *and* $\lambda = \Theta_k\left(k^{-b}\right)$ *for some* $a, b, r > 0$. *Furthermore, suppose Assumption (GF) (or resp. (IF)) holds, and* $\delta, \xi, \zeta = \Theta_{n,k}(1)$ *with probability at least* $1 - \varphi$. *Then, for any* $t \in (0, 1]$, *with probability at least* $1 - \delta - \varphi$ *(or resp.* $1 - \delta - \varphi - e^{-n}$), *the bias is bounded by:*

$$\mathcal{B} = \tilde{\mathcal{O}}_n\left(\rho^3 n^{-C}\right)$$

*for some constant*

$$C \overset{\text{def.}}{=} \min\{t(2r + a), 2\min\{b, 1 + ta\} - t(2 + a - 2r)_+\} > 0 \tag{32}$$

*with with following lower bound:*

$$C \geq \begin{cases} \min\{t(2r + a) - 2(ta + 1 - b), 2(b - 1 + t)\} & , b < 1 + ta; \\ \min\{t(2r + a), 2t(1 + a)\} & , b \geq 1 + ta. \end{cases} \tag{33}$$

*Proof.* For simplicity, we only prove the statement for all $a, r$ except when $a + 2 = 2r$. In such case, the argument follows similarly with $\mathcal{O}(\cdot)$ replaced by $\tilde{\mathcal{O}}(\cdot)$.

For any $k, n \in \mathbb{N}$, pick $k = \tilde{\Theta}_{n,k}(n^t)$ for some $t \in (0, 1]$. Suppose the event $\delta, \xi, \zeta = \Theta_{n,k}(1)$ happens, then:

$\mathcal{B}$

$$= \mathcal{O}_{n,k}\left(\left(\left(\frac{1 + \rho^2\zeta^2\xi^{-1} + \rho}{\delta}\right)\|\boldsymbol{\theta}_{>k}^*\|_{\boldsymbol{\Sigma}_{>k}}^2 + (\rho^2\zeta^2\xi^{-2} + \rho^3\zeta^2\xi^{-1})\left(n^{-b} + \frac{c_2\,\mathrm{Tr}[\boldsymbol{\Sigma}_{>k}]}{n}\right)^2 \|\boldsymbol{\theta}_{\leq k}^*\|_{\boldsymbol{\Sigma}_{\leq k}^{-1}}^2\right)\right) \tag{34}$$

$$= \mathcal{O}_{n,k}\left(\rho^3\left(k^{-(2r+a)} + \left(n^{-b} + \frac{k^{-a}}{n}\right)^2 k^{(2+a-2r)_+}\right)\right) \tag{35}$$

$$= \mathcal{O}_{n,k}\left(\rho^3\left(n^{-t(2r+a)} + \left(n^{-b} + n^{-(1+at)}\right)^2 n^{t(2+a-2r)_+}\right)\right) \tag{36}$$

$$= \mathcal{O}_{n,k}\left(\rho^3\left(n^{-t(2r+a)} + n^{-2\min\{b, 1+ta\}} n^{t(2+a-2r)_+}\right)\right)$$

$$= \mathcal{O}_{n,k}\left(\rho^3\left(n^{-t(2r+a)} + n^{-2\min\{b, 1+ta\}+t(2+a-2r)_+}\right)\right)$$

$$= \mathcal{O}_{n,k}\left(\rho^3\left(n^{-\min\{t(2r+a), 2\min\{b, 1+ta\}-t(2+a-2r)_+\}}\right)\right)$$

where we apply Proposition C.6, Lemma G.8 in line (34); we apply Lemma H.10: $\left\|\boldsymbol{\theta}_{\leq k}^*\right\|_{\boldsymbol{\Sigma}_{\leq k}^{-1}}^2 = \Theta_k\left(k^{(2+a-2r)_+}\right)$ where $x_+ \overset{\text{def.}}{=} \max\{x, 0\}$, as well as plug in $\mathrm{Tr}[\boldsymbol{\Sigma}_{>k}] = \Theta_k\left(k^{-a}\right)$ in line (35); we plug in $k = \tilde{\Theta}_{n,k}(n^t)$ for some $t \in (0, 1]$ in line (36).

Hence the bias $\mathcal{B}$ has a decay in form of $\mathcal{O}_n\left(\rho^3 n^{-C}\right)$ where

$$C \stackrel{\text{def.}}{=} \min\{t(2r+a), 2\min\{b, 1+ta\} - t(2+a-2r)_+\} \geq 0$$

is a constant defined in nested minima and maxima. To bound $C$ from below, we make two case distinctions: $b < 1 + ta$ or $b \geq 1 + ta$.

For $b < 1 + ta$, we have

$$
\begin{aligned}
C &= \min\{t(2r+a), 2b - t(2+a-2r)_+\} \\
&= \begin{cases} \min\{t(2r+a), 2b - t(2+a-2r)\} & , \; 2+a \geq 2r \\ \min\{t(2r+a), 2b\} & , \; 2+a < 2r \end{cases} \\
&= \begin{cases} \min\{t(2r+a), t(2r+a) + 2\overbrace{(1-t)}^{\geq 0} - 2(ta+1-b)\} & , \; 2+a \geq 2r \\ \min\{t(2r+a), 2b\} & , \; 2+a < 2r \end{cases} \\
&\geq \begin{cases} \min\{t(2r+a), t(2r+a) - 2\overbrace{(ta+1-b)}^{\geq 0})\} & , \; 2+a \geq 2r \\ \min\{t(2+a+a), 2b\} & , \; 2+a < 2r \end{cases} \\
&= \begin{cases} t(2r+a) - 2(ta+1-b) & , \; 2+a \geq 2r \\ 2\min\{ta+t, b\} & , \; 2+a < 2r \end{cases} \\
&\geq \begin{cases} t(2r+a) - 2(ta+1-b) & , \; 2+a \geq 2r \\ 2\min\{b-1+t, b\} & , \; 2+a < 2r \end{cases} \\
&= \begin{cases} t(2r+a) - 2(ta+1-b) & , \; 2+a \geq 2r \\ 2(b-1+t) & , \; 2+a < 2r \end{cases} \\
&\geq \min\{t(2r+a) - 2(ta+1-b), 2(b-1+t)\}.
\end{aligned}
$$

For $b \geq 1 + ta$, we have

$$
\begin{aligned}
C &= \min\{t(2r+a), 2(1+ta) - t(2+a-2r)_+\} \\
&= \begin{cases} \min\{t(2r+a), 2(1+ta) - t(2+a-2r)\} & , \; 2+a \geq 2r \\ \min\{t(2r+a), 2(1+ta)\} & , \; 2+a < 2r \end{cases} \\
&\geq \begin{cases} \min\{t(2r+a), t(2r+a) + 2(1-t)\} & , \; 2+a \geq 2r \\ \min\{t(2+a+a), 2(1+ta)\} & , \; 2+a < 2r \end{cases} \\
&= \begin{cases} t(2r+a) & , \; 2+a \geq 2r \\ \min\{2(t+ta), 2(1+ta)\} & , \; 2+a < 2r \end{cases} \\
&= \begin{cases} t(2r+a) & , \; 2+a \geq 2r \\ 2(t+ta) & , \; 2+a < 2r \end{cases} \\
&\geq \min\{t(2r+a), 2t(1+a)\}.
\end{aligned}
$$

Replacing $C$ by its lower bound, we obtain the claimed decay. $\qquad\square$

**Proposition D.2** (Asymptotic upper bound of bias under strong ridge and polynomial eigen-decay)**.** *Suppose:* $\lambda_k = \Theta_k\left(k^{-1-a}\right)$, $|\theta_k^*| = \mathcal{O}_k\left(k^{-r}\right)$ *and* $\lambda = \Theta_k\left(k^{-b}\right)$ *for some* $a, r > 0$ *and* $b \in (0, 1+a]$*. Fix a constant* $\delta \in (0, 1)$*. If, additionally, Assumption (GF) (or resp. (IF)) holds, then, with probability at least* $1 - \delta - o\left(n^{-1}\right)$ *(or resp.* $1 - \delta - 3e^{-c_1 n}$*), the bias is bounded by:*

$$\mathcal{B} = \tilde{\mathcal{O}}_n\left(n^{-\min\{sb, 2b\}}\right)$$

*with* $s \stackrel{\text{def.}}{=} \frac{2r+a}{1+a}$*.*

*Proof.* Use Proposition D.1 as the backbone, and use Propositions G.1 and G.2 to obtain the high probability guarantee for $\xi, \zeta = \Theta_{n,k}(1)$. Then for different value of $b$, we plug different values

of $t \in (0, 1]$ and set $k = \Theta_{n,k}(n^t)$, such that $\rho = \rho_{n,k} = \tilde{\mathcal{O}}_{n,k}(1)$ by Proposition G.3, in order to obtain different decay rates of bias.

Assume $b \in [0, a+1)$. Pick $k = \Theta_{k,n}(n^t)$ with $t = \frac{b}{1+a}$. As above, it holds that $\rho = \tilde{\Theta}_{n,k}(1)$ with high probability. This time, we plug in $t = \frac{b}{1+a}$ in the definition of $C$. By the choice of $b$, we have $b < 1 + ta$ and hence

$$
\begin{aligned}
C &\overset{\text{def.}}{=} \min\{t(2r + a), 2\min\{b, 1 + ta\} - t(2 + a - 2r)_+\} \\
&= \min\{t(2r + a), 2b - t(2 + a - 2r)_+\} \\
&= \begin{cases} \min\{t(2r + a), 2b - t(2 + a - 2r)\} &, 2 + a \geq 2r \\ \min\{t(2r + a), 2b\} &, 2 + a < 2r \end{cases} \\
&= \begin{cases} \min\{t(2r + a), t(2r + a) + 2\overbrace{(b - t(1 + a))}^{\geq 0}\} &, 2 + a \geq 2r \\ \min\{t(2r + a), 2b\} &, 2 + a < 2r \end{cases} \\
&\geq \begin{cases} \min\{t(2r + a), t(2r + a)\} &, 2 + a \geq 2r \\ \min\{t(2r + a), 2b\} &, 2 + a < 2r \end{cases} \\
&\geq \min\{t(2r + a), 2b\}.
\end{aligned}
$$

Finally, assume $b = a + 1$. Pick $k = \Theta_{k,n}(n^t)$ with $t = \frac{b}{1+a} = 1$. But this time, since $b \geq 1 + ta$, we plug in $t = \frac{b}{1+a} = 1$ into the second lower bound of $C$ in line (33):

$$
C \geq \min\{t(2r + a), 2t(1 + a)\} = \min\left\{\frac{b(2r + a)}{1 + a}, \frac{2b(1 + a)}{1 + a}\right\} = \min\{sb, 2b\}. \tag{37}
$$

$\square$

The argument for exponential decay follows analogously as above. Recall Assumption (EE): $\lambda_k = \Theta_k(e^{-ak})$ for some $a > 0$. Suppose $|\theta_k^*| = \Theta_k(e^{-rk})$ and $\lambda = \Theta_n\left(\frac{1}{n}e^{-bn}\right)$ [12] for some $b, r > 0$. Then the asymptotic bounds follows similarly to the polynomial case. For sake of completeness, in this section, we list out some propositions for illustration.

**Proposition D.3** (Asymptotic upper bound of bias under strong ridge and exponential eigen-decay). *Suppose $\lambda_k = \Theta_k(e^{-ak})$, $|\theta_k^*| = \Theta_k(e^{-rk})$ and $\lambda = \Theta_n(e^{-bn})$ for some $a, r > 0$ and $b \in (0, a)$. If Assumption (GF) (or resp. (IF)) holds, then with probability at least $1 - \delta - o_n\left(\frac{1}{n}\right)$ (or resp. $1 - \delta - 3e^{-c_1 n}$), the bias is bounded by:*

$$
\mathcal{B} = \begin{cases} \mathcal{O}_{n,k}\left(e^{-\min\{s,2\}bn}\right) &, a \neq 2r \\ \mathcal{O}_{n,k}\left(e^{-sbn} + ne^{-2bn}\right) &, a = 2r. \end{cases} \tag{38}
$$

*for some constants $c_1 > 0$, where $s = \frac{2r}{a} + 1 > 0$ is the source coefficient.*

*Remark* D.4. Note that the factor $n$ in the case of $a = 2r$ does not play a big role asymptotically: by taking a slightly smaller $\tilde{b} < b$, the decay is exponential in $\tilde{b}n$.

*Proof.* By Proposition C.6, with probability at least $1 - \delta$ (or resp. $1 - \delta - e^{-n}$), the bias is bounded by:

$$
\mathcal{B} \leq \left(\frac{1 + \rho^2 \zeta^2 \xi^{-1} + \rho}{\delta}\right) \|\boldsymbol{\theta}_{>k}^*\|_{\boldsymbol{\Sigma}_{>k}}^2 + c(\zeta^2 \xi^{-2} + \rho \zeta^2 \xi^{-1})\left(\frac{s_1(\mathbf{A}_k)}{n}\right)^2 \|\boldsymbol{\theta}_{\leq k}^*\|_{\boldsymbol{\Sigma}_{\leq k}^{-1}}^2. \tag{39}
$$

By Proposition G.1 and G.2, there exists some $c > 1$ such that, if $c\beta_k k \log k \leq n$ (or resp. $k/c \leq n$), then with probability at least $1 - 2\exp\left(-\frac{c}{\beta_k}\frac{n}{k}\right)$ (or resp. $1 - 2\exp(-c_1 n)$), it holds that

$$
\xi_{k,n} \geq \frac{1}{2}; \ \zeta_{k,n} \leq c_2.
$$

---

[12]The extra $\frac{1}{n}$ factor serves only for simplification purpose and has no effect on the choice of $b$.

Fix a $\delta \in (0,1)$. With probability at least $1 - \delta - 2\exp\left(-\frac{c}{\beta_k}\frac{n}{k}\right)$ (or resp. $1 - \delta - e^{-n} - 2e^{-c_1 n}$), it holds that:

$$\mathcal{B} = \mathcal{O}_{n,k}\left(\rho\left(\|\boldsymbol{\theta}^*_{>k}\|^2_{\boldsymbol{\Sigma}_{>k}} + \left(\frac{s_1(\mathbf{A}_k)}{n}\right)^2 \|\boldsymbol{\theta}^*_{\leq k}\|^2_{\boldsymbol{\Sigma}_{\leq k}^{-1}}\right)\right)$$

$$= \mathcal{O}_{n,k}\left(\rho\left(k^{-(2r+a)} + n^{-2}\left(e^{-bn}\right)^2\|\boldsymbol{\theta}^*_{\leq k}\|^2_{\boldsymbol{\Sigma}_{\leq k}^{-1}}\right)\right).$$

where the last inequality comes from Proposition G.4. For $b < a$, set $t \overset{\text{def.}}{=} \frac{b}{a} < 1$. By Proposition G.4 again, we have $\rho = \Theta_{k,n}(1)$ with probability at least $1 - o_n\left(\frac{1}{n}\right)$ (or resp. $1 - e^{-n}$). Set $s \overset{\text{def.}}{=} \frac{2r+a}{a}$ be the source coefficient and we write:

$$\mathcal{B} = \mathcal{O}_{n,k}\left(e^{-\frac{b}{a}(2r+a)n} + \left(e^{-bn}\right)^2\sum_{l=1}^{k}e^{(a-2r)l}\right)$$

$$= \mathcal{O}_{n,k}\left(e^{-sbn} + e^{-2bn}\sum_{l=1}^{\frac{b}{a}n}e^{(a-2r)l}\right)$$

$$= \begin{cases} \mathcal{O}_{n,k}\left(e^{-sbn} + e^{-2bn}\cdot e^{(a-2r)\frac{b}{a}n}\right) & , \; a > 2r \\ \mathcal{O}_{n,k}\left(e^{-sbn} + e^{-2bn}\cdot\frac{b}{a}n\right) & , \; a = 2r \\ \mathcal{O}_{n,k}\left(e^{-sbn} + e^{-2bn}\right) & , \; a < 2r \end{cases}$$

$$= \begin{cases} \mathcal{O}_{n,k}\left(e^{-sbn} + e^{-sbn}\right) & , \; a > 2r \\ \mathcal{O}_{n,k}\left(e^{-sbn} + ne^{-2bn}\right) & , \; a = 2r \\ \mathcal{O}_{n,k}\left(e^{-sbn} + e^{-2bn}\right) & , \; a < 2r \end{cases}$$

$$= \begin{cases} \mathcal{O}_{n,k}\left(e^{-\min\{s,2\}bn}\right) & , \; a \neq 2r \\ \mathcal{O}_{n,k}\left(e^{-sbn} + ne^{-2bn}\right) & , \; a = 2r. \end{cases}$$

$\square$

## D.2 Bias under weak ridge

**Proposition D.5** (Asymptotic upper bound of bias under weak/effectively no ridge for $s > 1$)**.** *Suppose Assumption (GF) (or resp. (IF)) holds, and the source coefficient $s > 1$. Then with high probability, the bias has decay:*

$$\mathcal{B} = \tilde{\mathcal{O}}_n\left(\lambda_n^{\min\{s,2\}}\right), \;\; for \; \lambda = \Omega_n\left(\lambda_n\right).$$

*Proof.* By definition, rewrite the bias into:

$$\mathcal{B} = \left\|\boldsymbol{\theta}^* - \hat{\boldsymbol{\theta}}(\mathbf{X}\boldsymbol{\theta}^*)\right\|^2_{\boldsymbol{\Sigma}}$$

$$= \left\|\boldsymbol{\theta}^* - \mathbf{X}^\top(\mathbf{X}\mathbf{X}^\top + n\lambda\mathbf{I}_n)^{-1}(\mathbf{X}\boldsymbol{\theta}^*)\right\|^2_{\boldsymbol{\Sigma}}$$

$$= \left\|\left(\mathbf{I}_p - \underbrace{\mathbf{X}^\top(\mathbf{X}\mathbf{X}^\top + n\lambda\mathbf{I}_n)^{-1}\mathbf{X}}_{\mathbf{P}_\lambda}\right)\boldsymbol{\theta}^*\right\|^2_{\boldsymbol{\Sigma}}. \tag{40}$$

Denote $\mathbf{P}_\lambda \overset{\text{def.}}{=} \mathbf{X}^\top(\mathbf{X}\mathbf{X}^\top + n\lambda\mathbf{I}_n)^{-1}\mathbf{X} \in \mathbb{R}^{p\times p}$. By Sherman-Morrison-Woodbury formula,

$$\mathbf{I}_p - \mathbf{P}_\lambda = \mathbf{I}_p - (n\lambda)^{-1}\mathbf{X}^\top\left(\mathbf{I}_n + (n\lambda)^{-1}\mathbf{X}\mathbf{X}^\top\right)^{-1}\mathbf{X} = \lambda\left(\lambda I_p + \frac{1}{n}\mathbf{X}^\top\mathbf{X}\right)^{-1} = \lambda\left(\lambda I_p + \hat{\boldsymbol{\Sigma}}\right)^{-1}$$

is a positive definite matrix such that the matrix $\mathbf{I}_p - \mathbf{P}_\lambda$ is monotonic increasing in $\lambda$, that is:

$$0 \preccurlyeq \mathbf{I}_p - \mathbf{P}_\lambda \preccurlyeq \mathbf{I}_p - \mathbf{P}_{\tilde{\lambda}} \preccurlyeq \mathbf{I}_p, \tag{41}$$

for all $\tilde{\lambda} \geq \lambda > 0$, since the map $\lambda \mapsto \frac{\lambda}{\lambda+x}$ is monotone. Hence, for $t \in [1, \min\{2, s\})$ [13],

$$\mathcal{B} = \|(\mathbf{I}_p - \mathbf{P}_\lambda)\boldsymbol{\theta}^*\|_{\boldsymbol{\Sigma}}^2 \tag{42}$$

$$= \left\|\boldsymbol{\Sigma}^{1/2}(\mathbf{I}_p - \mathbf{P}_\lambda)\boldsymbol{\theta}^*\right\|_2^2$$

$$\leq \left\|\boldsymbol{\Sigma}^{1/2}(\mathbf{I}_p - \mathbf{P}_\lambda)\boldsymbol{\Sigma}^{(t-1)/2}\right\|_{\mathrm{op}}^2 \cdot \left\|\boldsymbol{\Sigma}^{-(t-1)/2}\boldsymbol{\theta}^*\right\|_2^2$$

$$= \left\|\boldsymbol{\Sigma}^{1/2}(\mathbf{I}_p - \mathbf{P}_\lambda)^{1/2}(\mathbf{I}_p - \mathbf{P}_\lambda)^{(2-t)/2}(\mathbf{I}_p - \mathbf{P}_\lambda)^{(t-1)/2}\boldsymbol{\Sigma}^{(t-1)/2}\right\|_{\mathrm{op}}^2 \cdot \|\boldsymbol{\theta}^*\|_{\boldsymbol{\Sigma}^{1-t}}^2$$

$$\leq \left\|\boldsymbol{\Sigma}^{1/2}(\mathbf{I}_p - \mathbf{P}_\lambda)^{1/2}\right\|_{\mathrm{op}}^2 \cdot \left\|(\mathbf{I}_p - \mathbf{P}_\lambda)^{(2-t)/2}\right\|_{\mathrm{op}}^2 \cdot \left\|(\mathbf{I}_p - \mathbf{P}_\lambda)^{(t-1)/2}\boldsymbol{\Sigma}^{(t-1)/2}\right\|_{\mathrm{op}}^2 \cdot \|\boldsymbol{\theta}^*\|_{\boldsymbol{\Sigma}^{1-t}}^2$$

$$\leq \left\|\boldsymbol{\Sigma}^{1/2}(\mathbf{I}_p - \mathbf{P}_\lambda)^{1/2}\right\|_{\mathrm{op}}^2 \cdot \left\|\mathbf{I}_p^{(2-t)/2}\right\|_{\mathrm{op}}^2 \cdot \left\|(\mathbf{I}_p - \mathbf{P}_\lambda)^{(t-1)/2}\boldsymbol{\Sigma}^{(t-1)/2}\right\|_{\mathrm{op}}^2 \cdot \|\boldsymbol{\theta}^*\|_{\boldsymbol{\Sigma}^{1-t}}^2 \tag{43}$$

$$\leq \left\|\boldsymbol{\Sigma}^{1/2}(\mathbf{I}_p - \mathbf{P}_\lambda)^{1/2}\right\|_{\mathrm{op}}^2 \cdot \left\|\boldsymbol{\Sigma}^{(t-1)/2}(\mathbf{I}_p - \mathbf{P}_\lambda)^{(t-1)/2}\right\|_{\mathrm{op}}^2 \cdot \|\boldsymbol{\theta}^*\|_{\boldsymbol{\Sigma}^{1-t}}^2 \tag{44}$$

$$\leq \left\|\boldsymbol{\Sigma}^{1/2}(\mathbf{I}_p - \mathbf{P}_\lambda)^{1/2}\right\|_{\mathrm{op}}^2 \cdot \left\|\boldsymbol{\Sigma}^{1/2}(\mathbf{I}_p - \mathbf{P}_\lambda)^{1/2}\right\|_{\mathrm{op}}^{2(t-1)} \cdot \|\boldsymbol{\theta}^*\|_{\boldsymbol{\Sigma}^{1-t}}^2 \tag{45}$$

$$= \left\|\boldsymbol{\Sigma}^{1/2}(\mathbf{I}_p - \mathbf{P}_\lambda)^{1/2}\right\|_{\mathrm{op}}^{2t} \cdot \|\boldsymbol{\theta}^*\|_{\boldsymbol{\Sigma}^{1-t}}^2$$

$$= \left\|\boldsymbol{\Sigma}^{1/2}(\mathbf{I}_p - \mathbf{P}_\lambda)\boldsymbol{\Sigma}^{1/2}\right\|_{\mathrm{op}}^t \cdot \|\boldsymbol{\theta}^*\|_{\boldsymbol{\Sigma}^{1-t}}^2 \tag{46}$$

$$\leq \left\|\boldsymbol{\Sigma}^{1/2}(\mathbf{I}_p - \mathbf{P}_{\tilde{\lambda}})\boldsymbol{\Sigma}^{1/2}\right\|_{\mathrm{op}}^t \cdot \|\boldsymbol{\theta}^*\|_{\boldsymbol{\Sigma}^{1-t}}^2, \tag{47}$$

where we rewrite line (40) in line (42); we apply the monotonicity from line (41) in line (43); we use the fact that $\|\mathbf{M}_1\mathbf{M}_2\|_{\mathrm{op}} = \|\mathbf{M}_2\mathbf{M}_1\|_{\mathrm{op}}$ for symmetric matrices $\mathbf{M}_1, \mathbf{M}_2$ in line (44); we use Lemma H.17 to pull out the power $(t-1)$ from the matrix to its operator norm in line (45); we use the fact that $\left\|\mathbf{M}_1^{1/2}\mathbf{M}_2^{1/2}\right\|_{\mathrm{op}}^2 = \left\|\mathbf{M}_1^{1/2}\mathbf{M}_2^{1/2}(\mathbf{M}_1^{1/2}\mathbf{M}_2^{1/2})^\top\right\|_{\mathrm{op}} = \left\|\mathbf{M}_1^{1/2}\mathbf{M}_2\mathbf{M}_1^{1/2}\right\|_{\mathrm{op}}$ for any positive definite symmetric matrices [14] $\mathbf{M}_1, \mathbf{M}_2$ in line (46); by the monotonicity from line (41) again and the fact that $\boldsymbol{\Sigma}^{1/2}$ is positive definite, we choose $\tilde{\lambda} \geq \lambda \succcurlyeq \lambda_n$ to be strong in the sense of polynomial/exponential eigen-decay in line (47): $\tilde{\lambda} = \Theta_n\left(n^{-1-a}\right)$ for polynomial case and $\tilde{\lambda} = \Theta_n\left(e^{-an}\right)$ for exponential case.

Let $\mathbf{v} \in \mathbb{S}^{p-1} \subset \mathbb{R}^p$ be a unit vector such that $\left\|\boldsymbol{\Sigma}^{1/2}(\mathbf{I}_p - \mathbf{P}_{\tilde{\lambda}})\boldsymbol{\Sigma}^{1/2}\right\|_{\mathrm{op}}^2 = \left\|\boldsymbol{\Sigma}^{1/2}(\mathbf{I}_p - \mathbf{P}_{\tilde{\lambda}})\boldsymbol{\Sigma}^{1/2}\mathbf{v}\right\|_2^2$. By definition, we have

$$\left\|\boldsymbol{\Sigma}^{1/2}\mathbf{v}\right\|_{\boldsymbol{\Sigma}^{-1}}^2 = \|\mathbf{v}\|_2^2 = 1 < \infty,$$

hence the vector $\boldsymbol{\Sigma}^{1/2}\mathbf{v}$ lies on the interpolation space $\mathcal{H}^2$. Note that the expression

$$\left\|\boldsymbol{\Sigma}^{1/2}(\mathbf{I}_p - \mathbf{P}_{\tilde{\lambda}})\boldsymbol{\Sigma}^{1/2}\right\|_{\mathrm{op}}^2 = \left\|\boldsymbol{\Sigma}^{1/2}(\mathbf{I}_p - \mathbf{P}_{\tilde{\lambda}})\boldsymbol{\Sigma}^{1/2}\mathbf{v}\right\|_2^2 = \left\|(\mathbf{I}_p - \mathbf{P}_{\tilde{\lambda}})\left(\boldsymbol{\Sigma}^{1/2}\mathbf{v}\right)\right\|_{\boldsymbol{\Sigma}}^2$$

is just the bias term of another task $\boldsymbol{\Sigma}^{1/2}\mathbf{v}$ with source coefficient 2 and ridge $\tilde{\lambda} = \Omega_n\left(\lambda_n\right)$ on the same dataset $\mathbf{X}$. Apply Proposition D.2 with polynomial decay and Proposition D.3 with exponential decay on that new task with strong ridge $\tilde{\lambda}$, with high probability, the above term has a decay rate:

$$\left\|\boldsymbol{\Sigma}^{1/2}(\mathbf{I}_p - \mathbf{P}_{\tilde{\lambda}})\boldsymbol{\Sigma}^{1/2}\right\|_{\mathrm{op}}^2 = \left\|(\mathbf{I}_p - \mathbf{P}_{\tilde{\lambda}})\left(\boldsymbol{\Sigma}^{1/2}\mathbf{v}\right)\right\|_{\boldsymbol{\Sigma}}^2 = \tilde{\mathcal{O}}_n\left(\lambda_n^2\right). \tag{48}$$

---

[13] Recall that $s \stackrel{\text{def.}}{=} \begin{cases} \frac{2r+a}{1+a} & \text{, Assumption (PE) holds} \\ \frac{2r}{a} + 1 & \text{, Assumption (EE) holds} \end{cases}$ is the source coefficient.

[14] Same as above.

Plug in line (48) into line (47) to obtain the decay rate of the original bias: with high probability:

$$\mathcal{B} = \mathcal{O}_n \left( \left( \left\| \mathbf{\Sigma}^{1/2}(\mathbf{I}_p - \mathbf{P}_{\tilde{\lambda}})\mathbf{\Sigma}^{1/2} \right\|_{\mathrm{op}}^2 \right)^{t/2} \right) = \tilde{\mathcal{O}}_n \left( (\lambda_n^2)^{t/2} \right) = \tilde{\mathcal{O}}_n \left( \lambda_n^t \right).$$

This holds for any $t \in [1, \min\{s, 2\})$, hence we conclude that

$$\mathcal{B} = \tilde{\mathcal{O}}_n \left( \lambda_n^{\min\{s, 2\}} \right).$$

$\square$

For polynomial eigen-decay, we can still apply the Master inequality for $\mathcal{B}$ to obtain another upper bound:

**Proposition D.6** (Asymptotic upper bound of bias under weak/no ridge for any $s > 0$). *Suppose:* $\lambda_k = \Theta_k \left( k^{-1-a} \right)$, $|\theta_k^*| = \mathcal{O}_k \left( k^{-r} \right)$ *and* $\lambda = \Theta_k \left( k^{-b} \right)$ *for some* $a, b, r > 0$. *Fix a constant* $\delta \in (0, 1)$. *If, additionally, Assumption (GF) (or resp. (IF)) holds, then, with probability at least* $1 - \delta - \mathcal{O}_n \left( \frac{1}{\log n} \right)$ *(or resp.* $1 - \delta - 3e^{-c_1 n}$), *the bias is bounded by:*

$$\mathcal{B} = \tilde{\mathcal{O}}_n \left( n^{-C} \right)$$

*where*

$$C = (\min\{2(r - a), 2 - a\})_+ , \ b \in [1 + a, \infty] \tag{49}$$

*(or resp.*

$$C = \min\{2r + a, 2(1 + a)\}, \ b \in [1 + a, \infty].) \tag{50}$$

*Proof.* The proof is similar to Proposition D.2. Take $k = \Theta_{k,n} \left( \frac{n}{\log n} \right)$ (or resp. $k = n/c$ for some constant $c > 1$) and control $\xi, \zeta$ by Propositions G.1 and G.2. Now, by Proposition G.3, with probability at least $1 - \frac{c_1}{\log n}$, since $b \geq 1 + a$,

$$\rho = \tilde{\mathcal{O}}_{n,k} \left( n^a \right).$$

(or resp. $\rho = \mathcal{O}_{n,k} (1)$.) Now by Proposition D.1, since $b \in [1 + a, \infty]$ and we choose $t = 1$, with high probability,

$$\mathcal{B} = \tilde{\mathcal{O}}_{n,k} \left( \rho^3 n^{-\min\{t(2r+a), 2(1+at)\}} \right). \tag{51}$$

Plug in $\rho = \tilde{\mathcal{O}}_{n,k} \left( n^a \right)$ and $t = 1$ to obtain:

$$\mathcal{B} = \tilde{\mathcal{O}}_{n,k} \left( n^{-\min\{2(r-a), 2-a\}} \right). \tag{52}$$

The index in the above bound can be negative, causing the upper bound vacuous. By line (41), since $\mathcal{B} = \|(\mathbf{I}_p - \mathbf{P}_\lambda)\boldsymbol{\theta}^*\|_{\mathbf{\Sigma}}^2 \leq \|\boldsymbol{\theta}^*\|_{\mathbf{\Sigma}}^2$, there is a trivial bound that $\mathcal{B} = \mathcal{O}_n (1)$. Combining both results, we have

$$\mathcal{B} = \tilde{\mathcal{O}}_{n,k} \left( n^{-(\min\{2(r-a), 2-a\})_+} \right). \tag{53}$$

$\square$

### D.3 Variance with polynomial eigen-decay

**Proposition D.7** (Asymptotic upper bound of the variance term with strong ridge). *Suppose* $\lambda_k = \Theta_k \left( k^{-1-a} \right)$, $|\theta_k^*| = \Theta_k \left( k^{-r} \right)$, *and* $\lambda = \Theta_n \left( n^{-b} \right)$ *for some* $a, r > 0$ *and* $b \in (0, a + 1)$. *Furthermore, suppose Assumption (GF) (or resp. (IF)) holds. With probability at least* $1 - o_n \left( \frac{1}{n} \right)$ *(or resp.* $1 - 3e^{-c_1 n}$), *the variance term is bounded:*

$$\mathcal{V} = \mathcal{O}_n \left( n^{\frac{b}{(1+a)} - 1} \right)$$

*for* $n \in \mathbb{N}$ *large enough.*

*Proof.* We divide the case where Assumption (GF) or (IF) holds.

Suppose Assumption (GF) holds. By Proposition G.2, there exists constants $c > 1$, $c_1 > 0$ such that, for $k \le \frac{n}{c \log n}$, with probability at least $1 - 8e^{-c_1 n/k}$, it holds that

$$\xi_{n,k},\ \zeta_{n,k} = \Theta_{n,k}\,(1)\,.$$

Also, by definition, it always holds that

$$\frac{\mathrm{Tr}[\mathbf{X}_{>k}\boldsymbol{\Sigma}_{>k}\mathbf{X}_{>k}^\top]}{n\,\mathrm{Tr}[\boldsymbol{\Sigma}_{>k}^2]} = \frac{\frac{1}{n}\sum_{i=1}^n \left\|\boldsymbol{\Sigma}^{1/2}\mathbf{x}_i\right\|_2^2}{\mathrm{Tr}[\boldsymbol{\Sigma}_{>k}^2]} \le \sup_{\mathbf{x}} \frac{\left\|\boldsymbol{\Sigma}^{1/2}\mathbf{x}\right\|_2^2}{\mathrm{Tr}[\boldsymbol{\Sigma}_{>k}^2]} = \beta_k = \Theta_{n,k}\,(1)\,.$$

By Proposition C.7,

$$\mathcal{V}/\sigma^2 = \mathcal{O}_{n,k}\left(\rho^2\left(\frac{k}{n} + \frac{r_k(\boldsymbol{\Sigma})^2}{nR_k(\boldsymbol{\Sigma})}\right)\right)$$

Let $c > 1$ be the constant in Proposition G.3. Choose $k = \lfloor \frac{b}{1+a}n \rfloor \le \frac{n}{c \log n}$ for $n \in \mathbb{N}$ large enough. By Proposition G.3, with probability at least $1 - o_n\left(\frac{1}{n}\right)$,

$$\rho \overset{\text{def.}}{=} \frac{n\left\|\boldsymbol{\Sigma}_{>k}\right\|_{\mathrm{op}} + s_1(\mathbf{A}_k)}{s_n(\mathbf{A}_k)} = \mathcal{O}_{n,k}\left(\frac{n\lambda_k + n\lambda}{n\lambda}\right) = \mathcal{O}_{n,k}\,(1)$$

by the choice of $b \in (0, 1 + a)$. Combine the result with Lemma H.12, with probability at least $1 - o_n\left(\frac{1}{n}\right)$, we have

$$\mathcal{V} = \mathcal{O}_{n,k}\left(\frac{k}{n} + \frac{k^2}{kn}\right) = \mathcal{O}_{n,k}\left(\frac{n^{\frac{b}{1+a}}}{n}\right) = \mathcal{O}_{n,k}\left(n^{\frac{b}{1+a}-1}\right)$$

to conclude the claim.

Suppose Assumption (IF) holds. The argument follows analogously. As in Proposition G.2, choose $k = \lfloor \frac{b}{1+a} \rfloor < n/c$ for $n \in \mathbb{N}$ large enough. With probability at least $1 - 2e^{-c_1 n}$, we have

$$\xi_{n,k},\ \zeta_{n,k} = \Theta_{n,k}\,(1)\,.$$

By Proposition G.3, with probability at least $1 - e^{-n}$,

$$\rho = \Theta_{n,k}\,(1)$$

by the choice of $b \in (0, a + 1)$. By possibly choosing a new constant $c_1 > 0$, with probability at least $1 - 3e^{-c_1 n}$, we have $\xi, \zeta, \rho$ bounded. The rest of the argument follows similarly. $\qquad \square$

**Proposition D.8** (Asymptotic upper bound of variance term with weak/no ridge). *Suppose $\lambda_k = \Theta_k\left(k^{-1-a}\right)$, $|\theta_k^*| = \Theta_k\left(k^{-r}\right)$, and $\lambda = \Theta_n\left(n^{-b}\right)$ for some $a, r > 0$ and $b \in [a + 1, \infty]$. Furthermore, suppose Assumption (GF) (or resp. (IF)) holds. With probability at least $1 - o_n\left(\frac{1}{\log n}\right)$ (or resp. $1 - 3e^{-c_1 n}$), the variance term is bounded:*

$$\mathcal{V} = \mathcal{O}_n\left(n^{2a}\right)\,. \quad \text{(or resp. } \mathcal{V} = \mathcal{O}_n\,(1)\,.)$$

*Proof.* We divide the case where Assumption (GF) or (IF) holds.

Suppose Assumption (GF) holds. By Proposition G.2, there exists constants $c > 1$, $c_1 > 0$ such that, for $k \le \frac{n}{c \log n}$, with probability at least $1 - 8e^{-c_1 n/k}$, it holds that

$$\xi_{n,k},\ \zeta_{n,k} = \Theta_{n,k}\,(1)\,.$$

Choose $k = \lfloor \frac{n}{c \log n} \rfloor$. Then by Proposition G.3, with probability at least $1 - o_n\left(\frac{1}{\log n}\right)$, it holds that

$$\rho = \tilde{\mathcal{O}}_{n,k}\left(\frac{n^{-a} + n^{-a} + n^{-b+1}}{n^{-2a} + n^{-b+1}}\right) = \tilde{\mathcal{O}}_{n,k}\left(n^a\right)$$

by the choice of $b$. Moreover, since $k = \frac{n}{c \log n}$, by Proposition C.7, we have

$$\mathcal{V}/\sigma^2 = \mathcal{O}_{n,k}\left(\rho^2\left(\frac{k}{n} + \frac{r_k(\boldsymbol{\Sigma})^2}{nR_k(\boldsymbol{\Sigma})}\right)\right) = \tilde{\mathcal{O}}_{n,k}\left(n^{2a}\frac{k}{n}\right) = \tilde{\mathcal{O}}_{n,k}\left(n^{2a}\right)\,.$$

Suppose Assumption (IF) holds, choose $k = \lfloor n/c \rfloor$ for some constant $c > 1$ as in Propositions G.3. with probability at least $1 - 3e^{-c_1 n}$, it holds that

$$\xi, \; \zeta, \; \rho = \Theta_{n,k}(1).$$

Then by Proposition C.7, we have

$$\mathcal{V}/\sigma^2 = \mathcal{O}_{n,k}\left(\rho^2 \left(\frac{k}{n} + \frac{r_k(\boldsymbol{\Sigma})^2}{nR_k(\boldsymbol{\Sigma})}\right)\right) = \tilde{\mathcal{O}}_{n,k}\left(\frac{k}{n}\right) = \tilde{\mathcal{O}}_{n,k}(1).$$

$\square$

## D.4   Variance with exponential decay

**Proposition D.9** (Asymptotic upper bound of the variance term with strong ridge)**.** *Suppose* $\lambda_k = \Theta_k\left(e^{-ak}\right)$, $|\theta_k^*| = \Theta_k\left(e^{-rk}\right)$, *and* $\lambda = \Theta_n\left(\frac{1}{n}e^{-bn}\right)$ *for some* $a, r > 0$ *and* $b \in (0, a)$. *Furthermore, suppose Assumption* (GF) *(or resp.* (IF)*) holds. With probability at least* $1 - o_n\left(\frac{1}{n}\right)$ *(or resp.* $1 - 3e^{-c_1 n}$*), the variance term is bounded:*

$$\mathcal{V} = \mathcal{O}_n\left(n^{\frac{b}{a}-1}\right)$$

*for* $n \in \mathbb{N}$ *large enough.*

*Proof.* The argument is similar to Proposition D.7. The difference is the bound of $\rho$ and the second term in Proposition C.7: if $\xi, \zeta = \Theta_{n,k}(1)$, by Lemma H.12,

$$\mathcal{V}/\sigma^2 \leq \rho^2 \left(\zeta^2 \xi^{-1} \frac{k}{n} + \frac{\text{Tr}[\mathbf{Z}_{>k}\boldsymbol{\Sigma}_{>k}^2\mathbf{Z}_{>k}^\top]}{n\,\text{Tr}[\boldsymbol{\Sigma}_{>k}^2]} \frac{r_k(\boldsymbol{\Sigma})^2}{nR_k(\boldsymbol{\Sigma})}\right)$$

$$= \mathcal{O}_{n,k}\left(\rho^2 \left(\frac{k}{n} + \frac{1}{n^2}\right)\right)$$

$$= \mathcal{O}_{n,k}\left(\rho^2 \frac{k}{n}\right).$$

Choose $k = \lfloor \frac{b}{a} \rfloor < 1$. Combine Proposition G.4 and the above inequality to obtain the result.   $\square$

# E Matching lower bound

In this section, we provide the proof for the matching lower bounds presented on Table 1. A quick summary of propositions can be found in Table 4.

| Ridge | | strong | | weak | |
|---|---|---|---|---|---|
| Feature | | (IF) | (GF) | (IF) | (GF) |
| Poly (PE) | $\mathcal{B}$ | Proposition E.3 | - | Proposition E.3 | Proposition E.1 for $s \in [1,2]$ |
| | $\mathcal{V}$ | Proposition E.5 | - | Proposition E.5 | - |
| Exp (EE) | $\mathcal{B}$ | Proposition E.4 | - | Proposition E.1 for $s \in [1,2]$ | |
| | $\mathcal{V}$ | Proposition E.6 | - | - | |

Table 4: Quick summary of propositions used for proving the matching lower bounds on Table 1.

## E.1 Bias under weak ridge and source coefficient between 1 and 2

**Proposition E.1** (Asymptotic lower bound of bias, Proposition 4.4 in [31], Theorem 3.4 in [33]). *Suppose Assumption (PE)/(EE) holds. Then*

$$\sup_{\|\boldsymbol{\theta}^*\|_{\boldsymbol{\Sigma}^{1-s}} \leq 1} \mathcal{B} = \Omega_n\left(\lambda_n^s\right).$$

*where $s$ is the source coefficient.*

*Remark* E.2. Note that for both polynomial (with $2r \neq 2 + a$) and exponential decays (with $2r \neq a$), by Proposition D.5, we have $\mathcal{B} = \mathcal{O}\left(\lambda_n^{\max\{s,2\}}\right)$ for source coefficient $s \geq 1$. Hence when $s \in [1,2]$, the upper and lower bound matches.

*Proof.* Recall the expression of the bias in line (40):

$$\mathcal{B} = \left\|\hat{\boldsymbol{\theta}}(\mathbf{X}\boldsymbol{\theta}^*) - \boldsymbol{\theta}^*\right\|_{\boldsymbol{\Sigma}}^2 = \|(\mathbf{I}_p - \mathbf{P}_\lambda)\boldsymbol{\theta}^*\|_{\boldsymbol{\Sigma}}^2,$$

where $\mathbf{P} \overset{\text{def.}}{=} \mathbf{X}^\top(\mathbf{X}\mathbf{X}^\top + n\lambda\mathbf{I}_p)^{-1}\mathbf{X}$. Recall the source coefficient $s$ satisfies: $\|\boldsymbol{\theta}^*\|_{\boldsymbol{\Sigma}^{1-s}} < \infty$. By definition of operator norm, we have:

$$\sup_{\|\boldsymbol{\theta}^*\|_{\boldsymbol{\Sigma}^{1-s}} \leq 1} \mathcal{B} = \sup_{\|\boldsymbol{\theta}^*\|_{\boldsymbol{\Sigma}^{1-s}} \leq 1} \|(\mathbf{I}_p - \mathbf{P}_\lambda)\boldsymbol{\theta}^*\|_{\boldsymbol{\Sigma}}^2$$

$$= \sup_{\|\boldsymbol{\theta}^*\|_{\boldsymbol{\Sigma}^{1-s}} \leq 1} \left\|\boldsymbol{\Sigma}^{1/2}(\mathbf{I}_p - \mathbf{P}_\lambda)\boldsymbol{\Sigma}^{(s-1)/2}\boldsymbol{\Sigma}^{(1-s)/2}\boldsymbol{\theta}^*\right\|_2^2$$

$$= \left\|\boldsymbol{\Sigma}^{1/2}(\mathbf{I}_p - \mathbf{P}_\lambda)\boldsymbol{\Sigma}^{(s-1)/2}\right\|_{\text{op}}^2$$

$$= \left\|\boldsymbol{\Sigma}^{s/2} - \boldsymbol{\Sigma}^{1/2}\mathbf{P}_\lambda\boldsymbol{\Sigma}^{(s-1)/2}\right\|_{\text{op}}^2.$$

By [43], $\|\mathbf{M}_1 - \mathbf{M}_2\|_{\text{op}} \geq s_{n+1}(\mathbf{M}_1)$ for any operator $\mathbf{M}_2$ with rank at most $n$. Note that $\mathbf{P}_\lambda \in \mathbb{R}^{p \times p}$ is of rank $n$, hence

$$\sup_{\|\boldsymbol{\theta}^*\|_{\boldsymbol{\Sigma}^{1-s}} \leq 1} \mathcal{B} = \left\|\boldsymbol{\Sigma}^{s/2} - \boldsymbol{\Sigma}^{1/2}\mathbf{P}_\lambda\boldsymbol{\Sigma}^{(s-1)/2}\right\|_{\text{op}}^2 \geq \left(\lambda_{n+1}^{s/2}\right)^2 = \Omega_n\left(\lambda_n^s\right).$$

$\square$

## E.2 Independent features and prior signs

Instead of bounding the bias from below in the worst case scenario, we can also bound it in an average sense. In this case, we need to pose an assumption on the target coefficient $\boldsymbol{\theta}^*$.

**Assumption (PS)** (Prior Signs). *Assume target coefficient $\boldsymbol{\theta}^* \in \mathbb{R}^p$ is drawn from a distribution $\vartheta$ where its entries $\theta_k^*$'s are drawn independently of each other and have the same distributions as the random variables $-\theta_k^*$'s.*

**Proposition E.3** (Asymptotic lower bound of bias with polynomial decay and independent features). *Suppose Assumption (PS) holds and $\lambda_k = \Theta_k \left(k^{-1-a}\right)$, $\mathbb{E}_{\boldsymbol{\theta}^*}\left[(\theta_k^*)^2\right] = \mathcal{O}_k\left(k^{-2r}\right)$ and $\lambda = \Theta_n\left(n^{-b}\right)$ for some $a, b, r > 0$. Fix a constant $\delta \in (0, 1)$. If, additionally, Assumption (IF) holds, then, with probability at least $1 - \delta - 11e^{-c_1 n}$, the bias is bounded by:*

$$\mathbb{E}_{\boldsymbol{\theta}^*}[\mathcal{B}] = \tilde{\Omega}_n\left(n^{-b\min\{s,2\}}\right)$$

*for $s \overset{\text{def.}}{=} \frac{2r+a}{1+a}$.*

*Proof.* By Lemma H.14, it holds that:

$$\mathbb{E}_{\boldsymbol{\theta}^*}[\mathcal{B}] \geq \sum_{l=1}^{p} \frac{\lambda_l \mathbb{E}_{\boldsymbol{\theta}^*}\left[(\theta_l^*)^2\right]}{\left(1 + \lambda_l s_n(\mathbf{A}_{-l})^{-1} \left\|\mathbf{z}^{(l)}\right\|_2^2\right)^2}.$$

Hence, we want to bound the term $s_n(\mathbf{A}_{-l})^{-1}$ and $\left\|\mathbf{z}^{(l)}\right\|_2^2$ from above.

If Assumption (IF) holds, then apply Lemma G.7 on $\mathbf{A}_{-l}$ instead of $\mathbf{A}$, $l = 1, ..., p$, with probability at least $1 - 2e^{-n}$,

$$s_n(\mathbf{A}_{-l}) = \begin{cases} \Omega_n\left(n\lambda_n\right), & \text{if } l > n \\ \Omega_n\left(n\lambda_{n+1}\right), & \text{if } l \leq n \end{cases} = \Omega_n\left(n\lambda_n\right) = \Omega_n\left(\text{Tr}[\boldsymbol{\Sigma}_{>n}]\right).$$

By Lemma H.7, with probability at least $1 - 2e^{-n}$, we have,

$$\|\mathbf{z}_l\|_2^2 \leq c_1 n, \forall l = 1, ..., p.$$

Plug in the above two inequalities, by union bound, with probability at least $1 - 4e^{-n}$,

$$\frac{\lambda_l \mathbb{E}\left[(\theta_l^*)^2\right]}{(1 + \lambda_l s_n(\mathbf{A}_{-l})^{-1} \|\mathbf{z}_l\|_2^2)^2} = \Omega_n\left(\frac{\lambda_l \mathbb{E}\left[(\theta_l^*)^2\right]}{\left(1 + \frac{n\lambda_l}{\lambda_{n+1} r_n}\right)^2}\right)$$

where $r_k = \frac{\text{Tr}[\boldsymbol{\Sigma}_{>k}]}{\|\boldsymbol{\Sigma}_{>k}\|_{\text{op}}}$ for any $k$. If we choose $k = n/c$ for some constant $c > 1$ such that $\mathbf{A}_k = \Theta_n\left(n\lambda_n\right)$, then $\text{Tr}[\boldsymbol{\Sigma}_{>k}] = \Theta_n\left(\text{Tr}[\boldsymbol{\Sigma}_{>n}]\right)$. Hence

$$\frac{\lambda_l \mathbb{E}\left[(\theta_l^*)^2\right]}{(1 + \lambda_l s_n(\mathbf{A}_{-l})^{-1} \|\mathbf{z}_l\|_2^2)^2} = \Omega_n\left(\frac{\lambda_l \mathbb{E}\left[(\theta_l^*)^2\right]}{\left(1 + \frac{n\lambda_l}{\lambda_{k+1} r_k}\right)^2}\right)$$

By Lemma H.15, with probability at least $1 - 8e^{-n}$,

$$\mathbb{E}_{\boldsymbol{\theta}^*}[\mathcal{B}] = \Omega_n\left(\underbrace{\sum_{l=1}^{p} \frac{\lambda_l \mathbb{E}\left[(\theta_l^*)^2\right]}{\left(1 + \frac{n\lambda_l}{\lambda_{k+1} r_k}\right)^2}}_{\underline{B}}\right),$$

where $\underline{B}$ is defined as in Theorem H.16. since $r_k = \frac{\text{Tr}[\boldsymbol{\Sigma}_{>k}]}{\|\boldsymbol{\Sigma}_{>k}\|_{\text{op}}} = \Theta_k\left(\frac{k\lambda_k}{\lambda_k}\right) = \Theta_k(k)$, hence the fraction $\frac{r_k}{n} = \Theta_{n,k}(1)$ and $r_k$ satisfies the condition of Theorem H.16. By Theorem H.16, we have upper bound $\overline{\mathcal{B}}$ matching the lower bound $\underline{\mathcal{B}}$:

$$\underline{B} = \Theta_n\left(\underbrace{\|\boldsymbol{\theta}_{>k}^*\|_{\boldsymbol{\Sigma}_{>k}}^2 + \left(\frac{n\lambda + \text{Tr}[\boldsymbol{\Sigma}_{>k}]}{n}\right)^2 \|\boldsymbol{\theta}_{\leq k}^*\|_{\boldsymbol{\Sigma}_{\leq k}^{-1}}^2}_{\overline{B}}\right).$$

By Propositions C.6, D.2 and D.6, we have, with probability at least $1 - \delta - 3e^{c_1 n}$,

$$\mathcal{B} = \mathcal{O}_n\left(\overline{B}\right) = \mathcal{O}_n\left(n^{-b\min\{s,2\}}\right).$$

All together, with probability at least $1 - \delta - 3e^{-c_1 n} - 8e^{-n}$,

$$\mathbb{E}_{\boldsymbol{\theta}^*}\left[\mathcal{B}\right] = \Omega_n\left(\overline{B}\right) = \Omega_n\left(n^{-b\min\{s,2\}}\right).$$

$\square$

**Proposition E.4** (Asymptotic lower bound of bias with exponential decay and independent features under strong ridge)**.** *Suppose Assumption (PS) holds and $\lambda_k = \Theta_k\left(e^{-ak}\right)$, $\mathbb{E}_{\boldsymbol{\theta}^*}\left[(\theta_k^*)^2\right] = \mathcal{O}_k\left(e^{-2rk}\right)$ and $\lambda = \Theta_n\left(e^{-bn}\right)$ for some $a, r > 0$ and $b \in (0, a)$. Fix a constant $\delta \in (0, 1)$. If, additionally, Assumption (IF) holds, then, with probability at least $1 - \delta - 7e^{-c_1 n}$, the bias is bounded by:*

$$\mathbb{E}_{\boldsymbol{\theta}^*}\left[\mathcal{B}\right] = \tilde{\Omega}_n\left(e^{-bn\min\{s,2\}}\right)$$

*for $s \overset{\text{def.}}{=} \frac{2r}{a} + 1$.*

*Proof.* The proof is similar to Proposition E.3. By Lemma H.14, it holds that:

$$\mathbb{E}_{\boldsymbol{\theta}^*}\left[\mathcal{B}\right] \geq \sum_{l=1}^{p} \frac{\lambda_l \mathbb{E}_{\boldsymbol{\theta}^*}\left[(\theta_l^*)^2\right]}{\left(1 + \lambda_l s_n(\mathbf{A}_{-l})^{-1}\left\|\mathbf{z}^{(l)}\right\|_2^2\right)^2}.$$

By Lemma H.7, with probability at least $1 - 2e^{-n}$, we have,

$$\|\mathbf{z}_l\|_2^2 \leq c_1 n, \forall l = 1, ..., p.$$

By the choice of $b \in (0, a)$:

$$s_n(\mathbf{A}_{-l}) = \Omega_n\left(n\lambda\right) = \Omega_n\left(ne^{-bn}\right) = \Omega_n\left(n\lambda_{n+1}\right).$$

Plug in the above two inequalities, with probability at least $1 - 2e^{-n}$,

$$\frac{\lambda_l \mathbb{E}\left[(\theta_l^*)^2\right]}{(1 + \lambda_l s_n(\mathbf{A}_{-l})^{-1}\|\mathbf{z}_l\|_2^2)^2} = \Omega_n\left(\frac{\lambda_l \mathbb{E}\left[(\theta_l^*)^2\right]}{\left(1 + \frac{\lambda_l}{\lambda_{n+1}}\right)^2}\right) = \Omega_n\left(\frac{\lambda_l \mathbb{E}\left[(\theta_l^*)^2\right]}{\left(1 + \frac{n\lambda_l}{\lambda_{n+1}r_n}\right)^2}\right)$$

where $r_k = \frac{\text{Tr}[\boldsymbol{\Sigma}_{>k}]}{\|\boldsymbol{\Sigma}_{>k}\|_{\text{op}}} = \Theta_k\left(1\right)$. By Lemma H.15, with probability at least $1 - 4e^{-n}$,

$$\mathbb{E}_{\boldsymbol{\theta}^*}\left[\mathcal{B}\right] = \Omega_n\left(\underbrace{\sum_{l=1}^{p} \frac{\lambda_l \mathbb{E}\left[(\theta_l^*)^2\right]}{\left(1 + \frac{n\lambda_l}{\lambda_{k+1}r_k}\right)^2}}_{\underline{B}}\right),$$

where $\underline{B}$ is defined as in Theorem H.16. since $r_k = \frac{\text{Tr}[\boldsymbol{\Sigma}_{>k}]}{\|\boldsymbol{\Sigma}_{>k}\|_{\text{op}}} = \Theta_k\left(1\right)$, pick $k$ such that $r_k > 1$. This satisfies the condition of Theorem H.16. By Theorem H.16, we have upper bound $\overline{\mathcal{B}}$ matching the lower bound $\underline{\mathcal{B}}$:

$$\underline{B} = \Theta_n\left(\underbrace{\|\boldsymbol{\theta}_{>k}^*\|_{\boldsymbol{\Sigma}_{>k}}^2 + \left(\frac{n\lambda + \text{Tr}[\boldsymbol{\Sigma}_{>k}]}{n}\right)^2 \|\boldsymbol{\theta}_{\leq k}^*\|_{\boldsymbol{\Sigma}_{\leq k}^{-1}}^2}_{\overline{B}}\right).$$

By Propositions C.6, and D.3, we have, with probability at least $1 - \delta - 3e^{c_1 n}$,

$$\mathcal{B} = \mathcal{O}_n\left(\overline{B}\right) = \mathcal{O}_n\left(e^{-bn\min\{s,2\}}\right).$$

All together, with probability at least $1 - \delta - 3e^{-c_1 n} - 4e^{-n}$,

$$\mathbb{E}_{\boldsymbol{\theta}^*}\left[\mathcal{B}\right] = \Omega_n\left(\overline{B}\right) = \Omega_n\left(e^{-bn\min\{s,2\}}\right).$$

$\square$

**Lemma E.5** (Asymptotic lower bound of variance with polynomial eigen-decay and independent features). *Suppose $\lambda_k = \Theta_k\left(k^{-1-a}\right)$, $\lambda = \Theta_n\left(n^{-b}\right)$. Additionally, suppose Assumption (IF) holds. Then with probability at least $1 - ce^{-n/c}$, it holds that*

$$\mathcal{V} = \begin{cases} \Omega_n\left(n^{-1+\frac{b}{1+a}}\right) & , \ \lambda \text{ strong} \\ \Omega_n\left(1\right) & , \ \lambda \text{ weak} \end{cases}$$

*Proof.* This is a consequence of Propositions C.7, D.7 Lemma H.13 and Theorem H.16: with probability at least $1 - 3e^{-c_1 n}$, by choosing $k = \begin{cases} \lceil n^{\frac{b}{1+a}} \rceil & , \ \lambda \text{ strong} \\ n/c & , \ \lambda \text{ weak} \end{cases}$, we have

$$\mathcal{V} = \begin{cases} \mathcal{O}_n\left(n^{-1+\frac{b}{1+a}}\right) & , \ \lambda \text{ strong} \\ \mathcal{O}_n\left(1\right) & , \ \lambda \text{ weak} \end{cases} = \mathcal{O}_n\left(\overline{V}\right),$$

where $\overline{V} \overset{\text{def.}}{=} \frac{k}{n} + \frac{n\,\mathrm{Tr}[\boldsymbol{\Sigma}_{>k}^2]}{(n\lambda+\mathrm{Tr}[\boldsymbol{\Sigma}_{>k}])^2}$ is defined as in Theorem H.16. Since the polynomial eigen-decay $\lambda_k$ satisfies the condition in Theorem H.16, we have

$$\underline{V} = \Theta_{n,k}\left(\overline{V}\right).$$

By Lemma H.13, $\mathcal{V} = \begin{cases} \Omega_n\left(n^{-1+\frac{b}{1+a}}\right) & , \ \lambda \text{ strong} \\ \Omega_n\left(1\right) & , \ \lambda \text{ weak} \end{cases}$. By taking a larger constant $c > 0$, the above events hold with probability at least $1 - ce^{-n/c}$. $\square$

**Lemma E.6** (Asymptotic lower bound of variance with exponential eigen-decay and independent features). *Suppose $\lambda_k = \Theta_k\left(e^{-ak}\right)$, $\lambda = \Theta_n\left(e^{-bn}\right)$. Additionally, suppose Assumption (IF) holds. Then with probability at least $1 - ce^{-n/c}$, it holds that*

$$\mathcal{V} = \begin{cases} \mathcal{O}_n\left(n^{-1+\frac{b}{1+a}}\right) & , \ \lambda \text{ strong} \\ \mathcal{O}_n\left(1\right) & , \ \lambda \text{ weak} \end{cases}$$

*Proof.* The proof follows a similar pattern to that of Proposition E.5. Ensure that the exponential eigen-decay also meets the condition outlined in Theorem H.16. Consequently, $\underline{V} = \Theta_{n,k}\left(\overline{V}\right)$, and the remaining argument proceeds accordingly. $\square$

# F Under-parameterized regime

For the under-parameterized regime, one can clearly use the Master inequalities in Section C to bound the test error from above by setting $k = p < n$. However, one can even prove an asymptotic convergence in the following way.

we first prove the convergence of the empirical covariance matrix.

**Lemma F.1** (Convergence of $\hat{\Sigma}$). *Suppose Assumption (GF) (or resp. (IF)) holds. Fix an $\epsilon \in (0, 1)$. Then there exists a constant $c > 1$ such that, for $p < \frac{n^{1-\epsilon}}{c \log n}$ (or resp. $p < n^{1-\epsilon}/c$), with probability at least $1 - o_n(n^{-\epsilon})$, it holds that*

$$\left\| \mathbf{I}_p - \frac{1}{n} \mathbf{Z}^\top \mathbf{Z} \right\|_{op} = \left\| \Sigma - \hat{\Sigma} \right\|_{op} = o_{n,p}\left( n^{-\epsilon/2} \right),$$

*where $\Sigma \overset{\text{def.}}{=} \mathbb{E}\left[ \mathbf{x} \mathbf{x}^\top \right] \in \mathbb{R}^{p \times p}$ and $\hat{\Sigma} \overset{\text{def.}}{=} \frac{1}{n} \sum_{i=1}^n \mathbf{x}_i \mathbf{x}_i^\top \in \mathbb{R}^{p \times p}$.*

*Proof.* We argue similarly as in Proposition G.2. Fix an $\epsilon \in (0, 1)$.

Suppose Assumption (GF) holds. We have $\|\mathbf{z}\|_2^2 \leq \beta_p p$ by definition of $\beta_k$. Apply Theorem H.3 on the whitened input block $\mathbf{Z} \in \mathbb{R} n \times p$: there exists some constant $c_1 > 0$ such that, with probability at least $1 - 2pe^{-c_1 t^2}$, we have:

$$\sqrt{n} - t\sqrt{\beta_p p} \leq s_p(\mathbf{Z}) \leq s_1(\mathbf{Z}) \leq \sqrt{n} + t\sqrt{\beta_p p}.$$

By setting $t = \sqrt{\frac{\log n}{c_1}}$ and the choice of $p < \frac{n^{1-\epsilon}}{c \log n}$, with probability at least $1 - \frac{2}{cn^\epsilon \log n}$, we have

$$\sqrt{n}\left( 1 - \sqrt{\frac{\beta_p}{2c_1} n^{-\epsilon}} \right) \leq s_p(\mathbf{Z}) \leq s_1(\mathbf{Z}) \leq \sqrt{n}\left( 1 + \sqrt{\frac{\beta_p}{2c_1} n^{-\epsilon}} \right).$$

Hence, for $n$ large enough, with probability at least $1 - \frac{2}{cn^\epsilon \log n}$, it holds that:

$$\left\| \mathbf{I}_p - \frac{1}{n} \mathbf{Z}^\top \mathbf{Z} \right\|_{op} \leq \frac{2\beta_p}{2c_1} n^{-\epsilon} = o_n\left( n^{-\epsilon/2} \right).$$

and hence:

$$
\begin{aligned}
\left\| \Sigma - \hat{\Sigma} \right\|_{op} &\leq \left\| \Sigma^{1/2} \right\|_{op} \left\| \mathbf{I}_p - \frac{1}{n} \mathbf{Z}^\top \mathbf{Z} \right\|_{op} \left\| \Sigma^{1/2} \right\|_{op} \\
&\leq \left\| \Sigma^{1/2} \right\|_{op} \cdot \frac{2\beta_p}{2c_1} n^{-\epsilon} \cdot \left\| \Sigma^{1/2} \right\|_{op} \\
&= \frac{\lambda_1 \beta_p}{c_1} n^{-\epsilon} \\
&= o_n\left( n^{-\epsilon/2} \right).
\end{aligned}
$$

Suppose Assumption (GF) holds. By Lemma H.7 and union bound, we have $\|\mathbf{z}_i\|_2^2 \leq 2p$ for all $i = 1, ..., n$ with probability $1 - 2ne^{-c_1 n}$. When this event happens, apply Theorem H.3 on the whitened input block $\mathbf{Z} \in \mathbb{R}^{n \times p}$, the rest of the argument follows similarly. $\qquad\square$

**Proposition F.2** (Convergence of the bias and variance terms). *Suppose Assumption (GF) (or resp. (IF)) holds. Then there exists an $\epsilon \in (0, 1)$ and a constant $c > 1$ such that, for $p < \frac{n^{1-\epsilon}}{c \log n}$ (or resp. $p < n^{1-\epsilon}/c$), with probability at least $1 - o_n(n^{-\epsilon})$, it holds that*

$$\mathcal{B} = \Theta_{n,p}\left( \lambda^2 \sum_{k=1}^p \frac{\lambda_k (\theta_k^*)^2}{(\lambda_k + \lambda)^2} \right), \quad \mathcal{V} = \Theta_{n,p}\left( \frac{\sigma^2}{n} \sum_{k=1}^p \frac{\lambda_k^2}{(\lambda_k + \lambda)^2} \right).$$

*Proof.* We begin with $\mathcal{B}$. By line (40) and the formula $\mathbf{M}_1^{-1} - \mathbf{M}_2^{-1} = \mathbf{M}_1^{-1}(\mathbf{M}_1 - \mathbf{M}_2)\mathbf{M}_2^{-1}$, rewrite the bias term $\mathcal{B}$ as:

$$
\begin{aligned}
\mathcal{B} &= \|(\mathbf{I}_p - \mathbf{P}_\lambda)\boldsymbol{\theta}^*\|_{\boldsymbol{\Sigma}}^2 \\
&= \left\| \lambda(\lambda\mathbf{I}_p + \hat{\boldsymbol{\Sigma}})^{-1}\boldsymbol{\theta}^* \right\|_{\boldsymbol{\Sigma}}^2 \\
&= \left\| \lambda \left( (\lambda\mathbf{I}_p + \boldsymbol{\Sigma})^{-1} + (\lambda\mathbf{I}_p + \hat{\boldsymbol{\Sigma}})^{-1}(\hat{\boldsymbol{\Sigma}} - \boldsymbol{\Sigma})(\lambda\mathbf{I}_p + \boldsymbol{\Sigma})^{-1} \right) \boldsymbol{\theta}^* \right\|_{\boldsymbol{\Sigma}}^2.
\end{aligned}
$$

Denote $\boldsymbol{\Delta} = (\hat{\boldsymbol{\Sigma}} - \boldsymbol{\Sigma})(\lambda\mathbf{I}_p + \boldsymbol{\Sigma})^{-1}$. Note that

$$
\begin{aligned}
\|\boldsymbol{\Delta}\|_{\mathrm{op}} &= \left\| \boldsymbol{\Sigma}^{1/2}\left( \frac{1}{n}\mathbf{Z}^\top\mathbf{Z} - \mathbf{I}_p \right)\boldsymbol{\Sigma}^{1/2}\boldsymbol{\Sigma}^{-1/2}(\lambda\boldsymbol{\Sigma}^{-1} + \mathbf{I}_p)^{-1}\boldsymbol{\Sigma}^{-1/2} \right\|_{\mathrm{op}} \\
&\leq \left\| \frac{1}{n}\mathbf{Z}^\top\mathbf{Z} - \mathbf{I}_p \right\|_{\mathrm{op}} \cdot \left\| (\lambda\boldsymbol{\Sigma}^{-1} + \mathbf{I}_p)^{-1} \right\|_{\mathrm{op}} \cdot \left\| \boldsymbol{\Sigma}^{1/2}\boldsymbol{\Sigma}^{1/2}\boldsymbol{\Sigma}^{-1/2}\boldsymbol{\Sigma}^{-1/2} \right\|_{\mathrm{op}} \\
&= o_n\left( n^{-\epsilon/2} \right).
\end{aligned}
$$

We apply the matrix difference formula iteratively:

$$
\begin{aligned}
(\hat{\boldsymbol{\Sigma}} + \lambda\mathbf{I}_p)^{-1} &= (\boldsymbol{\Sigma} + \lambda\mathbf{I}_p)^{-1} + (\hat{\boldsymbol{\Sigma}} + \lambda\mathbf{I}_p)^{-1}(\hat{\boldsymbol{\Sigma}} - \boldsymbol{\Sigma})(\boldsymbol{\Sigma} + \lambda\mathbf{I}_p)^{-1} \\
&= (\boldsymbol{\Sigma} + \lambda\mathbf{I}_p)^{-1} + (\hat{\boldsymbol{\Sigma}} + \lambda\mathbf{I}_p)^{-1}\boldsymbol{\Delta} \\
&= (\boldsymbol{\Sigma} + \lambda\mathbf{I}_p)^{-1} + \left( (\boldsymbol{\Sigma} + \lambda\mathbf{I}_p)^{-1} + (\hat{\boldsymbol{\Sigma}} + \lambda\mathbf{I}_p)^{-1}\boldsymbol{\Delta} \right)\boldsymbol{\Delta} \\
&= ... \\
&= (\boldsymbol{\Sigma} + \lambda\mathbf{I}_p)^{-1} \sum_{t=0}^\infty \boldsymbol{\Delta}^t,
\end{aligned}
$$

we write:

$$
\begin{aligned}
\mathcal{B} &= \left\| \lambda(\hat{\boldsymbol{\Sigma}} + \lambda\mathbf{I}_p)^{-1}\boldsymbol{\theta}^* \right\|_{\boldsymbol{\Sigma}}^2 \\
&= \left\| \lambda\left( (\lambda\mathbf{I}_p + \boldsymbol{\Sigma})^{-1} + (\lambda\mathbf{I}_p + \hat{\boldsymbol{\Sigma}})^{-1}(\hat{\boldsymbol{\Sigma}} - \boldsymbol{\Sigma})(\lambda\mathbf{I}_p + \boldsymbol{\Sigma})^{-1} \right)\boldsymbol{\theta}^* \right\|_{\boldsymbol{\Sigma}}^2 \\
&= \left\| \lambda(\lambda\mathbf{I}_p + \boldsymbol{\Sigma})^{-1}\boldsymbol{\theta}^* + \lambda(\hat{\boldsymbol{\Sigma}} + \lambda\mathbf{I}_p)^{-1}\boldsymbol{\Delta}\boldsymbol{\theta}^* \right\|_{\boldsymbol{\Sigma}}^2 \\
&= \left\| \lambda(\lambda\mathbf{I}_p + \boldsymbol{\Sigma})^{-1}\left( \sum_{t=0}^\infty \boldsymbol{\Delta}^t \right)\boldsymbol{\theta}^* \right\|_{\boldsymbol{\Sigma}}^2 \\
&= \Theta_{n,p}\left( \left\| \lambda(\lambda\mathbf{I}_p + \boldsymbol{\Sigma})^{-1}\boldsymbol{\theta}^* \right\|_{\boldsymbol{\Sigma}}^2 \right).
\end{aligned}
$$

Finally, write

$$
\left\| \lambda(\lambda\mathbf{I}_p + \boldsymbol{\Sigma})^{-1}\boldsymbol{\theta}^* \right\|_{\boldsymbol{\Sigma}}^2 = \lambda^2 \sum_{k=1}^p \frac{\lambda_k(\theta_k^*)^2}{(\lambda_k + \lambda)^2}.
$$

Now we argue similarly for $\mathcal{V}$. By Lemma H.19:

$$
\begin{aligned}
\mathcal{V} &= \frac{\sigma^2}{n} \mathbb{E}_{\mathbf{x} \sim \mu} \left[ \left\| \left( \hat{\mathbf{\Sigma}} + \lambda \mathbf{I}_p \right)^{-1} \mathbf{x} \right\|_{\hat{\mathbf{\Sigma}}}^2 \right] \\
&= \frac{\sigma^2}{n} \mathbb{E}_{\mathbf{x} \sim \mu} \left[ \left\| (\mathbf{\Sigma} + \lambda \mathbf{I}_p)^{-1} \sum_{t=0}^{\infty} \mathbf{\Delta}^t \mathbf{x} \right\|_{\hat{\mathbf{\Sigma}}}^2 \right] \\
&= \frac{\sigma^2}{n} \mathbb{E}_{\mathbf{x}} \left[ \mathbf{x}^{\top} \left( \sum_{t=0}^{\infty} \mathbf{\Delta}^t \right)^{\top} (\mathbf{\Sigma} + \lambda \mathbf{I}_p)^{-1} \hat{\mathbf{\Sigma}} (\mathbf{\Sigma} + \lambda \mathbf{I}_p)^{-1} \sum_{t=0}^{\infty} \mathbf{\Delta}^t \mathbf{x} \right] \\
&= \frac{\sigma^2}{n} \mathbb{E}_{\mathbf{x}} \left[ \mathbf{x}^{\top} \left( \sum_{t=0}^{\infty} \mathbf{\Delta}^t \right)^{\top} (\mathbf{\Sigma} + \lambda \mathbf{I}_p)^{-1} \left( \mathbf{\Delta} + \mathbf{\Sigma} (\mathbf{\Sigma} + \lambda \mathbf{I}_p)^{-1} \right) \sum_{t=0}^{\infty} \mathbf{\Delta}^t \mathbf{x} \right] \\
&= \frac{\sigma^2}{n} \operatorname{Tr}[\mathbf{\Sigma}^2 (\mathbf{\Sigma} + \lambda \mathbf{I}_p)^{-2}] \left( 1 + \mathcal{O}_n \left( \|\mathbf{\Delta}\|_{\mathrm{op}} \right) \right).
\end{aligned}
$$

We argue similarly as in Proposition F.2: with probability at least $1 - o_n \left( n^{-\epsilon} \right)$, we have

$$
\mathcal{V} = \Theta_{n,p} \left( \frac{\sigma^2}{n} \sum_{k=1}^{p} \frac{\lambda_k^2}{(\lambda_k + \lambda)^2} \right).
$$

$\square$

Plugging in the values of $n, p$ and other metrics $a, b, r$ in Proposition F.2, we can obtain the

# G  Concentration of features

In the following, we shall give sufficient conditions to control the concentration coefficients $\zeta_{n,k}, \xi_{n,k}$ and $\rho_{n,k}$. Note that $\rho_{n,k}$, and $\xi_{n,k}$ depends on the whitened/isotropic features $\mathbf{z}$ but not on the spectrum $\mathbf{\Sigma}$. We start from the easiest to the most difficult.

**Proposition G.1** (Control on $\xi_{n,k}$). *Let $k \in \mathbb{N}$ be an integer. Recall that $\xi_{n,k} \overset{\text{def.}}{=} \frac{s_1(\mathbf{Z}_{\leq k}^\top \mathbf{Z}_{\leq k})}{n}$. If Assumption (GF) (or resp. (IF)) holds, then with probability at least $1 - 2\exp\left(-\frac{1}{2\beta_k^2}n\right)$ (or resp. $1 - 2\exp\left(-c_1 kn\right)$), it holds that*

$$\xi_{n,k} \geq \frac{1}{2}.$$

*Proof.* Since the largest singular value is larger than the average of the singular values,

$$\xi_{n,k} \overset{\text{def.}}{=} \frac{s_1(\mathbf{Z}_{\leq k}^\top \mathbf{Z}_{\leq k})}{n} \geq \frac{\frac{1}{k}\text{Tr}[\mathbf{Z}_{\leq k}^\top \mathbf{Z}_{\leq k}]}{n} = \frac{\text{Tr}[\mathbf{Z}_{\leq k}^\top \mathbf{Z}_{\leq k}]}{kn}.$$

By Lemma G.5, with respective probability,

$$\xi_{n,k} \geq \frac{1}{2}.$$

$\square$

**Proposition G.2** (Control on $\zeta_{n,k}$). *Let $k \leq n$ be an integer. Recall that $\zeta_{n,k} \overset{\text{def.}}{=} \frac{s_1(\mathbf{Z}_{\leq k}^\top \mathbf{Z}_{\leq k})}{s_n(\mathbf{Z}_{\leq k}^\top \mathbf{Z}_{\leq k})}$. If Assumption (GF) (or resp. (IF)) holds, then there exists some $c > 1$ such that, if $c\beta_k k \log k \leq n$ (or resp. $k/c \leq n$), then with probability at least $1 - 2\exp\left(-\frac{c}{\beta_k}\frac{n}{k}\right)$ (or resp. $1 - 2\exp\left(-c_1 n\right)$), it holds that*

$$\zeta_{n,k} \leq c_2.$$

*Proof.* For the case where Assumption (GF) holds, see Lemma 2 in [7]. Suppose Assumption (IF) holds, then by Theorem H.2, there exists constants $C_1, C_2 > 0$, such that, with probability at least $1 - 2\exp\left(-C_1 t^2\right)$, the spectrum of random matrix $\mathbf{Z}_{\leq k} \in \mathbb{R}^{n \times k}$ is bounded:

$$\sqrt{n} - \sqrt{C_2 k} - t \leq s_k(\mathbf{Z}_{\leq k}) \leq s_1(\mathbf{Z}_{\leq k}) \leq \sqrt{n} + \sqrt{C_2 k} + t.$$

Set $t = \sqrt{n}/4$ and $c = \frac{1}{16C_2}$ so that if $k/c \leq n$, the bound becomes:

$$\frac{1}{4}\sqrt{n} \leq s_k(\mathbf{Z}_{\leq k}) \leq s_1(\mathbf{Z}_{\leq k}) \leq \frac{3}{2}\sqrt{n},$$

and hence $\zeta_{n,k} \leq (\frac{3}{2}\sqrt{n})/(\frac{1}{4}\sqrt{n}) = 6$ with a probability at least $1 - 2\exp\left(-\frac{C_1}{16}n\right)$. Set $c_1 = \frac{C_1}{16}$ and $c_2 = 6$ to conclude the statement. $\square$

**Proposition G.3** (Control on $\rho_{n,k}$). *Let $k \leq n$ be an integer. Recall that $\rho_{n,k} \overset{\text{def.}}{=} \frac{n\|\mathbf{\Sigma}_{>k}\|_{op} + s_1(\mathbf{A}_k)}{s_n(\mathbf{A}_k)}$. If Assumption (PE) holds and Assumption (GF) (or resp. (IF)) holds, then there exists some $c > 1$ such that, if $k = \frac{n}{c\log n}$ (or resp. $k = n/c$), then with probability at least $1 - \mathcal{O}_n\left(\frac{1}{\log n}\right)$ (or resp. $1 - 3e^{-n}$), it holds that*

$$\rho_{n,k} = \tilde{\mathcal{O}}_n\left(n^a\right). \quad \text{(or resp. } \rho_{n,k} = \mathcal{O}_n\left(1\right).)$$

*If $\lambda = \Theta_n\left(n^{-b}\right)$ with $b \in (0, 1+a]$ and Assumption (GF) holds. If $k = \lceil n^{\frac{b}{1+a}} \rceil$, with probability at least $1 - \mathcal{O}_n\left(\frac{1}{n}\right)$, it holds that*

$$\rho_{n,k} = \mathcal{O}_n\left(1\right).$$

*Proof.* The statement where Assumption (GF) holds is proved in Theorems 4 and 5 in [7]. The statement where Assumption (IF) holds can be proved proved via Lemma G.7: since:

$$\rho = \frac{n\,\|\boldsymbol{\Sigma}_{>k}\|_{\mathrm{op}} + s_1(\mathbf{A}_k)}{s_n(\mathbf{A}_k)}$$

$$= \mathcal{O}_{k,n}\left(\frac{n\lambda_k + n\lambda}{n\lambda_{n+k} + n\lambda}\right)$$

$$= \mathcal{O}_{k,n}\left(\frac{n\lambda_k}{n\lambda_{n+k}}\right)$$

$$= \mathcal{O}_{k,n}\left(\frac{k^{-1-a}}{(n+k)^{-1-a}}\right)$$

$$= \mathcal{O}_{k,n}\left((1+n/k)^{1+a}\right)$$

$$= \mathcal{O}_{k,n}\left((1+c)^{1+a}\right)$$

$$= \mathcal{O}_{k,n}\left(1\right).$$

with a probability of at least $1 - 3e^{-n}$. $\qquad\square$

**Proposition G.4** (Control on $\rho_{n,k}$ under exponential eigen-decay). *Let $k \leq n$ be an integer. If $\lambda_k = \Theta_k\left(e^{-ak}\right)$, $\lambda = \Theta_n\left(e^{-bn}\right)$ for some $a > 0$ and $b \in (0, a)$, and Assumption (GF) (or resp. (IF)) holds, choose $k = \lceil \frac{b}{a}n \rceil$, then with probability at least $1 - o_n\left(\frac{1}{n}\right)$ (or resp. $1 - \exp(-n)$), it holds that*

$$s_1(\mathbf{A}_k) = \mathcal{O}_n\left(ne^{-bn}\right), \quad \rho_{n,k} = \mathcal{O}_n\left(1\right).$$

*Proof.* For the case where Assumption (GF) holds, by Corollary 1 in [7], with probability at least $1 - o_n\left(\frac{1}{n}\right)$, it holds that

$$s_1(\mathbf{A}_k) = \mathcal{O}_{n,k}\left(n\left(\lambda_{k+1} + \frac{\log k\,\mathrm{Tr}[\boldsymbol{\Sigma}_{>k}]}{n} + \lambda\right)\right)$$

$$= \mathcal{O}_{n,k}\left(n\left(e^{-ak} + e^{-bn}\right)\right)$$

$$= \mathcal{O}_{n,k}\left(n\left(e^{-a\cdot\frac{b}{a}n} + e^{-bn}\right)\right)$$

$$= \mathcal{O}_n\left(ne^{-bn}\right).$$

For the case where Assumption (IF) holds, the upper bound of $s_1(\mathbf{A}_k)$ is proven in Lemma G.7. In both cases, the derivation of bounding $\rho$ is the same:

$$\rho = \frac{n\,\|\boldsymbol{\Sigma}_{>k}\|_{\mathrm{op}} + s_1(\mathbf{A}_k)}{s_n(\mathbf{A}_k)} \leq \frac{n\,\|\boldsymbol{\Sigma}_{>k}\|_{\mathrm{op}} + s_1(\mathbf{A}_k)}{n\lambda} = \mathcal{O}_{n,k}\left(\frac{ne^{-bn}}{ne^{-bn}}\right) = \mathcal{O}_{n,k}\left(1\right).$$

$\qquad\square$

**Lemma G.5** (Lemma 1 in [7]). *Let $k$ be an integer. Suppose Assumption (GF) (or resp. (IF)) holds. Then with probability at least $1 - 2\exp\left(-\frac{1}{2\beta_k}n\right)$ (or. resp. $1 - 2\exp(-c_1kn)$), it holds that*

$$\frac{1}{2}kn \leq \mathrm{Tr}[\mathbf{Z}_{\leq k}^\top \mathbf{Z}_{\leq k}] \leq \frac{3}{2}kn.$$

*Proof.* If Assumption (GF) holds, then

$$\mathrm{Tr}[\mathbf{Z}_{\leq k}^\top \mathbf{Z}_{\leq k}] = \mathrm{Tr}[\mathbf{Z}_{\leq k}\mathbf{Z}_{\leq k}^\top] = \sum_{i=1}^n \|(\mathbf{z}_i)_{\leq k}\|_2^2 \leq \beta_k kn.$$

Set $M = \beta_k k$ and by Hoeffding's inequality, the above trace concentrates:

$$\mathbb{P}\left\{\left|\mathrm{Tr}[\mathbf{Z}_{\leq k}\mathbf{Z}_{\leq k}^\top] - kn\right| \geq t\right\} \leq 2\exp\left(-\frac{2t^2}{nM^2}\right)$$

Set $t = nk/2$ to conclude the statement.

Analogously, if Assumption (IF) holds, for $i = 1, ..., n$ and $l = 1, ..., k$, $(z_i^{(l)})^2 - 1$ is centered sub-exponential variable with sub-exponential norm $\left\| (z_i^{(l)})^2 - 1 \right\|_{\psi_1} \lesssim G^2$. By Lemma H.7, with probability at least $1 - 2\exp(-c_1 kn)$,

$$\left| \text{Tr}[\mathbf{Z}_{\leq k}^\top \mathbf{Z}_{\leq k}] - kn \right| = \left| \sum_{i=1}^n \sum_{l=1}^k (z_i^{(l)})^2 - kn \right| \leq \frac{1}{2} kn.$$

$\square$

**Lemma G.6** (Lemma 3 in [7])**.** *For any $k \leq n$ and $\delta \in (0,1)$, with probability at least $1 - \delta$, it holds that*

$$\frac{1}{n} \left\| \mathbf{X}_{>k} \boldsymbol{\theta}_{>k}^* \right\|_2^2 \leq \frac{1}{\delta} \left\| \boldsymbol{\theta}_{>k}^* \right\|_{\boldsymbol{\Sigma}_{>k}}^2.$$

*Proof.* Since $\mathbb{E}\left[ \frac{1}{n} \left\| \mathbf{X}_{>k} \boldsymbol{\theta}_{>k}^* \right\|_2^2 \right] = \left\| \boldsymbol{\theta}_{>k}^* \right\|_{\boldsymbol{\Sigma}_{>k}}^2$, we use Markov's inequality to obtain the result. $\square$

**Lemma G.7.** *Suppose Assumption (IF) hold. Then for any integer $k \leq n$,*

1. *If Assumption (PE)/(EE) holds, then with probability at least $1 - e^{-n}$, it holds that*
$$s_1(\mathbf{A}_k) = \mathcal{O}_{n,k}(n\lambda_k + n\lambda).$$

2. *If Assumption (PE) holds, then with probability at least $1 - 2e^{-n}$, it holds that*
$$s_n(\mathbf{A}_k) = \Omega_{n,k}(n\lambda_{n+k} + n\lambda),$$

*for $p$ large enough.*

*Proof.* First we prove statement 1. For any $k \in \mathbb{N}$,

$$r_k \overset{\text{def.}}{=} \frac{\text{Tr}[\boldsymbol{\Sigma}_{>k}]}{\|\boldsymbol{\Sigma}_{>k}\|_{\text{op}}} = \begin{cases} \Theta_k \left( \frac{\sum_{l=k+1}^p l^{-1-a}}{(k+1)^{-1-a}} \right) \\ \Theta_k \left( \frac{\sum_{l=k+1}^p e^{-al}}{e^{-ak}} \right) \end{cases} = \begin{cases} \Theta_k \left( \frac{k^{-a}}{k^{-1-a}} \right) \\ \Theta_k \left( \frac{e^{-ak}}{e^{-ak}} \right) \end{cases} = \begin{cases} \Theta_k(k) & \text{Assumption (PE) holds} \\ \Theta_k(1) & \text{Assumption (EE) holds} \end{cases}.$$

Hence, set $t = n$ in Theorem H.5, with probability at least $1 - e^{-n}$, it holds that

$$\left\| \frac{1}{n} \mathbf{X}_{>k}^\top \mathbf{X}_{>k} - \boldsymbol{\Sigma}_{>k} \right\|_{\text{op}} = \mathcal{O}_{n,k} \left( \|\boldsymbol{\Sigma}_{>k}\|_{\text{op}} \max \left\{ \frac{r_k}{n}, \sqrt{\frac{r_k}{n}}, \frac{t}{n}, \sqrt{\frac{t}{n}} \right\} \right) = \mathcal{O}_{n,k}(\lambda_k)$$

By triangle inequality, $\left\| \mathbf{X}_{>k}^\top \mathbf{X}_{>k} \right\|_{\text{op}} \leq n \left\| \frac{1}{n} \mathbf{X}_{>k}^\top \mathbf{X}_{>k} - \boldsymbol{\Sigma}_{>k} \right\|_{\text{op}} + n \|\boldsymbol{\Sigma}_{>k}\|_{\text{op}} = \mathcal{O}_{n,k}(n\lambda_k)$. Hence, with the same probability,

$$s_1(\mathbf{A}_k) = s_1(\mathbf{X}_{>k} \mathbf{X}_{>k}^\top + n\lambda \mathbf{I}_n) = s_1(\mathbf{X}_{>k}^\top \mathbf{X}_{>k}) + n\lambda = \mathcal{O}_{n,k}(n\lambda_k + n\lambda).$$

Now we prove statement 2. Note that the smallest singular value of a matrix does not increase after discarding a column: for some constant $\eta > 1$ to be determined, it holds that

$$s_n(\mathbf{X}_{>k} \mathbf{X}_{>k}^\top) = s_n(\mathbf{Z}_{>k} \boldsymbol{\Sigma}_{>k}^{1/2})^2 \geq s_n(\mathbf{Z}_{k:\eta n} \boldsymbol{\Sigma}_{k:\eta n}^{1/2})^2 \geq \lambda_{\eta n} s_n(\mathbf{Z}_{k:\eta n})^2.$$

By Assumption (IF), the columns of $\mathbf{Z}_{k:\eta n}$ are independent to each other. By Theorem H.2, with probability at least $1 - 2e^{-c_1 t^2}$, it holds that

$$s_{\min}(\mathbf{Z}_{k:\eta n}) = s_{\min\{n, \eta n - k\}}(\mathbf{Z}_{k:\eta n}) \geq \sqrt{\eta n} - \sqrt{c_2 n} - t.$$

Choose $t = \sqrt{n/c_1}$ and $\eta > \left( 2 + \frac{1}{c_1} + c_2 \right)^2$ (given that $p > \eta n$ large enough), then with probability at least $1 - 2e^{-n}$, it holds that

$$s_n(\mathbf{Z}_{k:\eta n}) = s_{\min}(\mathbf{Z}_{k:\eta n}) \geq \sqrt{n}.$$

Hence with the same probability, by Assumption (PE),

$$s_n(\mathbf{A}_k) = s_n(\mathbf{X}_{>k} \mathbf{X}_{>k}^\top + n\lambda \mathbf{I}_n) \geq n\lambda_{\eta n} + n\lambda = \Omega_{n,k}(n\lambda_n + n\lambda).$$

$\square$

**Lemma G.8** (Theorem 2 in [7]). *Suppose Assumption (GF) (or resp. (IF)) holds, then with probability at least $1$ (or resp. $1 - 2pe^{-c_1 n}$), it holds that*

$$\frac{s_1(\mathbf{A}_k)^2}{n^2} \leq \rho_{n,k}^2 \left( \lambda + \frac{c_2 \operatorname{Tr}[\mathbf{\Sigma}_{>k}]}{n} \right)^2.$$

*Suppose Assumptions (IF) holds. Furthermore, if Assumptions (PE) or (EE) holds, then one obtains a probability bound which allows arbitrary large $p$: with probability at least $1 - e^{-n}$, it holds that*

$$\frac{s_1(\mathbf{A}_k)^2}{n^2} \leq c_1 \left( \lambda + \frac{\operatorname{Tr}[\mathbf{\Sigma}_{>k}]}{n} \right)^2.$$

*Proof.* Since the trace of a matrix is the sum of its eigenvalues, we obtain:

$$\frac{s_1(\mathbf{A}_k)^2}{n^2} = \frac{s_1(\mathbf{A}_k)^2}{s_n(\mathbf{A}_k)^2} \frac{s_n(\mathbf{A}_k)^2}{n^2}$$

$$\leq \rho_{n,k}^2 \left( \frac{1}{n} \operatorname{Tr}\left[ \frac{1}{n} \mathbf{A}_k \right] \right)^2$$

$$\leq \rho_{n,k}^2 \left( \frac{1}{n} \left( \operatorname{Tr}[\lambda \mathbf{I}_n] + \operatorname{Tr}\left[ \frac{1}{n} \sum_{i=1}^{n} (\mathbf{x}_i)_{>k} (\mathbf{x}_i)_{>k}^\top \right] \right) \right)^2$$

$$\leq \rho_{n,k}^2 \left( \frac{1}{n} \left( n\lambda + \frac{1}{n} \sum_{i=1}^{n} \|(\mathbf{x}_i)_{>k}\|_2^2 \right) \right)^2$$

By Assumption (GF), since $\operatorname{ess\,sup}_{\mathbf{x}} \frac{\|\mathbf{x}_{>k}\|_2^2}{\operatorname{Tr}[\mathbf{\Sigma}_{>k}]} \leq \beta_k$, we have

$$\frac{s_1(\mathbf{A}_k)^2}{n^2} \leq \rho_{n,k}^2 \left( \frac{1}{n} \left( n\lambda + \sup_{\mathbf{x}} \|\mathbf{x}_{>k}\|_2^2 \right) \right)^2 \leq \rho_{n,k}^2 \left( \lambda + \frac{\beta_k \operatorname{Tr}[\mathbf{\Sigma}_{>k}]}{n} \right)^2.$$

If Assumption (IF) holds, then for each $l > k$, the random variable $(z^{(l)})^2 - 1$ is centered sub-exponential, hence with probability at least $1 - 2\exp(-c_1 n)$, it holds that

$$\left| \sum_{i=1}^{n} (z_i^{(l)})^2 - n \right| \leq \frac{1}{2} n. \tag{54}$$

By union bound, with probability at least $1 - 2(p - k)\exp(-c_1 n)$, it holds that

$$\frac{1}{n} \sum_{i=1}^{n} \|(\mathbf{x}_i)_{>k}\|_2^2 = \sum_{l>k} \lambda_l \frac{1}{n} \sum_{i=1}^{n} (z_i^{(l)})^2 \leq \sum_{l>k} \lambda_l \cdot \frac{3}{2} = \frac{3}{2} \operatorname{Tr}[\mathbf{\Sigma}_{>k}]. \tag{55}$$

If Assumptions (IF) and (PE)/(EE) hold, then by Lemma G.7: with probability at least $1 - e^{-n}$, it holds that

$$s_1(\mathbf{A}_k) = \mathcal{O}_{n,k}(n\lambda_n + n\lambda)$$

for any $k \in \mathbb{N}$. Hence with the same probability,

$$\frac{s_1(\mathbf{A}_k)^2}{n^2} = \mathcal{O}_n \left( \left( \frac{n\lambda + n\lambda_n}{n} \right)^2 \right) = \mathcal{O}_n \left( \left( \lambda + \frac{\operatorname{Tr}[\mathbf{\Sigma}_k]}{n} \right)^2 \right).$$

$\square$

# H Technical lemmata

**Theorem H.1** (Theorem 1 and Example 1 in [51]). *Assume that $\mathbf{M}_1, \mathbf{M}_2, .., \mathbf{M}_n$ are independent copies of a $d \times d$ positive semi-definite symmetric random matrix $\mathbf{M}$ with $\mathbb{E}[\mathbf{M}] = \boldsymbol{\Sigma}$, which satisfies:*

$$\|\mathbf{v}^\top \mathbf{M} \mathbf{v}\|_{\psi_1} \leq \kappa^2 \mathbf{v}^\top \mathbf{M} \mathbf{v} \tag{56}$$

*for some constant $\kappa \geq 1$ and for any $\mathbf{v} \in \mathbb{R}^d$. Then, for any $t > 0$, with probability at least $1 - e^{-t}$, the inequality holds:*

$$\left\| \frac{1}{n} \sum_{i=1} \mathbf{M}_i - \boldsymbol{\Sigma} \right\|_{op} \leq 20\kappa^2 \|\boldsymbol{\Sigma}\|_{op} \sqrt{\frac{4r+t}{n}}$$

*whenever $n \geq 4r + t$, and $r \overset{\text{def.}}{=} \frac{\mathrm{Tr}[\boldsymbol{\Sigma}]}{\|\boldsymbol{\Sigma}\|_{op}}$ is the effective rank.*

**Theorem H.2** (Theorem 5.39 and Remark 5.40 in [47]). *Let $\mathbf{A}$ be an $N \times n$ matrix with independent rows $\mathbf{A}_i$ of sub-Gaussian random vector with covariance $\boldsymbol{\Sigma} \overset{\text{def.}}{=} \mathbb{E}\left[\mathbf{A_i}\mathbf{A_i}^\top\right] \in \mathbb{R}^{n \times n}$. Then there exists constants $C_5, C_6 > 0$ [15] (depending only on the sub-Gaussian norm of entries of $\mathbf{A}$), such that for any $t \geq 0$, with probability at least $1 - 2e^{-C_5 t^2}$, we have*

$$\left\| \frac{1}{N} \mathbf{A}^\top \mathbf{A} - \boldsymbol{\Sigma} \right\|_{op} \leq \max\{\delta, \delta^2\} \|\boldsymbol{\Sigma}\|_{op}.$$

*where $\delta = C_6 \sqrt{\frac{n}{N}} + \frac{t}{N}$. In particular, if $\boldsymbol{\Sigma} = \mathbf{I}_n$, we have*

$$\sqrt{N} - \sqrt{C_6 n} - t \leq s_{\min}(\mathbf{A}) \leq s_{\max}(\mathbf{A}) \leq \sqrt{N} + \sqrt{C_6 n} + t.$$

**Theorem H.3** (Theorem 5.41 in [47]). *Let $\mathbf{A}$ be an $N \times n$ matrix whose rows $\mathbf{A}_i$ are independent isotropic random vectors in $\mathbb{R}^n$. Let $m > 0$ be a number such that $\|\mathbf{A}_i\|_2 \leq \sqrt{m}$ a.s. for all $i$. Then for every $t \geq 0$, it holds that:*

$$\sqrt{N} - t\sqrt{m} \leq s_{\min}(\mathbf{A}) \leq s_{\max}(\mathbf{A}) \leq \sqrt{N} + t\sqrt{m}.$$

**Theorem H.4** (Theorem 5.58 in [47]). *Let $\mathbf{A}$ be an $N \times n$ matrix ($N \geq n$) with independent columns $\mathbf{A}_i \in \mathbb{R}^N$ of sub-Gaussian isotropic random vector with with $\|\mathbf{A}_i\|_2 = \sqrt{N}$ almost surely. Then there exists constants $C_8, C_9 > 0$ (depending only on the sub-Gaussian norm of entries of $\mathbf{A}$), such that for any $t \geq 0$, with probability at least $1 - 2e^{-C_8 t^2}$, we have*

$$\sqrt{N} - C_9 \sqrt{n} - t \leq s_{\min}(\mathbf{A}) \leq s_{\max}(\mathbf{A}) \leq \sqrt{N} + C_9 \sqrt{n} + t.$$

**Theorem H.5** (Theorem 9 in [27]). *Let $\mathbf{A}$ be an $N \times n$ matrix with i.i.d. columns $\mathbf{C}_i \in \mathbb{R}^N$ of sub-Gaussian random vector with covariance $\boldsymbol{\Sigma} \overset{\text{def.}}{=} \mathbb{E}\left[\mathbf{C}_i \mathbf{C}_i^\top\right] \in \mathbb{R}^{N \times N}$. Then there exists a constant $c_1 > 0$, such that for any $t \geq 0$, with probability at least $1 - e^{-t}$, we have*

$$\left\| \frac{1}{n} \mathbf{A}\mathbf{A}^\top - \boldsymbol{\Sigma} \right\|_{op} \leq c_1 \|\boldsymbol{\Sigma}\|_{op} \max\left\{ \frac{r}{n}, \sqrt{\frac{r}{n}}, \frac{t}{n}, \sqrt{\frac{t}{n}} \right\},$$

*where $r = \frac{\mathrm{Tr}[\boldsymbol{\Sigma}]}{\|\boldsymbol{\Sigma}\|_{op}}$ is the effective rank of the covariance $\boldsymbol{\Sigma}$.*

*Remark* H.6. Theorem H.5 is different from Theorem H.2 in a few ways: first, the upper bound in the former one contains the term $\frac{n}{N}$, which requires $n < N$ in order to obtain a good concentration, while that in the latter one contains the term $\frac{r}{n} = \frac{\sum_{k=1}^N \boldsymbol{\Sigma}_{kk}}{n}$, which can still be bounded if $N > n$ and the decay of $\boldsymbol{\Sigma}_{kk}$ is fast enough; second, former one requires i.i.d. columns while the latter one requires only independent rows.

**Lemma H.7** (Sub-Exponential Deviation, see Corollary 5.17 in [47]). *Let $N \in \mathbb{N}$. Let $X_1, ..., X_N$ be independent centered random variables with sub-exponential norms bounded by $B$. Then for any $\delta > 0$,*

$$\mathbb{P}\left\{ |\sum_{i=1}^N X_i| > \delta N \right\} \leq 2\exp\left( -C_7 \min\left\{ \frac{\delta^2}{B^2}, \frac{\delta}{B} \right\} N \right),$$

---

[15]To be precise, we set $C_5 = (8e^2)^{-1} \cdot G^{-4}$ and $C_6 = 2e\sqrt{2\log 9} \cdot G^2$.

*where $C_7 > 0$ is an absolute constant.*

*In particular, if $X \sim \chi(N)$ is the Chi-square distribution, then $\mathbb{P}\left\{|\frac{X}{N} - 1| > t\right\} \leq 2e^{-Nt^2/8}$, $\forall t \in (0, 1)$.*

**Lemma H.8** (Lemma 28 in [46]; Lemma 14 in [7]). *Suppose that for some $k < N$, the matrix $\mathbf{A}_k$ is positive definite, then the following inequality holds:*

$$
\begin{aligned}
\mathcal{B} \leq & \|\boldsymbol{\theta}^*_{>k}\|^2_{\boldsymbol{\Sigma}_{>k}} + \frac{s_1(\mathbf{A}_k^{-1})^2}{s_n(\mathbf{A}_k^{-1})^2} \frac{s_1(\mathbf{Z}^\top_{\leq k}\mathbf{Z}_{\leq k})}{s_k(\mathbf{Z}^\top_{\leq k}\mathbf{Z}_{\leq k})^2} \|\mathbf{X}_{>k}\boldsymbol{\theta}^*_{>k}\|^2_2 \\
& + \frac{\|\boldsymbol{\theta}^*_{\leq k}\|^2_{\boldsymbol{\Sigma}^{-1}_{\leq k}}}{s_n(\mathbf{A}_k^{-1})^2 s_k(\mathbf{Z}^\top_{\leq k}\mathbf{Z}_{\leq k})^2} \\
& + \|\boldsymbol{\Sigma}_{>k}\|_{op}\, s_1(\mathbf{A}_k^{-1})\, \|\mathbf{X}_{>k}\boldsymbol{\theta}^*_{>k}\|^2_2 \\
& + \|\boldsymbol{\Sigma}_{>k}\|_{op}\, \frac{s_1(\mathbf{A}_k^{-1})}{s_n(\mathbf{A}_k^{-1})^2} \frac{s_1(\mathbf{Z}^\top_{\leq k}\mathbf{Z}_{\leq k})}{s_k(\mathbf{Z}^\top_{\leq k}\mathbf{Z}_{\leq k})^2} \|\boldsymbol{\theta}^*_{\leq k}\|^2_{\boldsymbol{\Sigma}^{-1}_{\leq k}}.
\end{aligned}
\tag{57}
$$

**Lemma H.9** (Lemma 27 in [46]; Lemma 13 in [7]). *For any $k \in \mathbb{N}$, we have*

$$
\mathcal{V}/\sigma^2 \leq \frac{s_1(\mathbf{A}_k^{-1})^2 \operatorname{Tr}[\mathbf{X}_{\leq k}\boldsymbol{\Sigma}^{-1}_{\leq k}\mathbf{X}^\top_{\leq k}]}{s_n(\mathbf{A}_k^{-1})^2 s_k(\boldsymbol{\Sigma}^{-1/2}_{\leq k}\mathbf{X}^\top_{\leq k}\mathbf{X}_{\leq k}\boldsymbol{\Sigma}^{-1/2}_{\leq k})^2} + s_1(\mathbf{A}_k^{-1})^2 \operatorname{Tr}[\mathbf{X}_{>k}\boldsymbol{\Sigma}_{>k}\mathbf{X}^\top_{>k}]
$$

*, where $\sigma^2 \overset{\text{def.}}{=} \mathbb{E}\left[\epsilon^2\right] \geq 0$.*

**Lemma H.10** (Lemma 18 in [7]). *Assume $\lambda_k = \Theta_k\left(k^{-1-a}\right)$ and $|\theta^*_k| = \mathcal{O}_k\left(k^{-r}\right)$ for some $a, r > 0$. Then the square norms $\|\boldsymbol{\theta}^*_{>k}\|^2_{\boldsymbol{\Sigma}_{>k}}$ and $\|\boldsymbol{\theta}^*_{\leq k}\|^2_{\boldsymbol{\Sigma}^{-1}_{\leq k}}$ have bounds:*

$$
\|\boldsymbol{\theta}^*_{>k}\|^2_{\boldsymbol{\Sigma}_{>k}} = \Theta_k\left(k^{-2r-a}\right), \qquad \|\boldsymbol{\theta}^*_{\leq k}\|^2_{\boldsymbol{\Sigma}^{-1}_{\leq k}} = \begin{cases} \Theta_k\left(k^{2-2r+a}\right), & 2r < 2+a \\ \Theta_k\left(\log k\right), & 2r = 2+a \\ \Theta_k\left(1\right), & 2r > 2+a \end{cases} = \tilde{\Theta}_k\left(k^{(2+a-2r)_+}\right),
$$

*where $(x)_+ \overset{\text{def.}}{=} \max\{x, 0\}$.*

**Lemma H.11.** *Assume $\lambda_k = \Theta_k\left(e^{-ak}\right)$ and $|\theta^*_k| = \mathcal{O}_k\left(e^{-rk}\right)$ for some $a, r > 0$. Then the square norms $\|\boldsymbol{\theta}^*_{>k}\|^2_{\boldsymbol{\Sigma}_{>k}}$ and $\|\boldsymbol{\theta}^*_{\leq k}\|^2_{\boldsymbol{\Sigma}^{-1}_{\leq k}}$ have bounds:*

$$
\|\boldsymbol{\theta}^*_{>k}\|^2_{\boldsymbol{\Sigma}_{>k}} = \Theta_k\left(k^{-2r-a}\right), \qquad \|\boldsymbol{\theta}^*_{\leq k}\|^2_{\boldsymbol{\Sigma}^{-1}_{\leq k}} \leq \begin{cases} \Theta_k\left(k^{(a-2r)}\right), & 2r < a \\ \Theta_k\left(k\right), & 2r = a \\ \Theta_k\left(1\right), & 2r > a \end{cases} = k^{\tilde{\Theta}_k((a-2r)_+)}.
$$

**Lemma H.12.** *Recall that*

$$
r_k \overset{\text{def.}}{=} \frac{\operatorname{Tr}[\boldsymbol{\Sigma}_{>k}]}{\|\boldsymbol{\Sigma}_{>k}\|_{op}}, \qquad R_k \overset{\text{def.}}{=} \frac{\operatorname{Tr}[\boldsymbol{\Sigma}_{>k}]^2}{\operatorname{Tr}[\boldsymbol{\Sigma}^2_{>k}]}.
$$

*If $\lambda_k = \Theta_k\left(k^{-1-a}\right)$ for some $a > 0$, then*

$$
r_k = \Theta_k\left(k\right), \qquad R_k = \Theta_k\left(k\right).
$$

*If $\lambda_k = \Theta_k\left(e^{-ak}\right)$ for some $a > 0$, then*

$$
r_k = \Theta_k\left(1\right), \qquad R_k = \Theta_k\left(1\right).
$$

*Proof.* By simple calculus, $\sum_{l=k+1}^p k^{-1-a} = \Theta_k\left(\int_k^\infty t^{-1-a}dt\right) = \Theta_k\left(t^{-a}\right)$, similarly, $\sum_{l=k+1}^p k^{-2-2a} = \Theta_k\left(\int_k^\infty t^{-2-2a}dt\right) = \Theta_k\left(t^{-1-2a}\right)$. If $\lambda_k = \Theta_k\left(k^{-1-a}\right)$ for some $a > 0$, then

$$
r_k = \Theta_k\left(\frac{k^{-a}}{\lambda_{k+1}}\right) = \Theta_k\left(k\right), \qquad R_k = \Theta_k\left(\frac{k^{-2a}}{k^{-1-2a}}\right) = \Theta_k\left(k\right).
$$

If $\lambda_k = \Theta_k\left(e^{-ak}\right)$, then $\sum_{l=k+1}^{p} \lambda_l = \Theta_k\left(\int_k^\infty e^{-at}dt\right) = \Theta_k\left(e^{-ak}\right)$. Hence,

$$r_k = \Theta_k\left(\frac{e^{-ak}}{e^{-a(k+1)}}\right) = \Theta_k\left(1\right), \quad R_k = \Theta_k\left(\frac{e^{-2ak}}{e^{-2ak}}\right) = \Theta_k\left(1\right).$$

$\square$

**Lemma H.13** (Lemma 7 in [46]). *Suppose Assumption (IF) holds. Then there exists some constant $c > 0$ such that, for any $k < n/c$, with probability at least $1 - ce^{-n/c}$, its holds that*

$$\mathcal{V} \geq \frac{1}{cn} \sum_{l=1} \min\left\{1, \frac{\lambda_l^2}{\lambda_{k+1}^2(1 + r_k/n)^2}\right\}$$

**Lemma H.14** (Lemma 8 in [46]). *Suppose Assumption (PS) holds. Furthermore, if Assumption (GF) (or resp. (IF)) holds, then with probability $1 - 1$, the following inequality holds:*

$$\mathbb{E}_{\boldsymbol{\theta}^* \sim \vartheta}\left[\mathcal{B}\right] \geq \sum_{l=1}^{p} \frac{\lambda_l \mathbb{E}_{\boldsymbol{\theta}^*}\left[(\theta_l^*)^2\right]}{\left(1 + \lambda_l s_n(\mathbf{A}_{-l})^{-1}\left\|\mathbf{z}^{(l)}\right\|_2^2\right)^2}. \tag{58}$$

*where the expectation is taken as described in Assumption (PS), and $\mathbf{A}_{-l} \overset{\text{def.}}{=} \sum_{l' \neq l} \lambda_{l'} \mathbf{z}^{(l')}(\mathbf{z}^{(l')})^\top + n\lambda\mathbf{I}_n \in \mathbb{R}^{n \times n}$ where $\mathbf{z}^{(l)} \in \mathbb{R}^n$ denotes the $l$-th column of the whitened feature block $\mathbf{Z} \overset{\text{def.}}{=} \mathbf{X}\boldsymbol{\Sigma}^{-1/2}$.*

*Proof.* By Assumption (PS), the expected value over $\boldsymbol{\theta}^*$ on the bias admits the following expression as in line (40):

$$\mathbb{E}_{\boldsymbol{\theta}^*}\left[\mathcal{B}\right] = \mathbb{E}_{\boldsymbol{\theta}^*}\left[\left\|(\mathbf{I}_p - \mathbf{P}_\lambda)\boldsymbol{\theta}^*\right\|_{\boldsymbol{\Sigma}}^2\right] = \sum_{l=1}^{p}[(\mathbf{I}_p - \mathbf{P}_\lambda)\boldsymbol{\Sigma}(\mathbf{I}_p - \mathbf{P}_\lambda)]_{ll} \cdot \mathbb{E}_{\boldsymbol{\theta}^*}\left[(\theta_l^*)^2\right]$$

where $\mathbf{P}_\lambda \overset{\text{def.}}{=} \mathbf{X}^\top(\mathbf{X}\mathbf{X}^\top + n\lambda\mathbf{I}_p)^{-1}\mathbf{X}$. Denote $\mathbf{z}^{(l)} \in \mathbb{R}^n$ be the $l$-th column of the whitened feature block $\mathbf{Z} \overset{\text{def.}}{=} \mathbf{X}\boldsymbol{\Sigma}^{-1/2} \in \mathbb{R}^{n \times p}$. Then the $l$-th diagonal element of the matrix $(\mathbf{I}_p - \mathbf{P}_\lambda)\boldsymbol{\Sigma}(\mathbf{I}_p - \mathbf{P}_\lambda)$ can be written as

$$[(\mathbf{I}_p - \mathbf{P}_\lambda)\boldsymbol{\Sigma}(\mathbf{I}_p - \mathbf{P}_\lambda)]_{ll} = \sum_{l'=1}^{p} \lambda_l \left(1 - \lambda_l(\mathbf{z}^{(l)})^\top \mathbf{A}^{-1}\mathbf{z}^{(l)}\right)^2 + \sum_{l' \neq l} \lambda_l \lambda_{l'}^2 \left((\mathbf{z}^{(i)})^\top \mathbf{A}^{-1}\mathbf{z}^{(l')}\right)^2$$

where $\mathbf{A} \overset{\text{def.}}{=} \mathbf{X}\mathbf{X}^\top + n\lambda\mathbf{I}_p$. If we write $\mathbf{A}_{-l} \overset{\text{def.}}{=} \mathbf{A} - \lambda_l \mathbf{z}^{(l)}(\mathbf{z}^{(l)})^\top$, by Sherman-Morrison-Woodbury formula, we have

$$1 - \lambda_l(\mathbf{z}^{(l)})^\top \mathbf{A}^{-1}\mathbf{z}^{(l)} = \frac{1}{1 + \lambda_l(\mathbf{z}^{(l)})^\top \mathbf{A}_{-l}^{-1}\mathbf{z}^{(l)}},$$

and hence the averaged bias is bounded by:

$$\begin{aligned}
\mathbb{E}_{\boldsymbol{\theta}^*}\left[\mathcal{B}\right] &\geq \sum_{l=1}^{p} \frac{\lambda_l \mathbb{E}_{\boldsymbol{\theta}^*}\left[(\theta_l^*)^2\right]}{(1 + \lambda_l(\mathbf{z}^{(l)})^\top \mathbf{A}_{-l}^{-1}\mathbf{z}^{(l)})^2} \\
&\geq \sum_{l=1}^{p} \frac{\lambda_l \mathbb{E}_{\boldsymbol{\theta}^*}\left[(\theta_l^*)^2\right]}{\left(1 + \lambda_l s_1(\mathbf{A}_{-l}^{-1})\left\|\mathbf{z}^{(l)}\right\|_2^2\right)^2} \\
&= \sum_{l=1}^{p} \frac{\lambda_l \mathbb{E}_{\boldsymbol{\theta}^*}\left[(\theta_l^*)^2\right]}{\left(1 + \lambda_l s_n(\mathbf{A}_{-l})^{-1}\left\|\mathbf{z}^{(l)}\right\|_2^2\right)^2}.
\end{aligned}$$

$\square$

**Lemma H.15** (Lemma 9 in [6]). *Suppose that $\{X_k\}_{k=1}^{p}$ is a sequence of non-negative random variables, and that $\{t_k\}_{k=1}^{p}$ is sequence of non-negative real numbers (with at least one of which strictly positive), such that for some $\delta \in (0,1)$, with a probability at least $1 - \delta$, $\eta_k > t_k$ for all $k = 1, ..., p$. Theb with probability at least $1 - 2\delta$,*

$$\sum_{k=1}^{p} \eta_k \geq \frac{1}{2} \sum_{k=1}^{p} t_k.$$

**Lemma H.16** (Theorem 10 in [46]). *Let $k \in \mathbb{N}$ be an integer. Denote*

$$\underline{B} \overset{\text{def.}}{=} \sum_{l=1}^{p} \frac{\lambda_l |\theta_l^*|^2}{(1 + \frac{n\lambda_l}{\lambda_{k+1} r_k})^2},$$

$$\overline{B} \overset{\text{def.}}{=} \|\boldsymbol{\theta}_{>k}^*\|_{\boldsymbol{\Sigma}_{>k}}^2 + \left(\frac{n\lambda + \text{Tr}[\boldsymbol{\Sigma}_{>k}]}{n}\right)^2 \|\boldsymbol{\theta}_{\leq k}^*\|_{\boldsymbol{\Sigma}_{\leq k}^{-1}}^2,$$

$$\underline{V} \overset{\text{def.}}{=} \frac{1}{n} \sum_{l=1}^{p} \min\left\{1, \frac{\lambda_l^2}{\lambda_{k+1}^2 (1 + r_k/n)^2}\right\},$$

$$\overline{V} \overset{\text{def.}}{=} \frac{k}{n} + \frac{n \text{Tr}[\boldsymbol{\Sigma}_{>k}^2]}{(n\lambda + \text{Tr}[\boldsymbol{\Sigma}_{>k}])^2}.$$

*Fix constants $a > 0$ and $b > \frac{1}{n}$. There exists a constant $c > 0$ that only depends on $a, b$ such that: if either $\frac{r_k}{n} \in (a, b)$ or $k = \min\{\kappa : r_\kappa > bn\}$, then*

$$c^{-1} \leq \underline{B}/\overline{B} \leq 1,$$
$$c^{-1} \leq \underline{V}/\overline{V} \leq 1.$$

**Lemma H.17** (Corde's inequality, [18]). *For any positive definite symmetric matrices [16] $\mathbf{M}_1, \mathbf{M}_2$ and positive number $m \in [0, 1]$, it holds that $\|\mathbf{M}_1^m \mathbf{M}_2^m\|_{op} \leq \|\mathbf{M}_1 \mathbf{M}_2\|_{op}^m$.*

**Proposition H.18.** *Let $\lambda > 0$. The bias term $\mathcal{B} \overset{\text{def.}}{=} \mathbb{E}_{\mathbf{x}}\left[\left(\mathbf{x}^\top \hat{\boldsymbol{\theta}}(\mathbf{X}\boldsymbol{\theta}^*) - \mathbf{x}^\top \boldsymbol{\theta}^*\right)^2\right]$ has the following expression:*

$$\mathcal{B} = \lambda^2 \left\|(\lambda \mathbf{I}_p + \hat{\boldsymbol{\Sigma}})^{-1} \boldsymbol{\theta}^*\right\|_{\boldsymbol{\Sigma}}^2.$$

*where $\hat{\boldsymbol{\Sigma}} \overset{\text{def.}}{=} \frac{1}{n} \mathbf{X}^\top \mathbf{X}$.*

*Proof.* By definition, rewrite the bias into:

$$\begin{aligned} \mathcal{B} &= \left\|\boldsymbol{\theta}^* - \hat{\boldsymbol{\theta}}(\mathbf{X}\boldsymbol{\theta}^*)\right\|_{\boldsymbol{\Sigma}}^2 \\ &= \left\|\boldsymbol{\theta}^* - \mathbf{X}^\top (\mathbf{X}\mathbf{X}^\top + n\lambda \mathbf{I}_n)^{-1}(\mathbf{X}\boldsymbol{\theta}^*)\right\|_{\boldsymbol{\Sigma}}^2 \\ &= \left\|\left(\mathbf{I}_p - \underbrace{\mathbf{X}^\top (\mathbf{X}\mathbf{X}^\top + n\lambda \mathbf{I}_n)^{-1}\mathbf{X}}_{\mathbf{P}_\lambda}\right) \boldsymbol{\theta}^*\right\|_{\boldsymbol{\Sigma}}^2. \end{aligned} \tag{59}$$

Denote $\mathbf{P}_\lambda \overset{\text{def.}}{=} \mathbf{X}^\top (\mathbf{X}\mathbf{X}^\top + n\lambda \mathbf{I}_n)^{-1}\mathbf{X} \in \mathbb{R}^{p \times p}$. By Sherman-Morrison-Woodbury formula,

$$\mathbf{I}_p - \mathbf{P}_\lambda = \mathbf{I}_p - (n\lambda)^{-1}\mathbf{X}^\top \left(\mathbf{I}_n + (n\lambda)^{-1}\mathbf{X}\mathbf{X}^\top\right)^{-1}\mathbf{X} = \lambda\left(\lambda \mathbf{I}_p + \frac{1}{n}\mathbf{X}^\top \mathbf{X}\right)^{-1} = \lambda\left(\lambda \mathbf{I}_p + \hat{\boldsymbol{\Sigma}}\right)^{-1}.$$

Hence $\mathcal{B} = \lambda^2 \left\|(\lambda \mathbf{I}_p + \hat{\boldsymbol{\Sigma}})^{-1}\boldsymbol{\theta}^*\right\|_{\boldsymbol{\Sigma}}^2.$ $\qquad\square$

**Proposition H.19** (Variance expression). *The variance term $\mathcal{V} \overset{\text{def.}}{=} \mathbb{E}_{\mathbf{x}, \boldsymbol{\epsilon}}\left[\left(\mathbf{x}^\top \hat{\boldsymbol{\theta}}^*(\boldsymbol{\epsilon})\right)^2\right]$ has the following expression:*

$$\mathcal{V} = \frac{\sigma^2}{n} \text{Tr}\left[\left(\hat{\boldsymbol{\Sigma}} + \lambda \mathbf{I}_p\right)^{-1} \boldsymbol{\Sigma} \left(\hat{\boldsymbol{\Sigma}} + \lambda \mathbf{I}_p\right)^{-1} \hat{\boldsymbol{\Sigma}}\right] = \frac{\sigma^2}{n} \mathbb{E}_{\mathbf{x} \sim \mu}\left[\left\|\left(\hat{\boldsymbol{\Sigma}} + \lambda \mathbf{I}_p\right)^{-1} \mathbf{x}\right\|_{\hat{\boldsymbol{\Sigma}}}^2\right].$$

---

[16]Or equivalently positive definite self-adjoint operator in a Hilbert space.

*Proof.* By definition,

$$\mathcal{V} = \mathbb{E}_{\boldsymbol{\epsilon}}\left[\left\|\hat{\boldsymbol{\theta}}(\boldsymbol{\epsilon})\right\|_{\boldsymbol{\Sigma}}^2\right]$$

$$= \mathbb{E}_{\boldsymbol{\epsilon}}\left[\boldsymbol{\epsilon}^\top(\mathbf{X}\mathbf{X}^\top + n\lambda\mathbf{I}_n)^{-1}\mathbf{X}\boldsymbol{\Sigma}\mathbf{X}^\top(\mathbf{X}\mathbf{X}^\top + n\lambda\mathbf{I}_n)^{-1}\boldsymbol{\epsilon}\right]$$

$$= \mathbb{E}_{\boldsymbol{\epsilon}}\left[\text{Tr}[(\mathbf{X}\mathbf{X}^\top + n\lambda\mathbf{I}_n)^{-1}\mathbf{X}\boldsymbol{\Sigma}\mathbf{X}^\top(\mathbf{X}\mathbf{X}^\top + n\lambda\mathbf{I}_n)^{-1}\boldsymbol{\epsilon}\boldsymbol{\epsilon}^\top]\right]$$

$$= \text{Tr}[(\mathbf{X}\mathbf{X}^\top + n\lambda\mathbf{I}_n)^{-1}\mathbf{X}\boldsymbol{\Sigma}\mathbf{X}^\top(\mathbf{X}\mathbf{X}^\top + n\lambda\mathbf{I}_n)^{-1}\mathbb{E}_{\boldsymbol{\epsilon}}\left[\boldsymbol{\epsilon}\boldsymbol{\epsilon}^\top\right]]$$

$$= \sigma^2\,\text{Tr}[(\mathbf{X}\mathbf{X}^\top + n\lambda\mathbf{I}_n)^{-1}\mathbf{X}\boldsymbol{\Sigma}\mathbf{X}^\top(\mathbf{X}\mathbf{X}^\top + n\lambda\mathbf{I}_n)^{-1}]$$

$$= \sigma^2\,\text{Tr}[\mathbf{X}(\mathbf{X}^\top\mathbf{X} + n\lambda\mathbf{I}_p)^{-1}\boldsymbol{\Sigma}(\mathbf{X}^\top\mathbf{X} + n\lambda\mathbf{I}_p)^{-1}\mathbf{X}^\top]$$

$$= \frac{\sigma^2}{n}\,\text{Tr}\left[\left(\frac{1}{n}\mathbf{X}^\top\mathbf{X} + \lambda\mathbf{I}_p\right)^{-1}\boldsymbol{\Sigma}\left(\frac{1}{n}\mathbf{X}^\top\mathbf{X} + \lambda\mathbf{I}_p\right)^{-1}\frac{1}{n}\mathbf{X}^\top\mathbf{X}\right]$$

$$= \frac{\sigma^2}{n}\,\text{Tr}\left[\left(\hat{\boldsymbol{\Sigma}} + \lambda\mathbf{I}_p\right)^{-1}\mathbb{E}_{\mathbf{x}\sim\mu}\left[\mathbf{x}\mathbf{x}^\top\right]\left(\hat{\boldsymbol{\Sigma}} + \lambda\mathbf{I}_p\right)^{-1}\hat{\boldsymbol{\Sigma}}\right]$$

$$= \frac{\sigma^2}{n}\mathbb{E}_{\mathbf{x}\sim\mu}\left[\text{Tr}\left[\left(\hat{\boldsymbol{\Sigma}} + \lambda\mathbf{I}_p\right)^{-1}\mathbf{x}\mathbf{x}^\top\left(\hat{\boldsymbol{\Sigma}} + \lambda\mathbf{I}_p\right)^{-1}\hat{\boldsymbol{\Sigma}}\right]\right]$$

$$= \frac{\sigma^2}{n}\mathbb{E}_{\mathbf{x}\sim\mu}\left[\mathbf{x}^\top\left(\hat{\boldsymbol{\Sigma}} + \lambda\mathbf{I}_p\right)^{-1}\hat{\boldsymbol{\Sigma}}\left(\hat{\boldsymbol{\Sigma}} + \lambda\mathbf{I}_p\right)^{-1}\mathbf{x}\right]$$

$$= \frac{\sigma^2}{n}\mathbb{E}_{\mathbf{x}\sim\mu}\left[\left\|\left(\hat{\boldsymbol{\Sigma}} + \lambda\mathbf{I}_p\right)^{-1}\mathbf{x}\right\|_{\hat{\boldsymbol{\Sigma}}}^2\right].$$

$\square$

# I  Experiments in details

Some of the results in Table 1 have already been validated by experiments in previous literature. Hence, this section will focus on the novel result of this paper:

1. Same decay rate for independent (IF)/generic (GF) features under strong ridge;
2. Decay of $\mathcal{B}$ for $s < 1$ under weak ridge;
3. Decay of $\mathcal{V}$ under weak ridge (tempered vs catastrophic overfitting).

All experiments were conducted on a computer with a 2.3 GHz Quad-Core Intel Core i7 processor. The code for the experiments is available in the supplementary materials.

## I.1  Bias under strong ridge

we first consider a simple example: let $\lambda_k = (\frac{2k-1}{2}\pi)^{-1-a}$, $\psi_k(\cdot) = \sqrt{2}\sin\left(\frac{2k-1}{2}\pi\cdot\right)$ such that $\|\psi_k\|_{L^2_\mu} = 1$ for $\mu = \text{unif}[0,1]$; let $\theta_k^* = (\frac{2k-1}{2}\pi)^{-r}$. For $p = \infty$ and $a = 1$, the regression coincides with the kernel ridge regression with kernel $k(x, x') = \min\{x, x'\}$ defined on the interval $[0, 1]$ by [48]. [30, 33] have conducted similar experiments on this kernel $k$. However, to simulate the regression for independent features (IF), the feature rank $p$ has to be finite. In the following experiment, we choose $p = 2000$, the sample size $n$ ranges from 100 to 1000, ridge $\lambda = \lambda_n = (\frac{2n-1}{2}\pi)^{-1-a}$.

The first thing to check is whether both independent features ((IF)) and generic features ((GF)) satisfy

$$\mathcal{B} = \mathcal{O}\left(n^{-b\tilde{s}}\right)$$

under strong ridge, where $\tilde{s} \stackrel{\text{def.}}{=} \min s, 2$ and $s = \frac{2a+r}{1+a}$. To accurately obtain the bias term $\mathcal{B}$, we compute the exact formula:

$$\mathcal{B} = \lambda^2 \left\|(\lambda\mathbf{I}_p + \hat{\boldsymbol{\Sigma}})^{-1}\boldsymbol{\theta}^*\right\|_{\boldsymbol{\Sigma}}^2,$$

as shown in Proposition H.18, rather than computing the squared difference $\left(\hat{f}(x) - f^*(x)\right)^2$ by evaluating on test points as done in [31], or by using an integral function as in [33]. To demonstrate the Gaussian Equivalent Property (GEP), we also compute $\mathcal{B}$ after replacing the Sine feature vector $\boldsymbol{\psi} = \psi_k(x)_{k=1}^p$ with a random Gaussian vector $\mathbf{z} \sim \mathcal{N}(0, 1)$ or a random Rademacher vector $\mathbf{z} \sim (\text{unif}\{\pm 1\})^p$. It is worth noting that the random Rademacher vector $\mathbf{z} \sim (\text{unif}\{\pm 1\})^p$ satisfies Assumption (IF). This shows that our statement holds more generally than under the Gaussian Design Assumption ((GD)). From Figure 5, we observe that for different choices of $a$ and $r$ (and hence $s$), the bias decays at its theoretical rate for all three different features.

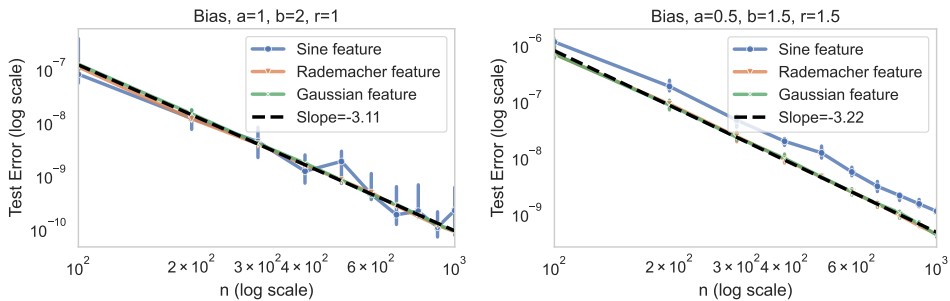

Figure 5: Decay of the bias term $\mathcal{B}$ under strong ridge $\lambda = \lambda_n = \Theta\left(n^{-1-a}\right)$. $\lambda_k = (\frac{2k-1}{2}\pi)^{-1-a}$, $\theta_k^* = (\frac{2k-1}{2}\pi)^{-r}$. Theoretical decay $\mathcal{B} = \mathcal{O}\left(n^{-(1+a)\tilde{s}}\right) = \mathcal{O}\left(n^{-(1+a)\tilde{s}}\right)$, where $\tilde{s} = \min\{s, 2\}$, source coefficient $s = \frac{2a+r}{1+a}$. (Left): $s = 1.5$ and $\mathcal{B} = \mathcal{O}\left(n^{-(1+1)\min\{1.5,2\}}\right) = \mathcal{O}\left(n^{-3}\right)$; (right): $s = 2.33 > 2$ and $\mathcal{B} = \mathcal{O}\left(n^{-(1+0.5)\min\{2.33,2\}}\right) = \mathcal{O}\left(n^{-3}\right)$, showing the saturation effect mentioned in [29]. All features demonstrate the same theoretical decay, validating the GEP.

## I.2 Bias under weak ridge

For weak ridge, we find that the decay rate is better than this theoretical rate (see Figure 6). One explanation for this is the estimation error of replacing the kernel $K(x, x') = \min\{x, x'\}$ by its finite rank approximation; another reason is that the decay flattens for large sample sizes $n \gg 1000$. However, the learning curve of independent features (Gaussian or Rademacher) behaves similarly to the dependent one (Sine). The left plot in Figure 6 fits our theoretical result, as $s > 1$, while the right plot shows that our theoretical bound is too pessimistic. However, we suspect this is due to the fact that the eigenfunctions on the one-dimensional input space are simply sines, which are uniformly bounded.

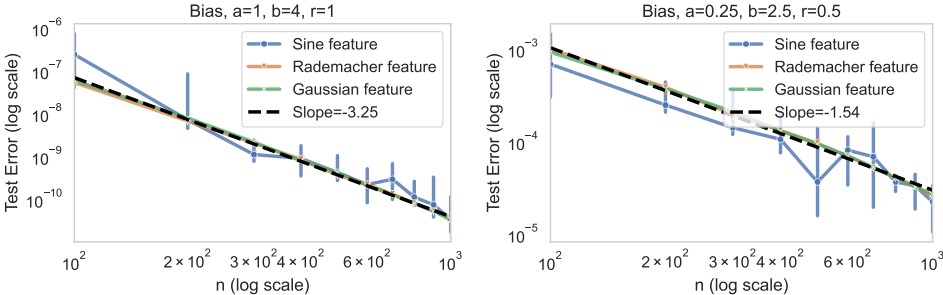

Figure 6: *Decay of the bias term $\mathcal{B}$ under weak ridge.* $\lambda = \Theta\left(n^{-b}\right)$. $\lambda_k = \left(\frac{2k-1}{2}\pi\right)^{-1-a}$, $\theta_k^* = \left(\frac{2k-1}{2}\pi\right)^{-r}$. Theoretical decay $\mathcal{B} = \mathcal{O}\left(n^{-(1+a)\tilde{s}}\right) = \mathcal{O}\left(n^{-(1+a)\tilde{s}}\right)$, where $\tilde{s} = \min\{s, 2\}$, source coefficient $s = \frac{2a+r}{1+a}$. (Left): $s = 1.5 > 1$ with theoretical bound $\mathcal{B} = \mathcal{O}\left(n^{-(1+0.5)\min\{2.33,2\}}\right) = \mathcal{O}\left(n^{-3}\right)$ for all features; (right): $s = 0.8 < 1$ with theoretical bound $\mathcal{B} = \mathcal{O}\left(n^{-(1+0.25)\min\{0.8,2\}}\right) = \mathcal{O}\left(n^{-1}\right)$ for Gaussian and Rademacher (independent) features, the empirical result for Sine features is better than its theoretical bound $\mathcal{B} = \mathcal{O}\left(n^{-(r-a)}\right) = \mathcal{O}\left(n^{-0.25}\right)$.

## I.3 Variance under strong ridge

Analogous to $\mathcal{B}$, we want to check whether both independent features ((IF)) and generic features ((GF)) satisfy

$$\mathcal{V} = \mathcal{O}\left(\sigma^2 n^{-1+\frac{b}{1+a}}\right),$$

under strong ridge, where $\sigma^2 = \mathbb{E}\left[\epsilon^2\right]$ is the noise level. To ease the computation, instead of computing the expression of $\mathcal{V}$ directly from Proposition H.19:

$$\mathcal{V} = \frac{\sigma^2}{n} \operatorname{Tr}\left[(\hat{\boldsymbol{\Sigma}} + \lambda \mathbf{I}_p)^{-1} \boldsymbol{\Sigma} (\hat{\boldsymbol{\Sigma}} + \lambda \mathbf{I}_p)^{-1} \hat{\boldsymbol{\Sigma}}\right]$$

$$= \frac{\sigma^2}{n} \operatorname{Tr}\left[\boldsymbol{\Sigma} (\hat{\boldsymbol{\Sigma}} + \lambda \mathbf{I}_p)^{-1} \hat{\boldsymbol{\Sigma}} (\hat{\boldsymbol{\Sigma}} + \lambda \mathbf{I}_p)^{-1}\right]$$

$$= \frac{\sigma^2}{n} \operatorname{Tr}\left[\boldsymbol{\Sigma} \mathbf{U} \mathbf{D}^2 (\mathbf{D} + \lambda \mathbf{I}_p)^{-2} \mathbf{U}^\top\right]$$

where $\hat{\boldsymbol{\Sigma}} = \mathbf{U} \mathbf{D} \mathbf{U}^\top$ is the singular value decomposition of $\hat{\boldsymbol{\Sigma}}$. Figure 7 confirms the Gaussian Equivalent Property (GEP) under strong ridge.

All dotted lines in Figures 5, 6 and 7 are regression of the learning curve with Gaussian features.

## I.4 Variance under weak ridge

As reported in [7, 14], by setting $\lambda = 0$, $\hat{f}$ is indeed the norm minimum interpolant, which may demonstrates tempered or catastrophic overfitting as $n \to \infty$. For this example, we focus on another setting where we take samples uniformly from a unit 2-disk and approximate $\mathcal{V}$ by evaluating the regressor of the zero function on the test point. We set $\lambda = 0$ and compute the kernel ridgeless

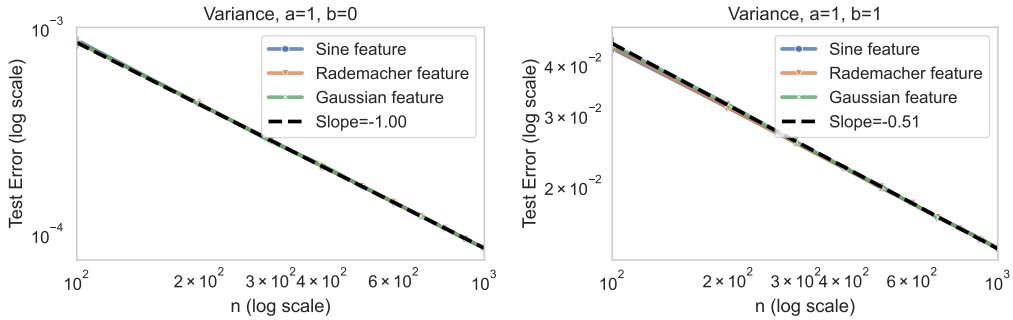

Figure 7: *Decay of the variance term $\mathcal{V}$ under strong ridge.* $\lambda = \Theta\left(n^{-b}\right)$. $\lambda_k = \left(\frac{2k-1}{2}\pi\right)^{-1-a}$. Theoretical decay $\mathcal{V} = \mathcal{O}\left(n^{-1+\frac{b}{1+a}}\right)$. (Left): Theoretical decay $\mathcal{V} = \mathcal{O}\left(n^{-1}\right)$ for all features; (right): Theoretical decay $\mathcal{V} = \mathcal{O}\left(n^{-1/2}\right)$ for all features.

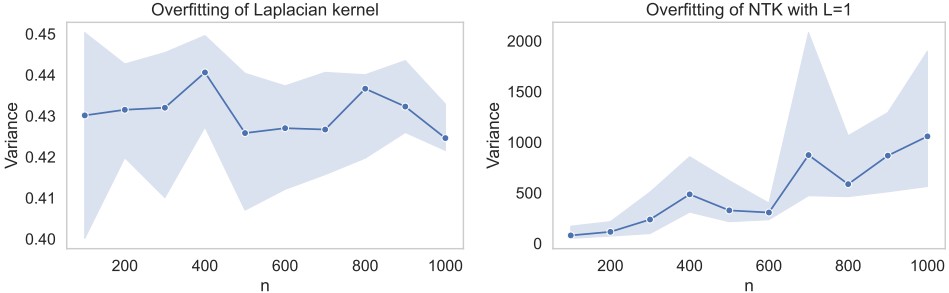

Figure 8: *The variance against the sample size $n$ under no ridge.* (Left): Tempered overfitting with Laplacian kernel. (Right): Catastrophic overfitting with NTK.

regression with the Laplacian kernel $K(x,z) = e^{-\|x-z\|_2}$ and the neural tangent kernel $K(x,z) = x^\top z \kappa_0(x^\top z) + \kappa_1(x^\top z)$ where $\kappa_0(t) \stackrel{\text{def.}}{=} 1 - \frac{1}{\pi}\arccos(t), \kappa_1(t) \stackrel{\text{def.}}{=} \frac{1}{\pi}\left(t(\pi - \arccos(t)) + \sqrt{1-t^2}\right)$.

In Figure 8, we can see that although having polynomial eigen-decay, the Laplacian kernel exhibits tempered overfitting, as termed by [35], while the NTK catastrophic overfitting.

# J   Tables

| Linear | Kernel |
|---|---|
| $\mathbf{x}$ | $K(x,\cdot)$ |
| $\mathbf{z}$ | $\psi_x$ |
| $\boldsymbol{\Sigma}$ | $\boldsymbol{\Lambda}$ |
| $\mathbf{Z}$ | $\boldsymbol{\Psi}$ |
| $\mathbf{X}\mathbf{X}^\top$ | $\mathbf{K}$ |
| $\lVert\cdot\rVert_2 = \lVert\cdot\rVert_{\mathbf{I}_p}$ | $\lVert\cdot\rVert_{\mathcal{H}}$ |
| $\lVert\cdot\rVert_{\boldsymbol{\Sigma}^{1-s}}$ | $\lVert\cdot\rVert_{\mathcal{H}^s}$ |
| $\lVert\cdot\rVert_{\boldsymbol{\Sigma}}$ | $\lVert\cdot\rVert_{L^2_\mu(\mathcal{X})}$ |
| $\lVert\cdot\rVert_\infty$ | $\lVert\cdot\rVert_\infty$ on $\mathrm{supp}(\mu)$ |
| $\boldsymbol{\theta}^*$ | $f^*$ |

Table 5: The translation of ridge regression setting in KRR setting (see Subsection A.6 for the elaboration.)

| Ridge | | strong | | weak | |
|---|---|---|---|---|---|
| Feature | | (IF) | (GF) | (IF) | (GF) |
| Poly (PE) | $\mathcal{B}$ | [16] | [31, 7] | [16] | [31, 7] |
| | $\mathcal{V}$ | [16] | [30, 7] | [16, 14] | [7] |
| Exp (EE) | $\mathcal{B}$ | this paper | [33] | this paper | [33] |
| | $\mathcal{V}$ | this paper | this paper | - | - |

| Ridge | | strong | | weak | |
|---|---|---|---|---|---|
| Feature | | (IF) | (GF) | (IF) | (GF) |
| Poly (PE) | $\mathcal{B}$ | - | $\mathcal{O}\left(n^{-b\bar{s}}\right)$ | - | $\mathcal{O}\left(n^{-(\min\{2(r-a),2-a\})_+}\right)$ |
| | $\mathcal{V}$ | - | $\tilde{\mathcal{O}}\left(\sigma^2 n^{-1+\frac{b}{a+1}}\right)$ | $\Theta\left(\sigma^2\right)$ | $\tilde{\mathcal{O}}\left(\sigma^2 n^{2a}\right)$ |
| Exp (EE) | $\mathcal{B}$ | - | - | - | - |
| | $\mathcal{V}$ | - | - | | catastrophic overfitting |

| Ridge | | strong | | weak | |
|---|---|---|---|---|---|
| Feature | | (IF) | (GF) | (IF) | (GF) |
| Poly (PE) | $\mathcal{B}$ | $\mathcal{O}\left(n^{-b\bar{s}}\right)$ | - | $\mathcal{O}\left(n^{-(1+a)\bar{s}}\right)$ | $\mathcal{O}\left(n^{-(1+a)\bar{s}}\right), s>1$ |
| | $\mathcal{V}$ | $\mathcal{O}\left(\sigma^2 n^{-1+\frac{b}{a+1}}\right)$ | - | - | - |
| Exp (EE) | $\mathcal{B}$ | - | $\mathcal{O}\left(e^{-b\bar{s}n}\right)$ | - | $\mathcal{O}\left(e^{-a\bar{s}n}\right), s>1$ |
| | $\mathcal{V}$ | - | - | | catastrophic overfitting |

| Ridge | | strong | | weak | |
|---|---|---|---|---|---|
| Feature | | (IF) | (GF) | (IF) | (GF) |
| Poly (PE) | $\mathcal{B}$ | $\Theta\left(n^{-b\bar{s}}\right)$ | $\Theta\left(n^{-b\bar{s}}\right)$ | $\Theta\left(n^{-(1+a)\bar{s}}\right)$ | $\begin{cases}\mathcal{O}\left(n^{-(1+a)\bar{s}}\right), & s>1 \\ \tilde{\mathcal{O}}\left(n^{-(\min\{2(r-a),2-a\})_+}\right), & s\le 1\end{cases}$ |
| | $\mathcal{V}$ | $\Theta\left(\sigma^2 n^{-1+\frac{b}{a+1}}\right)$ | $\Theta\left(\sigma^2 n^{-1+\frac{b}{a+1}}\right)$ | $\Theta\left(\sigma^2\right)$ | $\tilde{\mathcal{O}}\left(\sigma^2 n^{2a}\right), \Omega\left(\sigma^2\right)$ |
| Exp (EE) | $\mathcal{B}$ | $\Theta\left(e^{-b\bar{s}n}\right)$ | $\mathcal{O}\left(e^{-b\bar{s}n}\right)$ | $\mathcal{O}\left(e^{-a\bar{s}n}\right), s>1$ | $\mathcal{O}\left(e^{-a\bar{s}n}\right), s>1$ |
| | $\mathcal{V}$ | $\Theta\left(\sigma^2 n^{-1+\frac{b}{a}}\right)$ | $\mathcal{O}\left(\sigma^2 n^{-1+\frac{b}{a}}\right)$ | | catastrophic overfitting |

Table 6: Various filters for Table 1. (top): Recovered (in black), improved (in blue) and novel (denoted by "this paper") results over previous literature. (top 2): Recovered results under same assumptions. (top 3): Recovered results under weaker assumptions. (bottom): State-of-the-Art (SOTA) result. Black indicates results recovered results under our assumptions, blue indicates improved results over previous literature, orange indicates SOTA results from [31] under extra assumptions.

