# OpenReview forum: "A Comprehensive Analysis on the Learning Curve in Kernel Ridge Regression"
_NeurIPS.cc/2024/Conference — NeurIPS 2024 poster_

### Official Review · Reviewer_xgGe · 2024-07-07

**Soundness:** 2
**Presentation:** 2
**Contribution:** 2
**Rating:** 5
**Confidence:** 3

**Summary:**

This paper conducts a comprehensive study of the learning curves of kernel ridge regression (KRR) under minimal assumptions

**Strengths:**

The authors claimed that they provide a comprehensive analysis on the learning curves in kernel ridge regression.

**Weaknesses:**

The learning curves of kernel ridge regression have been extensively studied in recent literature. e.g.,
1. "Hugo Cui, Bruno Loureiro, Florent Krzakala, and Lenka Zdeborová. Generalization error rates in kernel regression: The crossover from the noiseless to noisy regime. Advances in Neural  Information Processing Systems, 34:10131–10143, 2021.",
2. "Yicheng Li, Haobo Zhang, and Qian Lin. On the asymptotic learning curves of kernel ridge  regression under power-law decay.", and
3.  "Bruno Loureiro, Cedric Gerbelot, Hugo Cui, Sebastian Goldt, Florent Krzakala, Marc Mezard,  and Lenka Zdeborová. Learning curves of generic features maps for realistic datasets with a  teacher-student model. Advances in Neural Information Processing Systems, 34:18137–18151,382 2021."

It is really hard for me to understand the difference between the current paper with the above papers.

**Questions:**

The writing style of this paper is quite challenging for me to engage with. I find the presentation could be improved in terms of clarity and readability.

For instance, could the author make their own results and assumptions in a more clear way? There is not even a single explicit theorem  provided for me to verify. It is quite unusual for a theoretical paper to be presented in this manner. I am uncertain if this document has been generated by an artificial intelligence.

---

> ### Author Rebuttal · Authors · 2024-08-02
>
> > The learning curves of kernel ridge regression have been extensively studied in recent literature...
> It is really hard for me to understand the difference between the current paper with the above papers.
>
> Thank you for your feedback. The three papers you mentioned are cited in the paper as references [16, 31, 34]. As explained in the paper, references [16, 34] assume the feature vectors are distributed according to Gaussian random vectors. This is what we call the IF case in our paper. These papers do not discuss the general feature case (GF). They also do not differentiate between exponential and polynomial decays and do not discuss differences between weak and strong regularization. Reference [31] employs a different assumption on the features, we have summarized the difference between their results and ours in Table 1 in the paper’s main body and in Table 5 in the appendix. For instance, Table 1 shows that our bounds are tighter than the results of [31].
>
> Since this is the only weakness noted by the reviewer, we would like to request further details on the reasoning behind the score of 3. Does our response above address your concerns? Are there any other questions or doubts we can address?
>
>
> > The writing style of this paper is quite challenging for me to engage with. I find the presentation could be improved in terms of clarity and readability.
> For instance, could the author make their own results and assumptions in a more clear way? There is not even a single explicit theorem provided for me to verify. It is quite unusual for a theoretical paper to be presented in this manner. I am uncertain if this document has been generated by an artificial intelligence.
>
>
> We would like to clarify that this paper was not written using ChatGPT or any other language model. We would appreciate it if the reviewer could specify which aspects of the presentation need to be improved.
>
> While we agree that some results are technical, we have tried to present them in the most reader-friendly way through Table 1 and Figure 1. For instance, Table 1 outlines the bounds we derived for three different scenarios:
> 1) polynomial vs. exponential eigendecay,
>  2) strong vs. weak regularization, and
> 3) independent vs. dependent features.
>
> All formal theorems are provided in the appendix, with Table 1 summarizing the key results, which we believe is a clearer way than listing each theorem individually. Additionally, Figure 1 illustrates the improvements achieved by these bounds in a diagram.
>
> Could the reviewer specify what aspects of the table and figure are unclear?
>
> Regarding the assumptions, we have a dedicated section (Section 2.2: Assumptions and Settings, line 96) that thoroughly covers them. Could you please further clarify your concerns? We are eager to improve the paper’s accessibility.

---

> > ### Comment · Reviewer_xgGe · 2024-08-10
> >
> > Thanks for clearing that up. It makes me feel more comfortable knowing the paper has not been written by an agent.
> >
> > I apologize if my previous comments were too harsh regarding the current presentation. The results are indeed interesting. Could you please revise your statement in a more formal manner? For example, you might present it as follows: 'Theorem: Under [specific assumptions], our results can be summarized in the following table...' Then, provide a brief description of the table. The current presentation might lead to unnecessary confusion regarding your results.
> >
> > I will increasing my score.

---

> > > ### Author Response · Authors · 2024-08-10
> > >
> > > Dear reviewer,
> > >
> > > Thank you for your constructive feedback! We sincerely appreciate your insights and we will reframe the results to improve their clarity. Since Neurips allows for an additional content page for the camera-ready version, we will move the most important theorem statements from the appendix to the main paper. We will then refer to the summary in the table as you suggested. Thank you once again.
> > >
> > > Best regards,
> > > The authors

---

### Official Review · Reviewer_v1Hz · 2024-07-07

**Soundness:** 3
**Presentation:** 3
**Contribution:** 3
**Rating:** 6
**Confidence:** 3

**Summary:**

A recent line of work has derived excess error rates for kernel ridge regression in a source and capacity setting under the assumption of Gaussian universality of the kernel features. This work investigates the validity of this assumption in this context.

The main result is that while the rates derived under Gaussian design are correct in the strong regularization regime, the excess error rate can be faster in the weak regularization regime.

**Strengths:**

This is a solid work. The manuscript is clearly written: the context is well explained and the reading is smooth. The results fit in an established literature studying excess error rates for kernel ridge, which recently has seen a revival of interest in the context of deep learning (NTK) and neural scaling laws. Therefore, I believe it is of significant interest to the theoretical community in NeurIPS.

**Weaknesses:**

A minor weakness of this work is that it mostly combines existing technical results. But I think this is minor since in the end the conclusions are novel and interesting

**Questions:**

- In the introduction, the authors say their work addresses three questions. While Q1 and Q2 are precisely addressed by the results, I find that the answer to Q3 falls short. First, the assumption (GF) is surely weaker, but it is still constraining. Second, some of the results in Table 1 are only upper bounds. I understand that most of the excess rate results in the classical kernel literature are also upper bounds, and the authors explicitly discuss that under (GF) it is not possible to derive a matching lower bound, but there is nothing telling us that the picture is not richer in these regimes. Perhaps my problem is with the phrasing of Q3, which differently from Q1 and Q2 is vague.

- I miss a discussion on the intuition behind the (GF) assumption. For instance, in the Gaussian design approximation, one possible intuition is the identification of "orthogonality" (of the features) with "independence". Do you have an intuitive understanding of the two conditions in L440? Why does strong regularization justifies independence?

- Related to the question above, how (GF) differs from the concentration assumption in previous work, e.g. (a1, a2, b1) in [36]. Note that the formulas derived under similar concentration conditions from [36] allow to recover exactly the Gaussian design rates from [16], see the contemporary recent work [DLM]. This suggests (GF) is strictly weaker?

- Since the main result in this work dialogues with previous literature, I suggest commenting and comparing the important notation (source, capacity, regularization decay, etc) in this work with the ones employed in the relevant Gaussian design literature, e.g. [10, 16, 35, 44]. For example, a table like Table 1 in [44] or Table 2 in [16].

- Can you please elaborate on Remark A.5?

[DLM] https://arxiv.org/abs/2405.15699

**Minor points**:

- The authors discuss the "over-parametrized" and "under-parametrized" regime in the text, but never define what they mean. While this can be inferred from the text, it would be good to precisely define it, since this terminology is used in different ways in the ML theory literature.

- For the sake of completeness, it would be better to add the definition of the source in (L106-L108) to the main text in a final version.

- Unpublished pre-prints in the bibliography are missing the arXiv identifiers.

- L110, define $\succeq$ in the notation section.

- L597, maybe $\psi_{k}$ and $\phi_{k}$ are switched?

- L634, "*Consider a kernel $\kappa:\mathcal{X}\times\mathcal{X}\to\mathbb{R}$ be a kernel with [...]*"

- Eq. below L651, missing right bracket.

- Assumption "Domain Regularity (DR)" in Appendix B.2 there are two items (i), (ii) but (iii) is mentioned in the paragraph below (L662-L669) twice.

- L679, precise what "$\lambda$" in $||\bar{\psi}_{i,j}||\lesssim \lambda^{(d-1)/4}$ is.

**Limitations:**

Limitations are discussed in the "Future potential work" section.

---

> ### Author Rebuttal · Authors · 2024-08-02
>
> > In the introduction, the authors say their work addresses three questions. While Q1 and Q2 are precisely addressed by the results, I find that the answer to Q3 falls short. First, the assumption (GF) is surely weaker, but it is still constraining.
>
> We refer to the **Author's Rebuttal** for a detailed answer.
>
> > ...some of the results in Table 1 are only upper bounds. I understand that most of the excess rate results in the classical kernel literature are also upper bounds, and the authors explicitly discuss that under (GF) it is not possible to derive a matching lower bound, but there is nothing telling us that the picture is not richer in these regimes...
>
> Thank you for raising this important question on a matching bound, which we did not have space to discuss in the main text.
> Previous literature which mentions GEP usually only focuses on the upper bound, however, we will put more discussion on the lower bound in a future edition.
> We refer to the **Author's Rebuttal** for a detailed answer.
>
> > I miss a discussion on the intuition behind the (GF) assumption... Do you have an intuitive understanding of the two conditions in L440?
>
> Again thank you for your question. The intuition behind (GF) is that the random distribution of the feature $\mathbf{z}$ is concentrated. By a Bernstein-type concentration inequality, one can show that independent features enjoy some concentration properties. See remark A.2 for more details.
>
> > Why does strong regularization justify independence?
>
> To be precise, strong regularization does not justify independence. Instead, we proved that the learning rates for the (GF) and (IF) cases are asymptotically identical under strong ridge, demonstrating a necessary condition for GEP.
>
> > ... how (GF) differs from the concentration assumption in previous work, e.g. (a1, a2, b1) in [36]... This suggests (GF) is strictly weaker?
>
> First, Assumption 1 in [36] requires the regularized tail rank to be bounded from below, in this paper’s notation, it is $\frac{\sum\_{l>k}\lambda\_l +\lambda}{\lambda\_{k+1}} \geq 2n$. Note that this cannot hold for exponential decay as LHS is bounded but not RHS. On the other hand, assumption (GF) does not have similar restrictions.
>
> Now we try to show how assumptions in [36] “imply” (GF).  As mentioned in Remark A.2 and A.3, the intuition of (GF) is the concentration of the truncated features at their expected value.
> Now suppose Assumption a2 in [36] holds, for any $\mathbf{x}\_{\leq k}=\Sigma\_{\leq k}^{1/2}\mathbf{z}\_{\leq k}$,  pick $\mathbf{A}=\Sigma\_{\leq k}^{-1}$ and we have w.h.p.
> $$ | \mathbf{x}\_{\leq k}^\top\Sigma\_{\leq k}\\mathbf{x}\_{\leq k} - tr[\Sigma\_{\leq k}\mathbf{A}] | \leq t \phi\_1(k) \\|\Sigma_{\leq k}^{1/2}\mathbf{A}\Sigma\_{\leq k}^{1/2}\\|\_{F} $$
> And hence, w.h.p.:
> $$| \\| \mathbf{x}\_{\leq k}\\|^2/k - 1 | \leq t \phi\_1(k), $$
> which implies the concentration of the first term in (GF).
>
> Similarly, Assumption 2 in [36] implies the concentration of the second term in (GF). At the moment, we do not know if the concentration of the third term in (GF) can be derived from any assumptions in [36], but as mentioned in Section A.2, the kernels satisfying the assumptions made in [36] form a proper subset of those which satisfies (GF).
>
> Hence, we believe our paper gives a more general result. If the reviewer could find any kernels which satisfies assumptions [36] but not (GF), please let us know as it would be of significant interest.
>
> > Since the main result in this work dialogues with previous literature, I suggest commenting and comparing the important notation (source, capacity, regularization decay, etc) in this work with the ones employed in the relevant Gaussian design literature...
>
> Thank you for your suggestion. We will put this into our consideration in our future edition to enhance readability and comparison with previous literature.
>
>
> > Can you please elaborate on Remark A.5?
>
> Suppose (IF) holds and each coordinate $z_l$ of the isotropic feature vector $\mathbf{z}$ is drawn i.i.d. from a random distribution with bounded support, hence there exists constants $a,b$ such that $z_l^2$ is a.s. In the interval $[a,b]$. By Hoeffding’s inequality, for any $t>0$,
> $$ \mathbb{P} \\{  | \\|\mathbf{z}\_{\leq k}\\|^2 - k   | > t  \\} \leq 2\exp(-2t^2/k(b-a))   $$
> Let  $t=\frac{k}{2}$, we have
> $$ \mathbb{P} \\{ \frac{3}{2} > \\|\mathbf{z}\_{\leq k}\\|^2/ k  > \frac{1}{2}  \\} \geq 1-  2\exp(-k/2(b-a))   $$
> Then w.h.p., the fraction $ \\|\mathbf{z}\_{\leq k}\\|^2/ k$ is concentrated at its expected value 1, similar argument holds for the other two fractions  $ \\| \mathbf{z}\_{> k}\\|\_{\Sigma\_{>k}}^2 / tr[\Sigma\_{>k}]$ and  $ \\| \mathbf{z}\_{> k}\\|\_{\Sigma\_{>k}^2}^2 / tr[\Sigma\_{>k}^2]$. By remark A.3, it suffices to show a weaker version of (GF). For technical simplicity, we require ess sup and ess inf in (GF) instead of a high probability bound.
>
> > **minor points: ...**
>
> Thank you very much for your effort improving our manuscript. We will adjust the points mentioned accordingly in our revised version.

---

> > ### Comment · Reviewer_v1Hz · 2024-08-11
> >
> > Thank you for your rebuttal, which address my main concerns. I maintain my score and recommendation towards acceptance.

---

### Official Review · Reviewer_x66w · 2024-07-12

**Soundness:** 3
**Presentation:** 3
**Contribution:** 3
**Rating:** 7
**Confidence:** 2

**Summary:**

This paper studies learning curves of kernel ridge regression (KRR) under minimal assumptions. The authors analyze the role of key properties of the kernel, such as its spectral eigen-decay, the characteristics of the eigenfunctions, and smoothness of the kernel. They also demonstrate the validity of the Gaussian Equivalent Property (GEP), which states that the generalization performance of KRR remains the same when the whitened features are replaced by standard Gaussian vectors. Additionally, they derive new improved bounds across several settings.

**Strengths:**

- The authors study the learning curves for various settings, including weak ridge vs strong ridge, independent features vs generic features, and polynomial vs exponential kernel eigenvalue decay. This comprehensive study provides a deeper understanding of the behavior of KRR in different scenarios.
- An improved bound is presented for the bias under the weak ridge assumption. In particular, the authors show that the generalization performance with independent (Gaussian) features and dependent (kernel) features coincides asymptotically and it solely depends on the eigen-decay under strong ridge regularization, hence validating the Gaussian Equivalent Property (GEP).
- The paper provides an answer to the key question "Under what condition the generalization error fully determined by the eigen-decay?"--- 1) in under-parameterized setting; or 2) with strong ridge in over-parameterized regime.

**Weaknesses:**

- The presented results seem to be based on a set of different assumptions while comparing with the current bounds. However, it is unclear how the discrepancy in the assumptions impacts the bounds. For example, the paper compares the presented bounds with related work using Hölder continuous kernels or the Embedding Condition, but it is not clear how the differences in assumptions affect the comparison.
- The numerical studies demonstrate the validity of the bounds for a couple of constructed kernels. However, it would be beneficial to consider more general/practical kernels to assess the practical impact of this work.

**Questions:**

Does the result hold under the assumptions used in the related work, e.g., Embedding Condition?

---

> ### Author Rebuttal · Authors · 2024-08-02
>
> Thank you for your time and effort reviewing our paper. Regarding your questions,
>
> > The presented results seem to be based on a set of different assumptions while comparing with the current bounds... it is not clear how the differences in assumptions affect the comparison.
>
> Section A.2 is dedicated to explaining and discussing the comparison between different assumptions from various papers. We kindly refer the reviewer to Section A.2 for more detailed information. We also refer to the **Author's Rebuttal** for a detailed answer.
>
> > The numerical studies demonstrate the validity of the bounds for a couple of constructed kernels. However, it would be beneficial to consider more general/practical kernels to assess the practical impact of this work.
>
> An important contribution of this paper is to provide a unifying theory connecting all scattered results from previous literature. We have derived the same optimal upper bound with a more general assumption.  Demonstrations on practical kernels can be seen in various previous literatures, including [12, 16, 31]. We will make this clear in the revision of the manuscript.
>
> >Does the result hold under the assumptions used in the related work, e.g., Embedding Condition?
>
> Yes. There is a detailed discussion in Section A.2 page 15 - 16.

---

> > ### Comment · Reviewer_x66w · 2024-08-11
> > **After rebuttal**
> >
> > Thanks for addressing my comments, I've raised my score accordingly.

---

### Official Review · Reviewer_8wF6 · 2024-07-15

**Soundness:** 3
**Presentation:** 3
**Contribution:** 4
**Rating:** 7
**Confidence:** 4

**Summary:**

This paper studies the learning curve of kernel ridge regression under both eigendecay assumption and the source condition assumption. Assuming the fixed input dimension setting, this paper derives the finite sample bound for the bias and variance where features can either be generic or independent. Depending on the assumptions used, matching lower bound are also provided. After this, the authors provide a unifying theory of the KRR test error and demonstrates the settings where Gaussian equivalence property holds. They also provide an answer for when the test error is fully determinded by the eigen-decay of the kernel. Finally, they provide simulation experiments to validate the bound on the test error.

**Strengths:**

By providing the matching upper and lower bounds for the test error under IF and GF where strong ridge is used, the paper shows that the Gaussian equivalence property only holds when strong ridge is used. In dosing so, the paper also provides a novel master inequality for both bias and variance.

In many cases, for example the expoenential eigendecay under strong ridge, the paper provides the sharpest learning rates so far.

The paper aslo addresses the question of when the learning rate of KRR is fully determined by the kernel eigenspectrum.

**Weaknesses:**

In demosntrating the GEP, the paper seems only provide the upper bound for bias and variance under generic features while a matching lower bound is missing. Given this, it is not completely convincing that the GEP holds although the author did show that the upper bound under GF matches with IF. So I was wondering if the author could explain this a bit or can detail the challenges in obtaining the lower bound.

Recently, there is a growing interest in studying the KRR when the output is infinite-dimensional, see e.g.
"Towards Optimal Sobolev Norm Rates for the Vector-Valued Regularized Least-Squares Algorithm." (2024)
&"Optimal Rates for Vector-Valued Spectral Regularization Learning Algorithms." (2024)

I am curious whether the results hold in the setting where the output is infinite dimensional. Maybe the author could draw some link between their results and this setting.

**Questions:**

See weakness

---

> ### Author Rebuttal · Authors · 2024-08-02
>
> Thank you for your time and effort reviewing our paper. Regarding your questions,
>
> > In demosnstrating the GEP, the paper seems to only provide the upper bound for bias and variance under generic features while a matching lower bound is missing... I was wondering if the author could explain this a bit or can detail the challenges in obtaining the lower bound.
>
> This is indeed a good question to ask for the lower bound in the (GF) case, which we did not have space to discuss in the main text. We refer to the **Author's Rebuttal** for a detailed answer.
>
> Previous works that mention GEP typically focus on the upper bound; hence, our paper also demonstrates GEP on the upper bound. However, the lower bound is complex, our paper is a first step understanding it better, and we will explore it further in a future work dedicated to these questions.
>
> We will update the paper to provide a detailed discussion of the aspects discussed in this answer as well as clarify the GEP claim, thank you for your comment.
>
> > Recently, there is a growing interest in studying the KRR when the output is infinite-dimensional...
> I am curious whether the results hold in the setting where the output is infinite-dimensional. Maybe the author could draw some link between their results and this setting.
>
> Thank you for pointing out this interesting setting. We were not aware of it. We looked at the proof technique and it seems it relies on an assumption similar to the embedding condition and relies on similar proof techniques (spectral calculus and concentration inequalities). A direct comparison seems difficult, as we for instance do not see a way to establish a comparison between different eigendecay and regularization regimes.
>
> That said, we do expect that many of these results could be extended to the infinite-dimensional setting, for instance, by projecting the target function to finite dimensions (via a Schauder basis truncation) then applying our results, and then (isometrically) embedding the learned (finite-rank) up be encoding it as weights to the chosen given Schauder basis.  In effect, one can view the current results as doing precisely this using the standard orthonormal basis of Euclidean space $\mathbb{R}^D$ as the Schauder basis and “truncating” at the full-dimension $D$ (in which case the truncation reduces precisely to the identity map, and the embedding is also the identity map).  This type of argument has successfully been used in the approximation-theory literature; see e.g. [1] for the Hilbert case (where the Schauder basis can be taken to be orthonormal) or [2] (in the Banach or even Fréchet case) where the basis is a genuine Schauder basis (eg a Wavelet basis when the target space is a Besov space over any E-thick domain with Lipschitz boundary (see Theorem Theorem 3.13 in [3] such as the Euclidean unit Euclidean ball).
>
> [1] Lanthaler, Samuel. "Operator learning with PCA-Net: upper and lower complexity bounds." Journal of Machine Learning Research 24.318 (2023): 1-67.
>
> [2] Galimberti, Luca, Anastasis Kratsios, and Giulia Livieri. "Designing universal causal deep learning models: The case of infinite-dimensional dynamical systems from the stochastic analysis." arXiv preprint arXiv:2210.13300 (2022).
>
> [3] Triebel, Hans. Function spaces and wavelets on domains. No. 7. European Mathematical Society, 2008.

---

> > ### Comment · Reviewer_8wF6 · 2024-08-08
> >
> > Thanks for replying. I will maintain my score.

---

### Author Rebuttal · Authors · 2024-08-02

Thank you for your time and effort on reviewing our paper. Since some of you asked for the same or similar questions concerning important points in our paper, we will refer the reviewers to here from their individual rebuttal sections.

---

## A matching lower bound for (GF)

we can summarize the existence of a matching lower bound in the cases considered in the paper with the following table:

| Ridge    | Strong                | Weak                    |
|----------|-----------------------|-------------------------|
| Feature  | (GF)                  | (GF)                    |
| Bias     | Yes, with modification in Prop. E.3 | Yes, if $1 \leq s \leq 2$ |
| Variance | Unknown               | No, because of Fig. 2   |

We discuss the results one by one:

- Under the strong ridge and (GF) assumptions, we could modify Prop. E.3 to obtain a matching lower bound for the bias. In more detail, in the second equation in the proof, we have $s_n(\mathbf{A}_{-l}) \geq n \lambda = \Omega(n \lambda_n)$ by the strong ridge assumption. The remainder of the proof follows similarly. We will include this modification in the revised version of the paper.

- Under the strong ridge and (GF) assumptions, we were not able to derive any matching lower bound, since the independent feature assumption (IF) seems necessary in the proof. However, based on results from the previous literature and under stronger assumptions, one could prove that the upper bound has a matching lower bound. See the orange-coloured result in Table 5.

- Under weak ridge and (GF) assumptions, we did prove a matching lower bound for the bias term with source coefficient $s \in [1,2]$. See Lemma E.1. For $s$ not in $[1,2]$, as far as we know, there is no matching lower bound result in the kernel setting. We would be happy if you could point out any results that we have overlooked.

- Under the weak ridge and (GF) assumptions, we demonstrated there is no matching lower bound for the variance term by Fig. 2, since there can be different overfitting phenomena for different kernels.

To address the comment from the reviewer, we will refine the statements made in the paper about GEP in the next revision, according to the above explanations.

Please see Section E for more details on the lower bounds.

---

## How general is the (GF) assumption?

Kernels satisfying (GF) includes:
- dot-product kernels on hyperspheres
- kernels with bounded eigenfunctions
- radial base function (RBF)
- shift-invariant kernels
- kernels on hypercubes.

As far as we know, the kernels satisfying the assumptions made in any existing theoretical papers on KRR learning rate (for instance [19,23,29-31,33,36]) form a proper subset of the above list. Although (GF) is constraining, it is clearly the most “general” in this sense. For more discussion, please see Section A.2.

---

Thank you again for reading our response. We are looking forward to hearing your feedback, and if possible, we kindly ask you for raising your score and confidence after you have considered our rebuttals.

---

### Decision · Program_Chairs · 2024-09-25

**Decision:**

Accept (poster)

**Comment:**

This paper proposes a unifying analysis on the learning rates of kernel ridge regression that connects existing results in the literature and offers several improvements. The reviewers unanimously recognized it as a valuable contribution after the rebuttal.

Several important suggestions were made by the reviewers to present this analysis in a more formal and complete manner, including a formal statement of the derived learning rates in the main text, a more precise reformulation of Q3 and a more explicit discussion on matching lower bounds. Please take into account of these points (some of which were addressed by the authors during the rebuttal) in the preparation of the final version.